# Pestalotioid Species Associated with Medicinal Plants in Southwest China and Thailand

Y. R. Sun,[a,b,c] R. S. Jayawardena,[b,c] J. E. Sun,[a] ⓘ Y. Wang[a]

[a]Department of Plant Pathology, College of Agriculture, Guizhou University, Guiyang, China
[b]Center of Excellence in Fungal Research, Mae Fah Luang University, Chiang Rai, Thailand
[c]School of Science, Mae Fah Luang University, Chiang Rai, Thailand

**ABSTRACT**   In this paper, a total of 26 pestalotioid isolates associated with different medicinal plants from southwest China and Thailand were studied. Based on morphological examinations and multigene analyses of three gene loci (*ITS*, *tef1-α*, and *tub2*), these 26 isolates represent 17 species distributed in three genera, including seven new species and eight new records. The concatenated three loci tree was used to infer the occurrence of sexual recombination within each pestalotioid genus through the pairwise homoplasy index (PHI) test implemented in SplitsTree. Further, simplifying the description of pestalotioid species is discussed, and a checklist for pestalotioid species associated with medicinal plants worldwide is provided.

**IMPORTANCE**   Pestalotioid species are an important fungal group, occurring commonly as plant pathogens, endophytes, and saprophytes. The study of pestalotioid species associated with medicinal plants is significant for agriculture, industry, and pharmaceutical industry but remains poorly studied. In this study, we report 17 pestalotioid species related to medicinal plants based on morphology and molecular analyses. Our study significantly enriches the species richness of pestalotioids and provides a basis for follow-up studies.

**KEYWORDS**  7 new species, 8 new records, diversity, endophytes, plant pathogens, phylogeny, saprophytes, taxonomy

Medicinal plants play a crucial role in the development of human cultures and are a rich source of natural products with both biological and chemical properties. Medicinal plants are used in health care, serve as treatments for various diseases, and have been used since prehistoric times across the world (1, 2). It is estimated that more than 70% of the world's population relies on medicinal plants (3). Microfungi can affect the growth and quality of medicinal plants. Some endophytes isolated from medicinal plants have broad developmental prospects (4, 5). Microfungi associated with medicinal plants are research hot spots (6–9).

Pestalotioid species are a very common group of fungi that form important associations with different plants as pathogens, endophytes, or saprophytes and are widely distributed in tropical and temperate regions (10–17). Traditional taxonomy of pestalotioid species mainly depend on their hosts and color intensities of the median conidial cell (10, 18–21). With the development of DNA-based phylogenetic analysis, the traditional classification system based on hosts and conidial colors is controversial. The use of molecular data in resolving pestalotioid species was revisited by Maharachchikumbura et al. (13), who separated this group into three genera, *viz.* *Neopestalotiopsis*, *Pestalotiopsis*, and *Pseudopestalotiopsis*. *Neopestalotiopsis* differs from *Pseudopestalotiopsis* and *Pestalotiopsis* by its versicolorous (two upper median cells darker than the lowest median cell) median cells and indistinct conidiophores, while *Pseudopestalotiopsis* can be easily distinguished

Address correspondence to Y. Wang, yongwangbis@aliyun.com.

The authors declare no conflict of interest.

from *Pestalotiopsis* by darker-colored concolorous (for those possessing equally pigmented median cells) median cells (13).

As important plant pathogens, pestalotioid species are almost ubiquitous in agricultural and noncultivated ecosystems, causing multiple diseases and huge economic losses (22–30). For example, gray blight disease of tea plants is caused by *Pseudopestalotiopsis* spp. and *Pestalotiopsis* spp. and accounts for at least 17% production damage in southern India (31) and 10 to 20% yield loss in Japan (32). *Neopestalotiopsis clavispora* caused the leaf blight of *Elettaria cardamomum* in India (33) and leaf spot of *Taxus chinensis* in China (34). Diogo et al. (22) reported that pestalotioid fungi caused stem girdling and dieback in young eucalyptus plants in Portugal. Li et al. (35) identified five new pestalotioid species associated with symptomatic leaves of *Camellia oleifera* in China. Thus, it is necessary to study the pathogenic pestalotioid species related to medicinal plants, which could provide the research foundation for the prevention and treatment of diseases and reduce economic losses.

The study of endophytic fungi in medicinal plants is of great significance for elucidating their distribution, growth and developmental characteristics, and resource regeneration (6, 7, 9, 36, 37). Many pestalotioid fungi have been found as endophytes from different medicinal plants with rich secondary metabolites (5, 36, 38–41). For example, the endophytic fungus *Pestalotiopsis versicolor* was isolated from the healthy leaves of *Taxus cuspidata*, and it is an excellent candidate for an alternate source of Paclitaxel supply (42). Therefore, the study of endophytic pestalotioid species related to medicinal plants could be of great importance to pharmaceuticals and therapeutic medicine.

This study aims to identify the pestalotioid fungi associated with medicinal plants in southwest China and Thailand based on morphology and molecular analyses. This paper describes, illustrates, and compares seven new species and eight new records with allied species. In addition, we provide a checklist for pestalotioid species associated with medicinal plants worldwide.

## RESULTS

***Neopestalotiopsis* Maharachch., K.D. Hyde, and Crous, Stud. Mycol. 79:135 (2014). (i) Phylogenetic analyses.** The combined data sets consist of 100 *Neopestalotiopsis* strains along with the outgroup *Pestalotiopsis diversiseta* (MFLUCC 12–0287) and *P. spathulata* (CBS 356.86), which were analyzed to infer the interspecific relationships within *Neopestalotiopsis*. The aligned sequence matrix comprised internal transcribed spacers (*ITS*; 1 to 485), translation elongation factor 1 (*tef1-α*; 486 to 982), and partial *β*-tubulin region (*tub2*; 983 to 1,423), sequence data for a total of 1,423 sites, including coded alignment gaps. Similar tree topologies were obtained by maximum likelihood (ML) and Bayesian posterior probability (BYPP) methods, and the most likely tree (−ln = 7671.251111) is presented in Fig. 1. The phylogenetic tree, which analyzed the 15 *Neopestalotiopsis* isolates from medicinal plants, indicated four novel species and three new records.

**(ii) Genealogical concordance phylogenetic species recognition analysis.** The pairwise homoplasy index (PHI) test revealed that there is no significant recombination ($\Phi w$ = 0.06), between *N. amomi* and its closely related taxa *N. eucalypticola* (CBS 264.37), *N. magna* (MFLUCC 12-0652), and *N. zingiberis* (GUCC 21001) (Fig. 2a). Additionally, based on the PHI test results, there is no significant recombination ($\Phi w$ = 0.11) between *N. hyperici* and its closely related taxa *N. acrostichi* (MFLUCC 17–1754), *N. lusitanica* (MEAN 1320), *N. protearum* (CBS 114178), *N. rhododendricola* (KUN-HKAS 123204), and *N. rhododendri* (GUCC 21504) (Fig. 2b). Similar result also occurred in *N. photiniae* ($\Phi w$ = 1.0) (Fig. 2c) and *N. suphanburiensis* ($\Phi w$ = 1.0) (Fig. 2d), indicating there is no significant recombination between them and their closely related taxa.

**(iii) Taxonomy.** *(a) Neopestalotiopsis amomi* Y.R. Sun and Yong Wang bis, sp. nov. Fungal names number: FN 571230; Facesoffungi number: FoF 12912 (Fig. 3).

Etymology: refers to the name of the host plant from which the fungus was isolated.

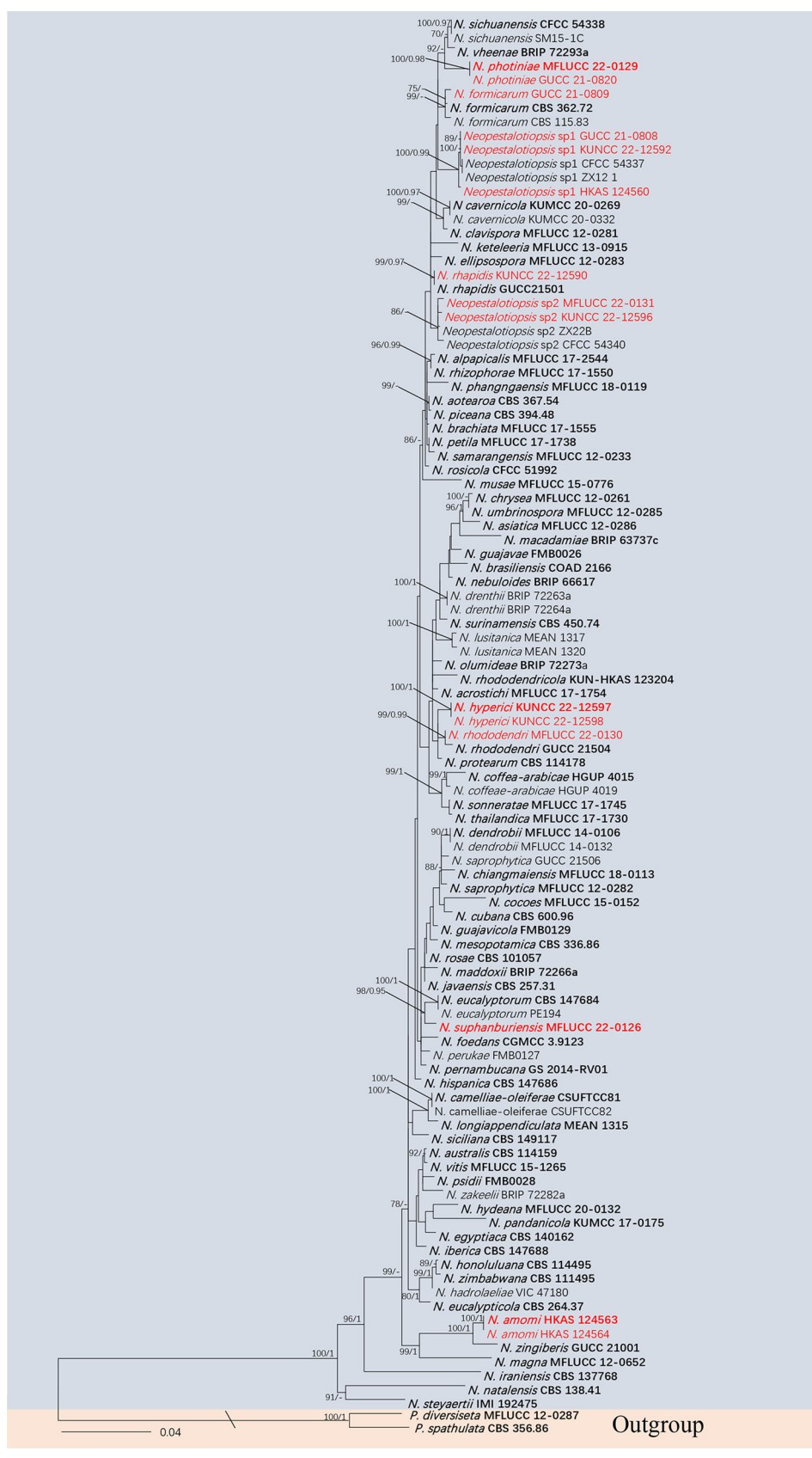

**FIG 1** Maximum likelihood (RAxML) tree for *Neopestalotiopsis* based on the analysis of a combined data set of *ITS*, *tef1-α*, and *tub2* sequence data. The tree is rooted with *Pestalotiopsis diversiseta* (MFLUCC 12-0287) and *P. spathulata* (CBS 356.86). Bootstrap support values for ML greater than 75% and Bayesian posterior probabilities greater than 0.95 are given near nodes, respectively. The new isolates are in red, and ex-type strains are in bold.

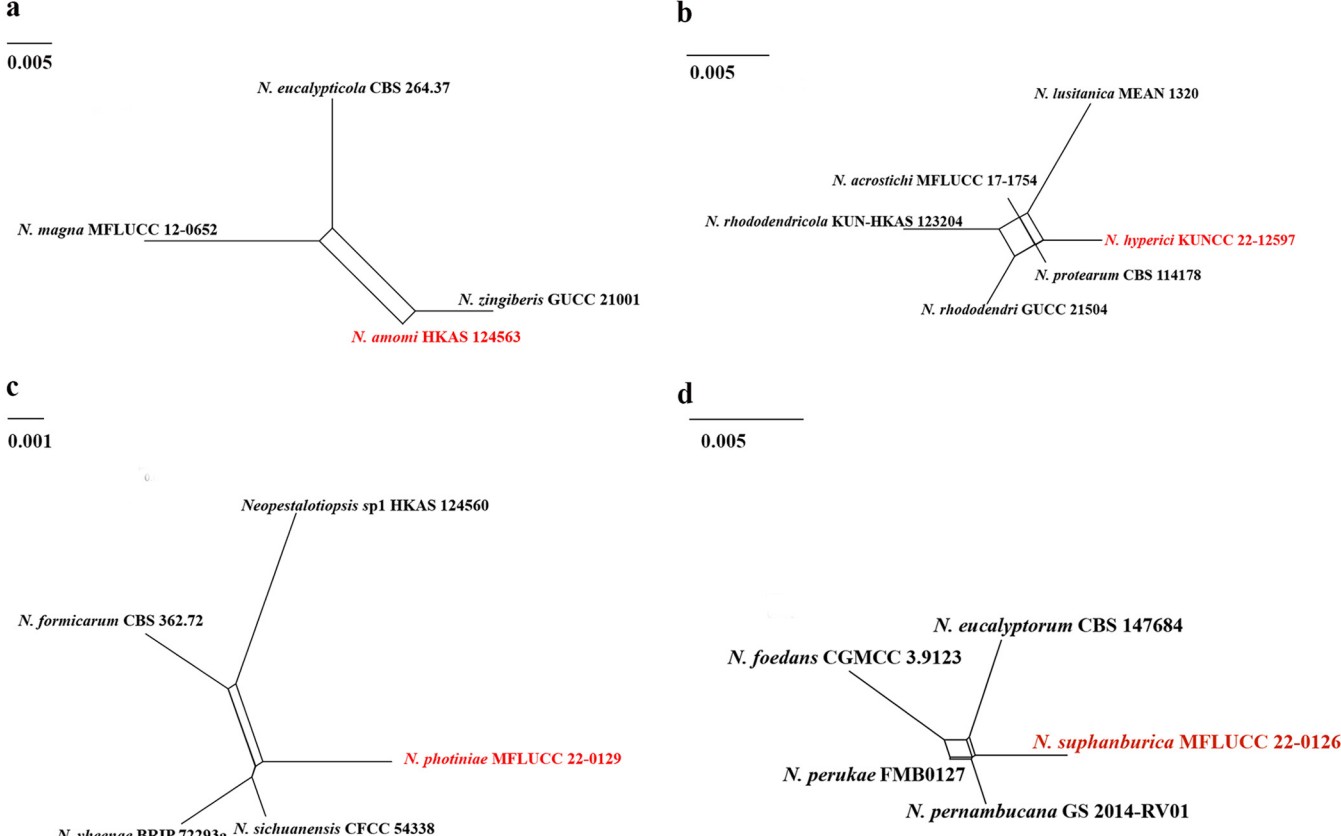

**FIG 2** (a, b, c and d) Split graphs showing the results of a PHI test of new *Neopestalotiopsis* species with their most closely related species using Log-Det transformation and splits decomposition options. The new taxon in each graph is shown in red font.

Holotype: HKAS 124563.

Associated with leaf blight of *Amomum villosum*. Symptoms: irregular shape, pale to brown, slightly sunken spots appear on the leaves of *Amomum villosum*, which later expand outward. Sexual morph: not observed. Asexual morph: conidiomata solitary, subglobose to globose, unilocular, brown, semi-immersed on leaves. Conidiophores 3 to 5 $\mu$m long, often reduced to conidiogenous cells. Conidiogenous cells 1 to 2 $\mu$m wide, subcylindrical, ampulliform, hyaline. Conidia 18 to 30 × 4 to 7 $\mu$m ($\bar{x}$ = 25 × 6 $\mu$m, $n$ = 40), length/width (L/W) ratio of 4.2, fusiform, straight to slightly curved, 4 septate; basal cell obconic with a truncate base, hyaline, smooth walled, 3 to 7 $\mu$m long; three median cells 12 to 19 $\mu$m long ($\bar{x}$ = 16 $\mu$m, $n$ = 40), pale brown to brown, concolorous, wall rugose, septa darker than the rest of the cell; second cell from base pale brown to brown, 3 to 8 $\mu$m long; third cell pale brown to brown, 3 to 7 $\mu$m long; fourth cell pale brown to brown, 3 to 7 $\mu$m long; apical cell 2 to 5 $\mu$m long, hyaline, conic to acute; with 2 to 3 tubular appendages on the apical cell, inserted at different loci in a crest at the apex of the apical cell, unbranched, 7 to 17 $\mu$m long; single basal appendage, unbranched, tubular, centric, 2 to 5 $\mu$m long.

Culture characteristics: conidia germinated on potato dextrose agar (PDA) within 12 h from single-spore isolation. Colony diameter reached 8 cm after 2 weeks at 25°C on PDA medium and appeared circular, with a flat, rough surface and was white from above and below.

Material examined: China, Guizhou Province, Qiannan Bouyei and Miao Autonomous Prefecture, Luodian District, leaf blight of *Amomum villosum* (Zingiberaceae), 3 September 2021, Y.R. Sun, L8 (HKAS 124563, holotype); ibid., on leaf blight of *Amomum villosum*, 3 September 2021, Y.R. Sun, L8-1 (HKAS 124564).

Notes: *Neopestalotiopsis amomi* was isolated from the diseased leaves of *Amomum villosum* in China. Two collections HKAS 124563 and HKAS 124564 clustered together

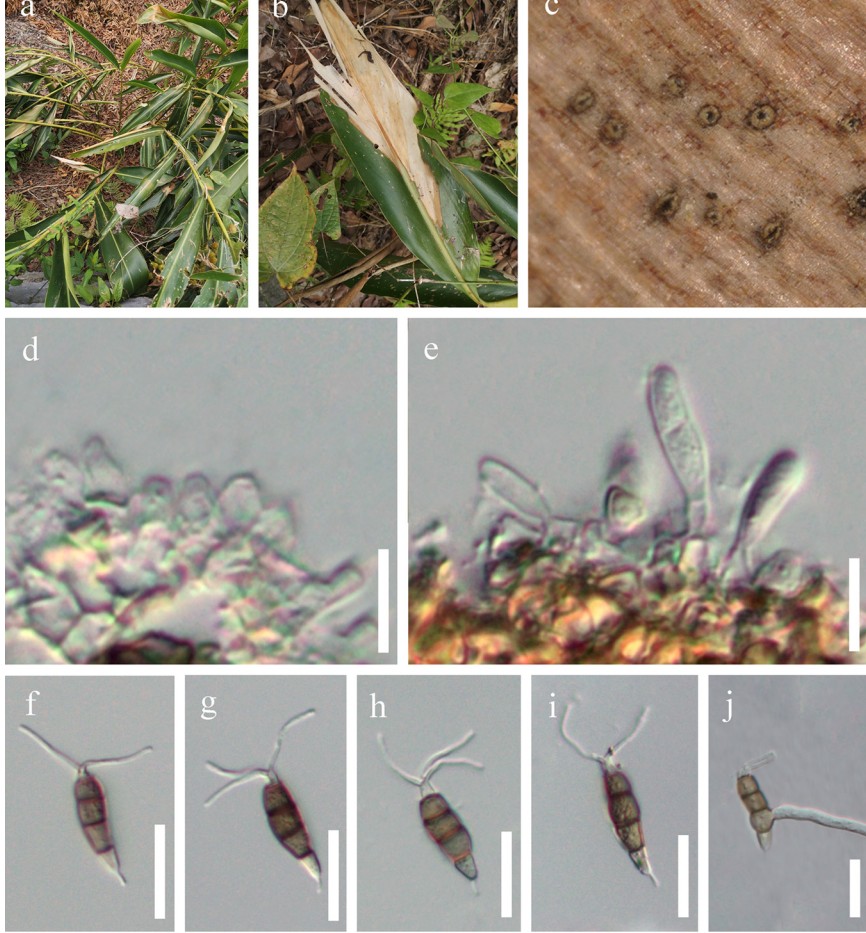

**FIG 3** *Neopestalotiopsis amomi* (HKAS 124563, holotype). (a) Host. (b) Leaf blight on *Acrostichum aureum*. (c) Close-up view of conidiomata. (d) Conidiogenous cells. (e) Immature conidia attached to conidiogenous cells. (f to i) Conidia. (j) Germinated conidium; scale bars: 10 $\mu$m (d and e) and 20 $\mu$m (f to j).

with good support (ML-BS = 100%, BYPP = 1) and formed a sister clade to *N. zingiberis* (GUCC 21001), which was also isolated from a *Zingiberaceae* plant (43). The former differs in producing thinner conidia (4 to 7 $\mu$m in *N. amomi* versus 6 to 9.5 $\mu$m in *N. zingiberis*) and shorter conidiophores (3 to 5 $\mu$m in *N. amomi* versus 12 to 25 $\mu$m in *N. zingiberis*). In addition, there are 4 bp different between HKAS 124563 and GUCC 21001 in the *ITS* gene and 10 bp different in the *tef1-α* gene. *Neopestalotiopsis amomi* also differs by smaller conidia (18 to 30 × 4 to 7 $\mu$m versus 42 to 46 × 9.5 to 12 $\mu$m) from *N. magna*. The PHI test on *N. amomi* indicated that there is no significant recombination ($\Phi w$ = 0.06) between *N. amomi* and its closely related taxa. Thus, we introduce *N. amomi* as a new species.

*(b) Neopestalotiopsis hyperici* Y.R. Sun and Yong Wang bis, sp. nov. Fungal names number: FN 571228; Facesoffungi number: FoF 12913 (Fig. 4).

Etymology: the specific epithet is referring to *Hypericum*, the host plant that the fungus was isolated from.

Holotype: HKAS 124561.

Associated with leaf spots of *Hypericum monogynum*. Symptoms: irregular shape, pale to brown, slightly sunken spots appear on the leaves of *Hypericum* sp., which later expand outward. Sexual morph: not observed. Asexual morph: conidiomata solitary, unilocular, dark. Conidiophores often reduced to conidiogenous cells. Conidiogenous cells indistinct. Conidia 17 to 22 (to 24) × 5 to 8 $\mu$m ($\overline{x}$ = 19 × 7 $\mu$m, $n$ = 30), L/W ratio of 2.8, fusoid, subcylindrical, straight to slightly curved, 4 septate; basal cell conic to obconic with a truncate

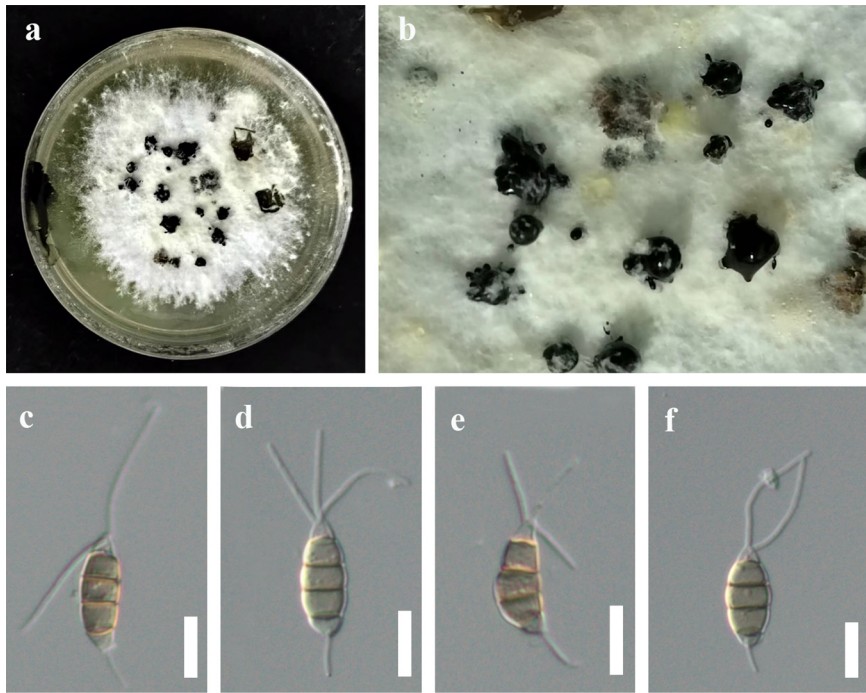

**FIG 4** *Neopestalotiopsis hyperici* (HKAS 124561, holotype). (a) Culture. (b) Close-up view of conidiomata. (c to f) Conidia; scale bars: 10 $\mu$m (c to f).

base, hyaline to subhyaline, 2 to 4 $\mu$m long; three median cells 10 to 14 (to 17) $\mu$m long ($\bar{x}$ = 12 $\mu$m, $n$ = 30), wall rugose, concolorous; second cell from base pale brown to brown, 3 to 5 $\mu$m long; third cell pale brown to brown, 3 to 6 $\mu$m long; fourth cell pale brown to brown, 2 to 6 $\mu$m long; apical cell 1 to 4 $\mu$m long, hyaline, rugose and thin walled; with 2 to 3 tubular apical appendages, arising from the apical crest, unbranched, filiform, 11 to 23 $\mu$m long; single basal appendage 4 to 7 $\mu$m long, unbranched, tubular, centric.

Culture characteristics: colonies on PDA reached up to 10 cm after 2 weeks, dense mycelium was on the surface, and colonies appeared white from above and below. Fruiting bodies were observed after 14 days.

Material examined: China, Guizhou Province, Guiyang City, Baiyun District, Changpoling National Forest Park, leaf spot of *Hypericum monogynum* (*Clusiaceae*), 20 August 2021, Y.R. Sun, CL5-1 (HKAS 124561, holotype); ex-type culture, KUNCC 22-12597 = GUCC 21-0812; ibid., on leaf spots of *Hypericum monogynum*, 20 August 2021, Y.R. Sun, CL5-1-1, living culture KUNCC 22-12598 = GUCC 21-0811.

Notes: *Neopestalotiopsis hyperici* is related to *N. rhododendri* and *N. protearum* in the phylogenetic analysis (Fig. 1), but they can be distinct from concolorous conidia and the size of their median cells (10 to 14 $\mu$m in *N. hyperici* versus 14 to 20 $\mu$m in *N. rhododendri* versus 16 to 17 $\mu$m in *N. protearum*) (13, 44). In addition, there are 13 bp different between *N. hyperici* and *N. rhododendri* and 10 bp different between *N. hyperici* and *N. protearum* in the *tef1-α* region. Moreover, the PHI test on *N. hyperici* indicated that there is no significant recombination ($\Phi$w = 0.11) between *N. hyperici* and its closely related taxa (Fig. 2b). Thus, we introduce *N. hyperici* as a new species.

*(c) Neopestalotiopsis photiniae* Y.R. Sun and Yong Wang bis, sp. nov. Fungal names number: FN 571231; Facesoffungi number: FoF 12914 (Fig. 5).

Etymology: referring to the host plant from which the fungus was isolated.

Holotype: HKAS 125895.

Associated with leaf spots of *Photinia serratifolia*. Symptoms: irregular shape, pale to brown, slightly sunken spots appear on the leaves of *Photinia serratifolia*, which later expand outward. Small spots gradually enlarged, changing to brown circular ring spots

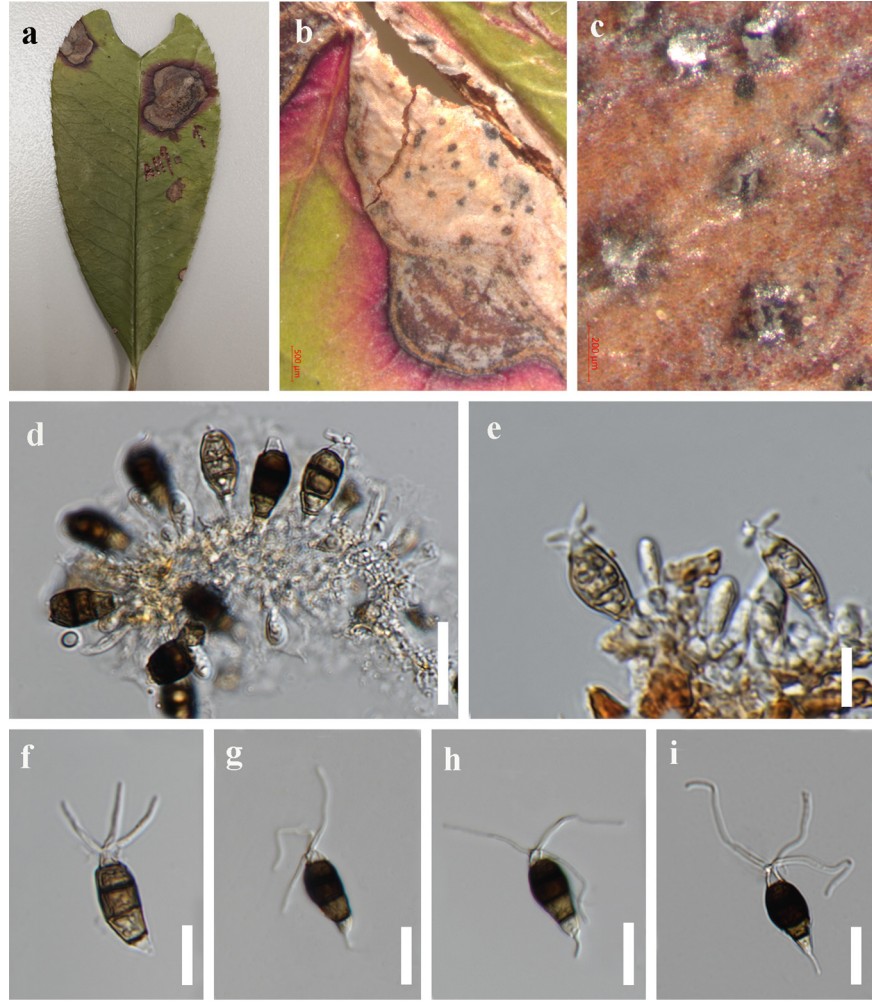

FIG 5 *Neopestalotiopsis photiniae* (HKAS 125895, holotype). (a) Host. (b) Leaf spot on *Photinia serratifolia*. (c) Close-up view of conidiomata. (d and e) Conidia attached to conidiogenous cells. (f to i) Conidia; scale bars, 200 $\mu$m (c), 20 $\mu$m (d), and 10 $\mu$m (f to i).

with a dark brown border. Sexual morph: not observed. Asexual morph: conidiomata solitary, subglobose to globose, unilocular, dark brown, semi-immersed on leaves. Conidiophores indistinct, often reduced to conidiogenous cells. Conidiogenous cells 1 to 3 $\times$ 2 to 4 $\mu$m, subcylindrical, ampulliform, hyaline. Conidia 20 to 29 $\times$ 5 to 12 $\mu$m ($\overline{x}$ = 23 $\times$ 9 $\mu$m, $n$ = 40), L/W ratio of 2.6, broadly fusiform, straight to slightly curved, 4 septate; basal cell obconic with a truncate base, hyaline to pale brown, 1 to 5 $\mu$m long; three median cells 13 to 19 $\mu$m long ($\overline{x}$ = 16 $\mu$m, $n$ = 40), brown to dark, wall rugose, versicolorous; second cell from base pale brown to brown, 4 to 6 $\mu$m long; the third and fourth cells, dark brown to black, are not easily distinguished, septate indistinct, 10 to 13 $\mu$m long; apical cell 2 to 4 $\mu$m long, hyaline, conic to acute; with 2 to 3 tubular appendages on the apical cell, inserted at different loci in a crest at the apex of the apical cell, unbranched, 17 to 33 $\mu$m long; single basal appendage, unbranched, tubular, centric, 1 to 6 $\mu$m long.

Culture characteristics: conidia germinated on PDA within 12 h at 25°C from single-spore isolation. Apical cells produced germ tubes. Colony diameter reached 80 mm after 3 weeks at 25°C on PDA medium, were circular with a rough, flat surface, and appeared white from above and below.

Material examined: China, Guizhou Province, Guiyang City, Nanming District, Xiaochehe Road, Guiyang Ahahu National Wetland Park, on leaf spots of *Photinia serratifolia* (*Rosaceae*), 21 September 2019, Y.R. Sun, AH9 (HKAS 125895, holotype); ex-type

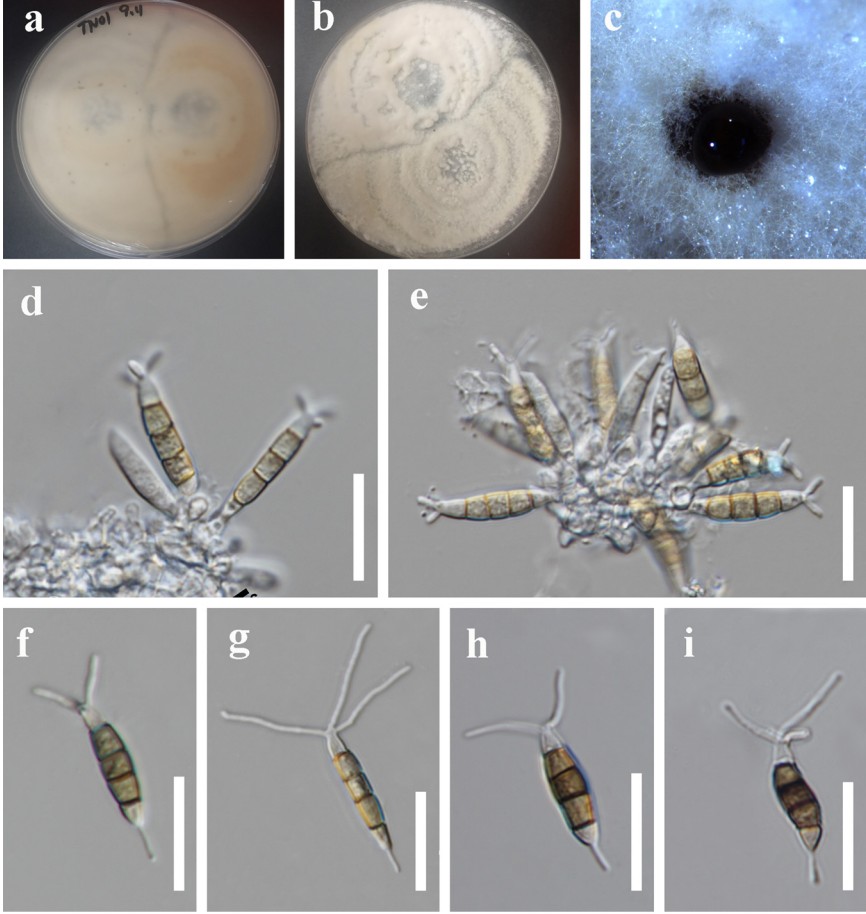

**FIG 6** *Neopestalotiopsis suphanburiensis* (MFLU 22-0168, holotype). (a and b) Cultures. (c) Colony in culture. (d and e) Conidia attached to conidiogenous cells. (f to i) Conidia; scale bars, 10 $\mu$m (d to i).

culture, MFLUCC 22-0129; ibid., on leaf spots of *Photinia* sp. (*Rosaceae*), 21 September 2019, Y.R. Sun, AH9-1, living culture, GUCC 21-0820.

Notes: *Neopestalotiopsis photiniae* is phylogenetically sister to *N. sichuanensis* and *N. vheenae* (Fig. 1). *Neopestalotiopsis photiniae* differs by its thinner conidia (L/W ratio = 2.6 versus L/W ratio = 4.1) from *N. sichuanensis* (45). *Neopestalotiopsis photiniae* is morphologically distinguishable from *N. vheenae* in shorter conidiogenous cells (1 to 3 × 2 to 4 $\mu$m versus 5 to 10 × 3 to 5 $\mu$m) (23). The result of the PHI test showed that there is no obvious recombination ($\Phi$w = 1.0) between *N. photiniae* and its closely related taxa (Fig. 2c). Therefore, *N. photiniae* is introduced as a new species.

*(d) Neopestalotiopsis suphanburiensis* Y.R. Sun and Yong Wang bis, sp. nov. Fungal names number: FN 571232; Facesoffungi number: FoF 12916 (Fig. 6).

Etymology: refers to the province where the fungus was collected, Suphan Buri Province.

Holotype: Mae Fah Luang University (MFLU) 22-0168.

Saprobic on stems of an unidentified plant. Sexual morph: not observed. Asexual morph: conidiomata solitary, subglobose to globose, unilocular, brown to dark, immersed on stems. Conidiophores indistinct, often reduced to conidiogenous cells. Conidiogenous cells subcylindrical, ampulliform, hyaline. Conidia 19 to 29 × 4 to 7 $\mu$m ($\overline{x}$ = 25 × 5 $\mu$m, n = 40), L/W ratio = 4.9, fusiform, straight to slightly curved, 4 septate; basal cell obconic with a truncate base, hyaline, smooth walled, 3 to 7 $\mu$m long; three median cells 12 to 19 $\mu$m long ($\overline{x}$ = 16 $\mu$m, n = 40), pale brown to brown, wall rugose, concolor, septa darker than the rest of the cell, versicolorous; second cell from base pale brown to brown, 3 to 7 $\mu$m long; third cell pale brown to brown, 3 to 6 $\mu$m long;

fourth cell pale brown to brown, 4 to 6 $\mu$m long; apical cell 3 to 6 $\mu$m long, hyaline, conic to acute; with 2 to 3 tubular appendages on the apical cell, inserted at different loci in a crest at the apex of the apical cell, unbranched, 9 to 21 $\mu$m long; single basal appendage, unbranched, tubular, centric, 2 to 11 $\mu$m long.

Culture characteristics: colony diameter reached 8 cm after 2 weeks at 25°C on PDA medium. Colonies were circular with a rough, flat surface and appeared white from above and white to pale gray from below.

Material examined: Thailand, Suphan Buri Province, dead stem of an unidentified plant, 5 September 2020, S Wang, TN01 (MFLU 22-0168, holotype); ex-type culture, MFLUCC 22-0126.

Notes: *Neopestalotiopsis suphanburiensis* is phylogenetically sister to *N. eucalypto-rum*, which was isolated from leaves and stems of *Eucalyptus globulus* (Fig. 1). In morphology, *N. suphanburiensis* differs from *N. eucalyptorum* in having thinner conidia (4 to 7 $\mu$m versus 7.6 to 8.1 $\mu$m). In addition, there are 10 bp different (without gap, 445 bp) in the *tef1-α* region. The PHI test on *N. suphanburiensis* also indicated that there is no significant recombination ($\Phi$w = 1.0) between *N. suphanburiensis* and its closely related taxa (Fig. 2d). We thus introduce *N. suphanburiensis* as a new species.

*(e) Neopestalotiopsis* sp. 1. Associated with leaf spot of *Cyrtomium fortunei, Lithocarpus* sp., and *Smilax scobinicaulis* (Fig. 7). Symptoms: irregular shape, pale brown, small spots gradually enlarged, changing to brown circular ring spots with a dark brown border. Sexual morph: not observed. Asexual morph: conidiomata solitary, subglobose to globose, unilocular, dark brown, semi-immersed on leaves. Conidiophores indistinct, often reduced to conidiogenous cells. Conidiogenous cells subcylindrical or ampulliform, hyaline. Conidia 21 to 31 × 4 to 7 $\mu$m ($\overline{x}$ = 26 × 6 $\mu$m, n = 30), L/W ratio of 4.4, fusiform, straight to slightly curved, 4 septate; basal cell obconic with a truncate base, hyaline, 3 to 7 $\mu$m long; three median cells doliiform to cylindrical, 11 to 18 $\mu$m long ($\overline{x}$ = 15 $\mu$m, n = 30), yellow to brown, concolorous, septa darker than the rest of the cell; second cell from base yellow to brown, 3 to 6 $\mu$m long; third cell yellow to brown, 3 to 7 $\mu$m long; fourth cell yellow to brown, 4 to 7 $\mu$m long; apical cell 2 to 6 $\mu$m long, hyaline, conic to acute; with 1 to 4 tubular appendages on apical cell, inserted at different loci in a crest at the apex of the apical cell, unbranched, 13 to 26 $\mu$m long; single basal appendage, unbranched, tubular, centric, 2 to 7 $\mu$m long.

Culture characteristics: conidia germinated on PDA within 12 h at 25°C from single-spore isolation. Apical cells produced germ tubes. Colony diameter reached 80 mm after 2 weeks at 25°C on PDA medium. Colonies were circular with a rough, flat surface and appeared white from above and yellow from below.

Material examined: China, Guizhou Province, Tongren City, Jiangkou District, Yamugou Parkland, leaf spot of *Lithocarpus* sp. (*Fagaceae*), 20 May 2022, Y.R. Sun, JK15-2 (HKAS 124565); living culture, KUNCC 22-12592 = GUCC 21-0808; China, Guizhou Province, Qiannan Bouyei and Miao Autonomous Prefecture, Libo District, leaf spot of *Smilax china* (*Liliaceae*), 12 March 2022, Y.R. Sun, bb1 (HKAS 124560); China, Guizhou Province, Guiyang City, Baiyun District, Changpoling National Forest Park, leaf spot of *Dryopteris crassirhizoma* (*Dryopteridaceae*), 20 August 2021, Y.R. Sun, CL1-2, living culture, GUCC 21-0813.

Notes: three strains (KUNCC 22-12592, HKAS 124560, and GUCC 21–0813) have identical *ITS, tef1-α*, and *tub2* sequences as isolates CFCC-54337 and ZX12-1, which were previously provided by Jiang et al. (45). However, they did not introduce it as a new species due to lack of distinguished characters from close clades. In this study, these five strains clustered together and formed a distinct clade in the tree. We keep these five strains as *Neopestalotiopsis* sp.1 for the same reasons as before. We speculate that *Neopestalotiopsis* sp.1 could be a common phytopathogen, as it has been found in leaf spots on different plants.

*(f) Neopestalotiopsis* sp. 2. Saprobic on *Ceiba pentandra* leaves and endophytic from *Pinellia ternata* (Fig. 8). Sexual morph: not observed. Asexual morph: conidiomata solitary, unilocular, dark, immersed on stems. Conidiophores indistinct, often reduced to

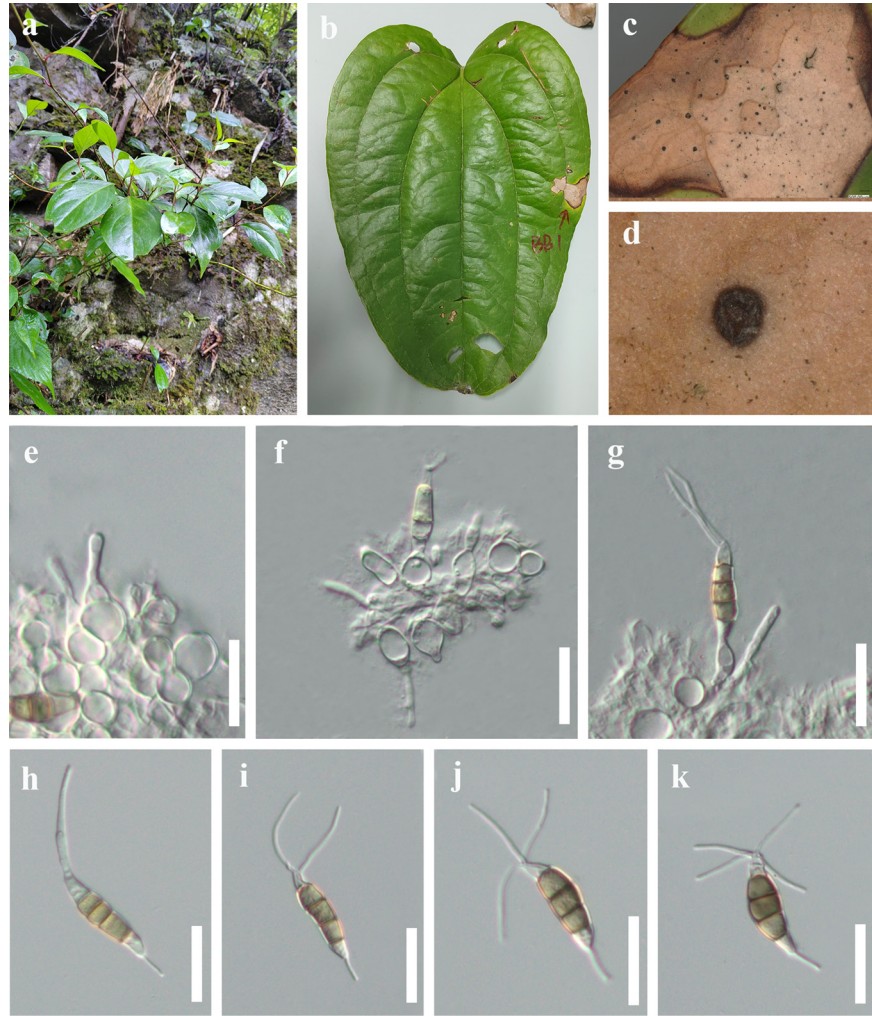

**FIG 7** *Neopestalotiopsis* sp.1 (HKAS 124560). (a) Host. (b) Leaf spot on *Smilax scobinicaulis*. (c and d) Close-up view of conidiomata. (e to g) Conidia attached to conidiogenous cells. (h to k) Conidia; scale bars, 20 $\mu$m (e to k).

conidiogenous cells. Conidiogenous cells indistinct. Conidia 19 to 25 × 6 to 8 $\mu$m ($\overline{x}$ = 22 × 7 $\mu$m, $n$ = 30), L/W ratio of 3.1, fusoid, ellipsoid to subcylindrical, straight to slightly curved, 4 septate; basal cell conic to obconic with a truncate base, hyaline to subhyaline, 3 to 6 $\mu$m long; three median cells 13 to 15 $\mu$m long ($\overline{x}$ = 14 $\mu$m, $n$ = 30), wall rugose, versicolorous, septa darker than the rest of the cell; second cell from base pale brown to brown, 3 to 5 $\mu$m long; third cell brown, 3 to 6 $\mu$m long; fourth cell brown, 3 to 6 $\mu$m long; apical cell 2 to 4 $\mu$m long, hyaline, rugose and thin walled; with 2 (seldom 3) tubular apical appendages, arising from the apical crest, unbranched, filiform, 11 and 20 $\mu$m long; single basal appendage 2 to 5 $\mu$m long, unbranched, tubular, centric.

Culture characteristics: colonies on PDA reached up to 8 cm in 2 weeks, with dense aerial mycelium on the surface with undulate edge; white. Fruiting bodies were observed after 14 days.

Material examined: Thailand, Chiang Rai Province, dead leaves of *Ceiba pentandra* (*Bombacaceae*), 16 Jan 2021, Y.R. Sun, CR20 (MFLU 22-0170); living culture, MFLUCC 22-0131; China, Guizhou Province, Guiyang City, Nanming District, Guiyang Medicinal Botanical Garden, on healthy leaves of *Pinellia ternata* (*Araceae*), 1 May 2022, Y.R. Sun, E2, living culture KUNCC 22-12596 = GUCC 21-0805.

Notes: our isolates KUNCC 22-12596 and MFLUCC 22-0170 clustered together with

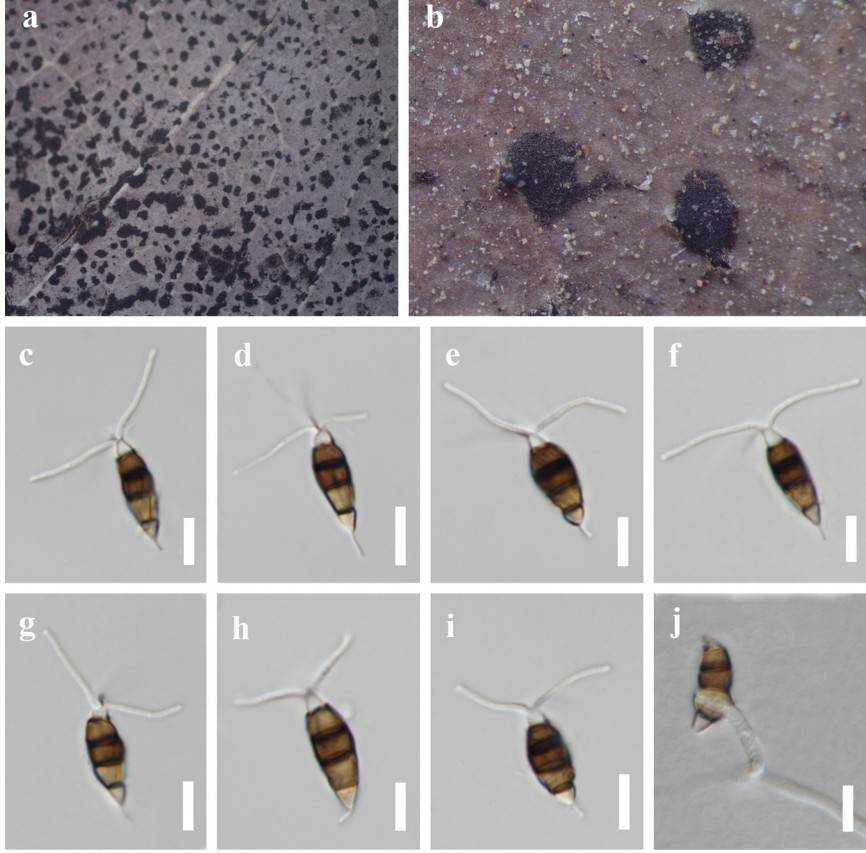

**FIG 8** *Neopestalotiopsis* sp. 2 (MFLU 22-0170). (a and b) Conidiomata on the host. (c to i) Conidia. (j) Germinated conidium; scale bars, 10 $\mu$m (c to j).

*Neopestalotiopsis* sp. 2 (CFCC 54340 and ZX22B), and these four isolates formed a distinct clade in the phylogenetic tree (Fig. 1). Four isolates have similar characteristics. We keep these four strains as *Neopestalotiopsis* sp. 2 as explained above. Interestingly, KUNCC 22-12596, MFLUCC 22-0170, ZX22B, and CFCC 54340 have different habitats. KUNCC 22-12596 was endophytic in healthy leaves of *Pinellia ternata*, MFLUCC 22-0170 was saprobic on decaying leaves of *Ceiba pentandra*, and CFCC 54340 and ZX22B were isolated from leaf spots of *Castanea mollissima*.

(g) *Neopestalotiopsis formicarum* Maharachch., K.D. Hyde, and Crous. Material examined: China, Guizhou Province, Guiyang City, Nanming District, Xiaochehe Road, Guiyang Ahahu National Wetland Park, on leaf spots of *Photinia serrulate* (*Rosaceae*), 21 September 2019, Y.R. Sun, AH11, living culture, GUCC 21-0809.

Notes: *Neopestalotiopsis formicarum* was introduced by Maharachchikumbura et al. (13) as a saprobic species isolated from dead ants in Ghana and plant debris from Cuba. Later, many studies have proven that *N. formicarum* is a serious phytopathogen, which can cause leaf fall disease in rubber trees in Thailand (46), leaf spot pathogens of the guarana plant in Brazil (47), and leaf brown blight of jabuticaba in Taiwan province, China (48). In this study, a new *N. formicarum* taxon was isolated from leaf spots of *Photinia serrulate* in China.

(h) *Neopestalotiopsis rhapidis* Qi Yang and Yong Wang bis. Material examined: China, Guizhou Province, Qiannan Bouyei and Miao Autonomous Prefecture, Libo District, leaf spots of *Podocarpus macrophyllus* (*Podocarpaceae*), 12 March 2022, Y.R. Sun, ML3 (HKAS 124559); living culture, KUNCC 22-12590 = GUCC 21-0806.

Notes: *Neopestalotiopsis rhapidis* was introduced by Yang et al. (44) from leaf spot of *Rhapis excelsa* (*Arecaceae*) in China. Our isolate KUNCC 22-12590 clustered together with *N. rhapidis* (GUCC 21501) in the phylogenetic tree. These two species have

overlapping conidial measurements (17 to 25 × 5 to 8 $\mu$m for KUNCC 22 to 12590 versus (22 to) 25.5 × 4 (to 6) $\mu$m for GUCC 21501) (44). Both isolates were associated with leaf spots in China. Therefore, we identify KUNCC 22-12590 and GUCC 21501 to be conspecific species, and KUNCC 22-12590 represents a new host record.

*(i) Neopestalotiopsis rhododendri* Qi Yang and Yong Wang bis. Material examined: Thailand, Chiang Mai Province, Mae Taeng District, Mushroom Research Center, leaf spots of *Dracaena fragrans* (*Liliaceae*), 15 September 2020, S Wang, LD1, living culture, MFLUCC 22-0130.

Notes: *Neopestalotiopsis rhododendri* was introduced by Yang et al. (44) from the diseased leaf of *Rhododendron simsii* (*Ericaceae*) in China. Based on our phylogenetic analysis of combined *ITS*, *tef1-α*, and *tub2* sequence data, our isolate MFLUCC 22-0130 clustered with the type species *N. rhododendri* (GUCC 21504) with good support (ML-BS = 99% and BYPP = 0.99). Our collection also shares similar morphological features with the holotype of *N. rhododendri* (GUCC 21504). Both isolates were associated with leaf spots. Therefore, we identify our collection as *N. rhododendri*, which represents a new host and geographical record.

**Pestalotiopsis Steyaert, Bull. Jard. bot. État Brux. 19:300 (1949). (i) Phylogenetic analyses.** The phylogenetic tree (*Pestalotiopsis*) comprised 120 ingroups and two outgroups, *Neopestalotiopsis protearum* (CBS 114178), and *N. cubana* (CBS 600.96). A total of 1,496 characters including gaps (543 for *ITS*, 516 for *tef1-α*, and 437 for *tub2*) were included in the phylogenetic analysis. Similar tree topologies were obtained by ML and BYPP methods, and the most likely tree ($-$ln = 12,403.616855) is presented (Fig. 9). The phylogenetic tree analyzed 10 *Pestalotiopsis* taxa isolated from medicinal plants and revealed three novel species and three new records of *Pestalotiopsis*.

**(ii) Genealogical concordance phylogenetic species recognition analysis.** The PHI test revealed that there is no significant recombination (Φw = 0.26) between *P. chiangmaiensis* and its closely related taxa *P. smilacicola* (MFLUCC 22-0125), *P. dracontomelon* (MFLUCC 10-0149), and *P. rhizophorae* (MFLUCC 17-0416) (Fig. 10a). The *P. loeiana* (MFLUCC 22-0123)-based PHI test confirmed that there is no significant recombination (Φw = 0.13) between *P. loeiana* and its closely related taxa *P. chiangmaiensis* (MFLUCC 22-0127), *P. nanningensis* (CSUFTCC10), *P. rhizophorae* (MFLUCC 17-0416), and *P. thailandica* (MFLUCC 17-1616) (Fig. 10b).

**(iii) Taxonomy.** *(a) Pestalotiopsis chiangmaiensis* Y.R. Sun and Yong Wang bis, sp. nov. Fungal names number: FN 571225; Facesoffungi number: FoF 04525 (Fig. 11).

Etymology: refers to the location where the fungus was encountered.

Holotype: MFLU 22-0164.

Associated with leaf strips of *Phyllostachys edulis*. Sexual morph: not observed. Asexual morph: conidiomata on PDA pycnidial, subglobose to globose, solitary or aggregated, dark, semi-immersed or partly erumpent; exuding black conidial masses. Conidiophores hyaline, smooth, simple, reduced to conidiogenous cells. Conidiogenous cells 5 to 11 × 1 to 3 $\mu$m, cylindrical to subcylindrical or ampulliform to lageniform, hyaline, smooth. Conidia pale brown, fusiform, straight to slightly curved, (3 to) 4 septate, 16 to 26 × 4 to 7 $\mu$m ($\overline{x}$ = 21 × 5 $\mu$m, $n$ = 40), L/W ratio of 4.2; basal cell obconic with a truncate base, hyaline or sometimes pale brown, smooth walled, 2 to 6 $\mu$m long; three median cells 10 to 16 $\mu$m long ($\overline{x}$ = 14 $\mu$m, $n$ = 40), pale brown, concolorous, wall rugose, septa darker than the rest of the cell, somewhat constricted at the septa; second cell from base pale brown, 3 to 6 $\mu$m long; third cell brown, 3 to 6 $\mu$m long; fourth cell brown, 3 to 6 $\mu$m long; apical cell 2 to 5 $\mu$m long, hyaline, conic to acute; with 2 (to 3) tubular appendages on the apical cell, inserted at different loci in a crest at the apex of the apical cell, unbranched, 8 to 13 $\mu$m long; single basal appendage 2 to 7 $\mu$m, unbranched, tubular, centric.

Culture characteristics: colonies on PDA reached 5 to 6 cm in diameter after 7 days at 25°C, colonies were filamentous to circular, medium dense, aerial mycelium on surface flat or raised, with filiform (curled) margin, fluffy, white from above and below; fruiting bodies black.

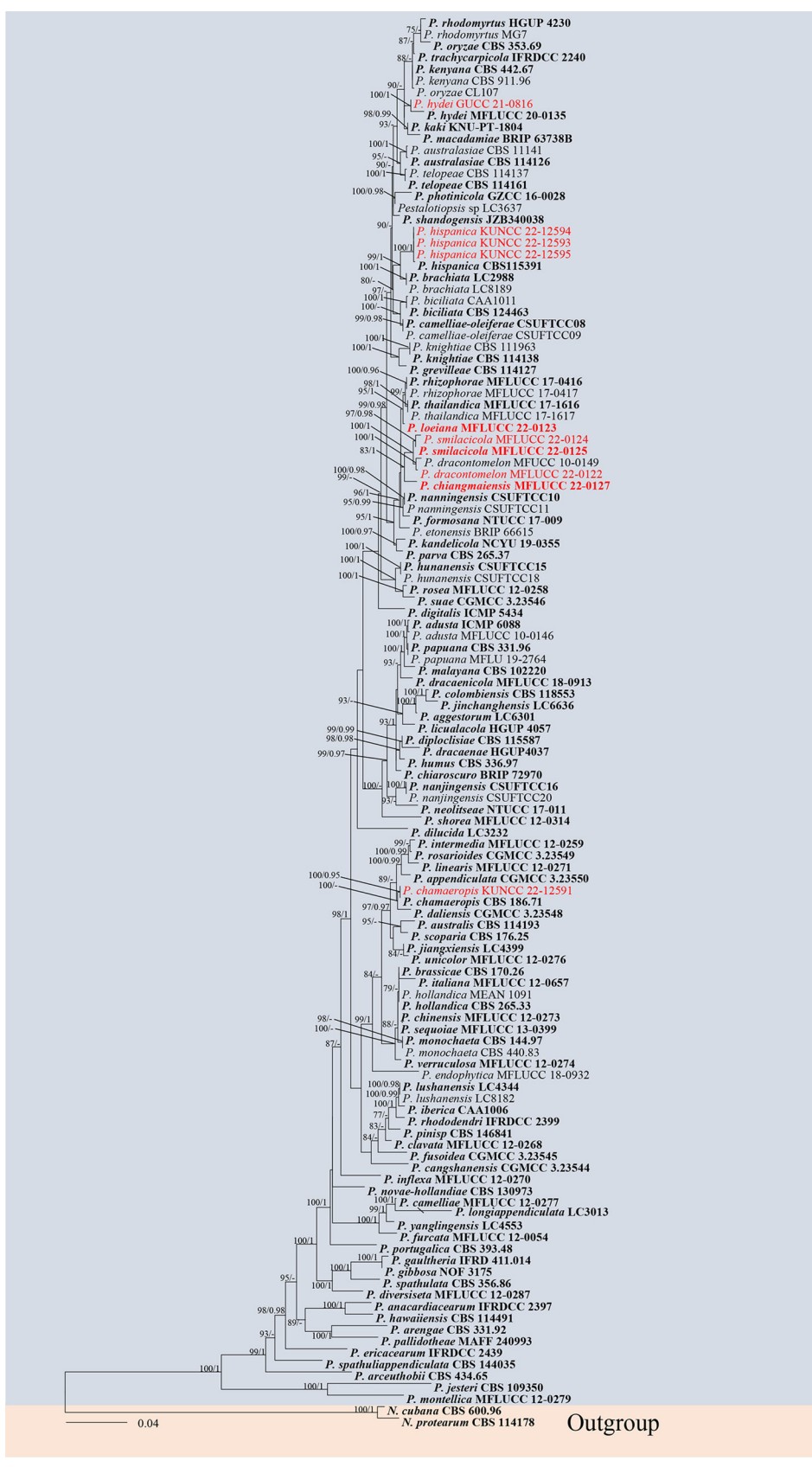

**FIG 9** Maximum likelihood (RAxML) tree for *Pestalotiopsis* based on analysis of a combined data set of *ITS*, *tef1-α*, and *tub2* sequence data. The tree is rooted with *Neopestalotiopsis protearum* (CBS 114178) and *N. cubana* (CBS 600.96). Bootstrap support values for ML greater than 75% and Bayesian posterior probabilities greater than 0.95 are given near nodes, respectively. The new isolates are in red, and the ex-type strains are in bold.

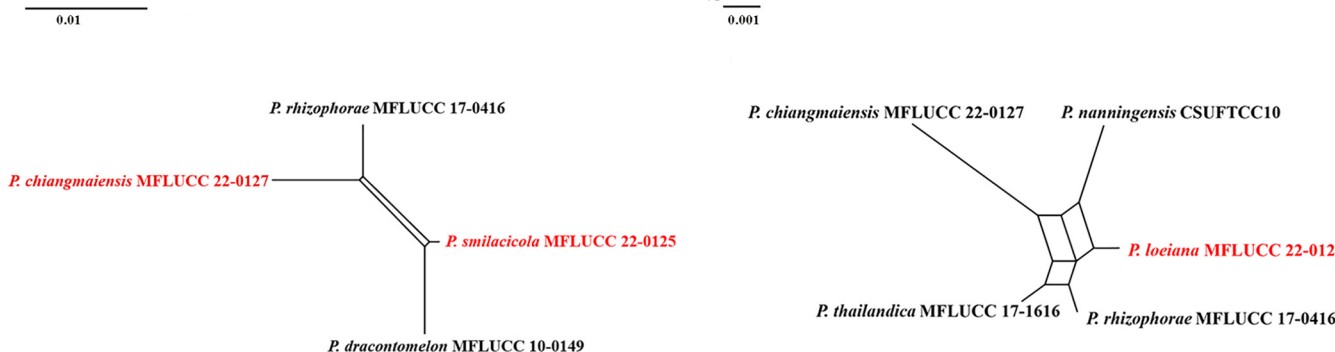

**FIG 10** (a and b) Split graphs showing the results of the PHI test of new *Pestalotiopsis* species with their most closely related species using Log-Det transformation and split decomposition options. The new taxon in each graph is shown in red font.

Material examined: Thailand, Chiang Mai Province, Mae Taeng District, Mushroom Research Center, leaf strip of *Phyllostachys edulis* (*Poaceae*), 15 July 2020, Y.R. Sun, M18 (MFLU 22-0164, holotype); ex-type culture, MFLUCC 22-0127.

Notes: *Pestalotiopsis chiangmaiensis* formed a distinct lineage and was sister to *P. smilacicola* and *P. dracontomelon* in the phylogenetic tree (Fig. 11). It differs by longer conidiogenous cells (5 to 11 $\mu$m versus 1 to 4 $\mu$m) than *P. smilacicola* and shorter apical appendages (8 to 13 $\mu$m versus 10 to 22 $\mu$m) than *P. dracontomelon*. In addition, there are 14 bp different (without gap, 474 bp) in the *tef1-α* region between *P. chiangmaiensis* (MFLUCC 22-0127) and *P. smilacicola* (MFLUCC 22-0125) and 15 bp different (without gap, 464 bp) between *P. chiangmaiensis* (MFLUCC 22-0127) and *P. dracontomelon* (MFLUCC 10-0149). The PHI test on *P. chiangmaiensis* also showed that there is no significant recombination ($\Phi$w = 0.26) between *P. chiangmaiensis* and its closely related taxa (Fig. 10a). Therefore, we introduce *P. chiangmaiensis* as a new species.

*(b) Pestalotiopsis loeiana* Y.R. Sun and Yong Wang bis, sp. nov. Fungal names number: FN 571226; Facesoffungi number: FoF 12919 (Fig. 12).

Etymology: refers to the collected site, Loei Province.

Holotype: MFLU 22-0167.

Saprobic on dead leaves. Sexual morph: not observed. Asexual morph: conidiomata solitary, black, semi-immersed on leaves. Conidiophores indistinct and conidiogenous cells indistinct. Conidia 17 and 22 × 4 and 6 $\mu$m ($\overline{x}$ = 19 × 5 $\mu$m, n = 40), L/W ratio of 3.7, fusiform, straight to slightly curved, 4 septate; basal cell obconic with a truncate base, hyaline or sometimes pale brown, rugose walled, 3 to 6 $\mu$m long, with 1 to 3 basal appendages, unbranched, tubular, centric, 3 to 13 $\mu$m long ($\overline{x}$ = 9 $\mu$m); three median cells 10 to 14 $\mu$m ($\overline{x}$ = 12, n = 40), doliiform to cylindrical, brown, concolorous, wall rugose, septa darker than the rest of the cell, somewhat constricted at the septa; second cell from base brown, 3 to 6 $\mu$m long; third cell brown, 3 to 5 $\mu$m long; fourth cell brown, 2 to 5 $\mu$m long; apical cell 3 to 5 $\mu$m long, hyaline, conic to acute; with 1 to 3 tubular appendages on the apical cell, inserted at different loci in a crest at the apex of the apical cell, unbranched, 13 to 24 $\mu$m long.

Culture characteristics: colonies on PDA reached 8 cm in diameter after 2 weeks at 25°C, colonies filamentous to circular, medium dense, mycelium on surface flat or raised, with filiform margin, fluffy, yellow circle in the middle surrounded by white mycelium from above, light yellow to pale brown from the reverse.

Material examined: Thailand, Loei Province, dead leaves of an identified plant, 27 February 2020, J.Y. Zhang, JY1 (MFLU 22-0167, holotype); ex-type culture, MFLUCC 22-0123.

Notes: *Pestalotiopsis loeiana* (MFLUCC 22-0123) is phylogenetically sister to *P. rhizophorae* and *P. thailandica*, which were isolated from leaf spots of mangroves (Fig. 11). Morphologically, *P. loeiana* is distinguishable by its more than one basal appendage

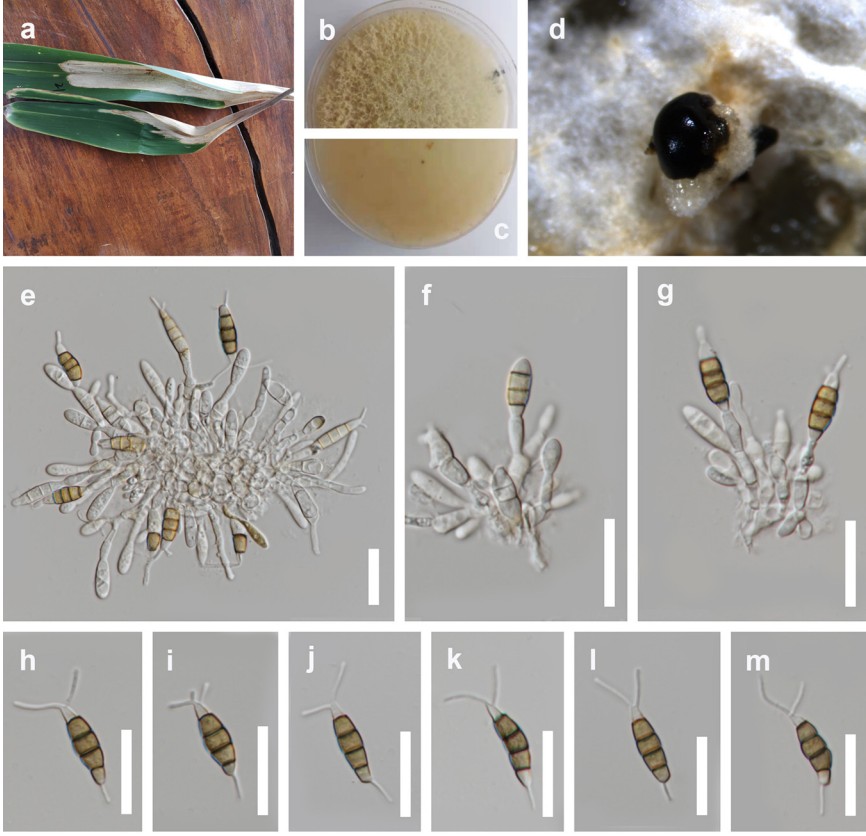

**FIG 11** *Pestalotiopsis chiangmaiensis* (MFLU 22-0164, holotype). (a) Host. (b and c) Cultures. (d) Colonies on PDA. (e to g) Conidiogenous cells and developing conidia. (h to m) Conidia; scale bars, 20 $\mu$m (e to m).

and smaller conidia (17 to 22 $\times$ 4 to 6 $\mu$m in *P. loeiana* versus 21 to 27 $\times$ 6. 5 to 7. 5 $\mu$m in *P. rhizophorae* versus 21 to 25 $\times$ 6 to 7 in *P. thailandica*). The result of the PHI test ($\Phi$w = 0.13) also showed that there is no significant recombination between *P. loeiana* and its closely related taxa (Fig. 10b). Therefore, we introduce *P. loeiana* as a new species.

(c) *Pestalotiopsis smilacicola* Y.R. Sun and Yong Wang bis, sp. nov. Fungal names number: FN 571227; Facesoffungi number: FoF 12921 (Fig. 13).

Etymology: refers to the host plant from which the fungus was isolated.

Holotype: MFLU 22-0165.

Associated with leaf spots of *Smilax* sp. Symptoms subcircular to irregular shape, brown, slightly sunken spots appear on the leaves of *Smilax china*, which later expand outward. Small auburn spots appeared initially and then gradually enlarged. Sexual morph: not observed. Asexual morph: conidiomata solitary, subglobose, unilocular, black, semi-immersed on leaves. Conidiomatal wall 7 to 10 $\mu$m wide, thin walled, pale brown. Conidiophores indistinct. Conidiogenous cells 1 to 4 $\times$ 1 to 3 $\mu$m, subcylindrical to ampulliform, hyaline, smooth. Conidia 18 to 22 $\times$ 4 to 7 $\mu$m ($\bar{x}$ = 20 $\times$ 5 $\mu$m, n = 40), L/W ratio of 3.8, fusiform, straight to slightly curved, 4 septate; basal cell obconic with a truncate base, hyaline or sometimes pale brown, smooth walled, 3 to 5 $\mu$m long; three median cells 9 to 15 $\mu$m long ($\bar{x}$ = 12 $\mu$m, n = 40), pale brown to brown, concolorous, wall rugose, septa darker than the rest of the cell, somewhat constricted at the septa; second cell from base pale brown to brown, 3 to 5 $\mu$m long; third cell brown, 2 to 5 $\mu$m long; fourth cell brown, 3 to 5 $\mu$m long; apical cell 2 to 5 $\mu$m long, hyaline, conic to acute; with 2 to 3 tubular appendages on the apical cell, inserted at different

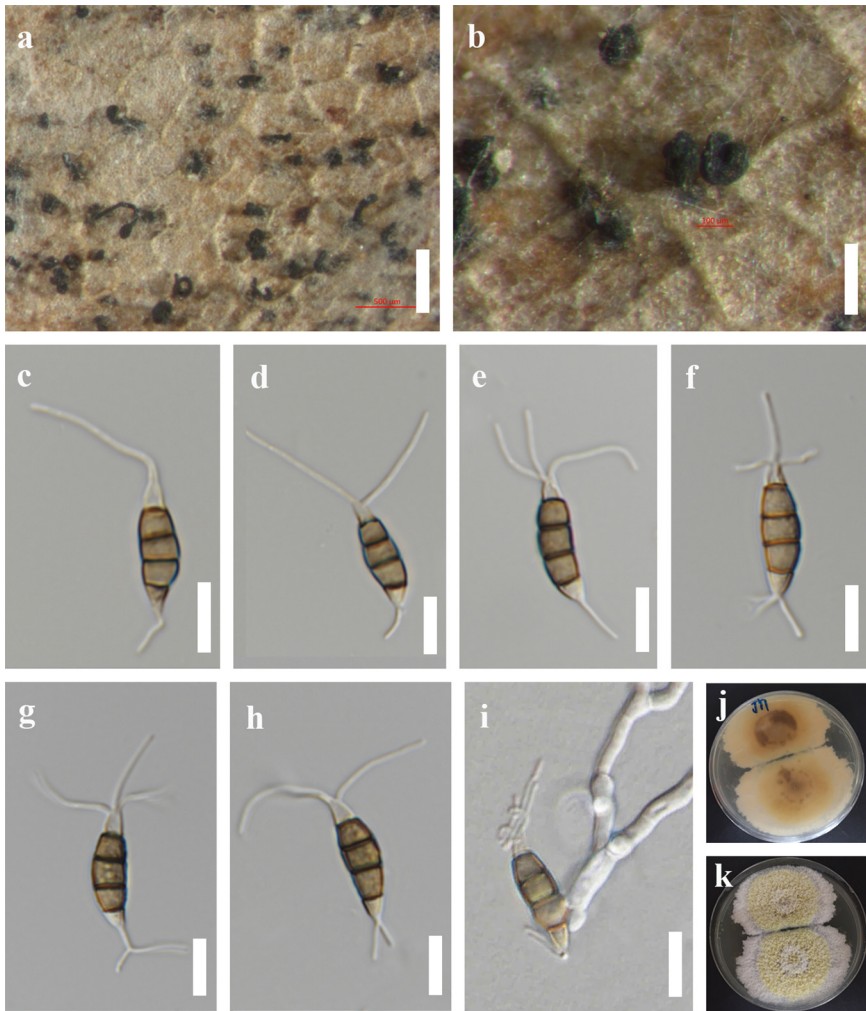

**FIG 12** *Pestalotiopsis loeiana* (MFLU 22-0167, holotype). (a and b) Conidiomata on the host. (c to h) Conidia. (i) Germinated conidium. (j and k) Colonies on PDA; scale bars, 500 $\mu$m (a), 200 $\mu$m (b), and 10 $\mu$m (c to i).

loci in a crest at the apex of the apical cell, unbranched, 6 to 14 $\mu$m long; single basal appendage, unbranched, tubular, centric, 2 to 6 $\mu$m long.

Culture characteristics: colonies on PDA reached 10 cm in diameter after 2 weeks at 25°C, colonies filamentous to circular, medium dense, aerial mycelium on surface flat or raised, with filiform margin, fluffy, white from above and reverse.

Material examined: Thailand, Chiang Mai Province, Mae Taeng District, Mushroom Research Center, leaf spots of *Smilax* sp. (*Liliaceae*), 16 July 2020, Y.R. Sun, M26 (MFLU 22-0165, holotype), ex-type culture, MFLUCC 22-0125. ibid., leaf spots of *Smilax china* (*Liliaceae*), 15 July 2020, Y.R. Sun, M13 living culture, MFLUCC 22-0124.

Notes: two isolates MFLUCC 22-0125 and MFLUCC 22-0124 share similar morphology. These two isolates clustered together and formed a sister clade to *P. dracontomelon* (MFLUCC 10-0149) in the phylogenetic tree. There is only 1 bp different in *tef1-α* and *tub2* genes and 3 bp different in the *ITS* gene between these two isolates. For the differences between *P. smilacicola* and its related species see the notes of *P. chiangmaiensis* (this study). Therefore, these two isolates are identified as conspecific, representing a new species.

*(d) Pestalotiopsis chamaeropis* Maharachch., K.D. Hyde, and Crous. Material examined: China, Guizhou Province, Guiyang City, Nanming District, Guiyang Medicinal Botanical Garden, on healthy leaves of *Peristrophe japonica* (*Acanthaceae*), 1 May 2022, Y.R. Sun, E33, living culture KUNCC 22-12591 = GUCC 21-0800.

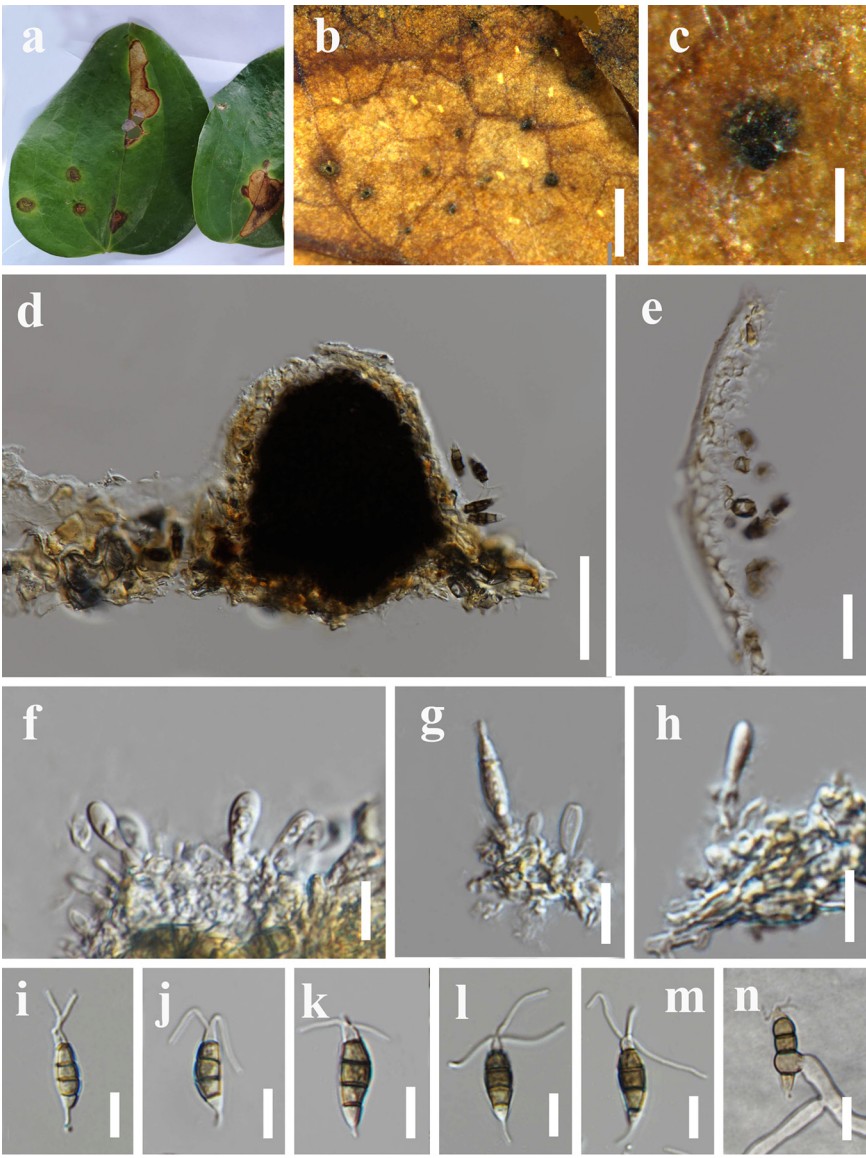

**FIG 13** *Pestalotiopsis smilacicola* (MFLU 22-0165, holotype). (a) Host. (b and c) Close-up view of conidiomata. (d) Section through conidioma. (e) Section through pycnidial wall. (f to h) Immature conidia attached to conidiogenous cells. (i to m) Conidia. (n) Germinated conidium; scale bars, 1,000 $\mu$m (b), 200 $\mu$m (c), 50 $\mu$m (d), 20 $\mu$m (e), 10 $\mu$m (f to n).

Notes: *Pestalotiopsis chamaeropis* was originally reported on leaves of *Chamaerops humilis* in Italy by Maharachchikumbura et al. (13). Subsequently, many studies have proven that *P. chamaeropis* is a serious phytopathogen, which can cause diseases of *Camellia sinensis* and *Camellia oleifera* and *Eurya nitida* (in China), *Erica arborea* (in Tunisia), Japanese andromeda (in Japan), and *Prostanthera rotundifolia* (in Australia) (34, 49–56). Park et al. (57) reported *P. chamaeropis* as an endophyte from the leaves of woody plants in Korea. In this study, our strain KUNCC 22-12591 is phylogenetically clustered with *P. chamaeropis* CBS 186.71 with good support (ML-BS = 100% and BYPP = 0.95), and it has overlapping characteristics with *P. chamaeropis* (CBS 186.71). Thus, we identify KUNCC 22-12591 as *P. chamaeropis*, representing a new host record.

*(e) Pestalotiopsis dracontomelon* Maharachch and K.D. Hyde. Material examined: Thailand, Chiang Rai Province, Mae Fah Luang University, leaf spots of *Podocarpus* sp. (*Podocarpaceae*), 15 January 2019, Y.R. Sun, S18 (MFLU 22-0166); living culture, MFLUCC 22-0122.

Notes: *Pestalotiopsis dracontomelon* was isolated from diseased leaves of *Dracontomelon mangifera* (*Anacardiaceae*) in Thailand (58). Our isolate MFLUCC 22-0122 was grouped with *P. dracontomelon* (MFLUCC 10-0149) in the phylogenetic tree. Morphologically, they have overlapping conidial measurements (19 to 26 × 5.5 to 8 $\mu$m for MFLUCC 22-0122 versus 18 to 23 × 5.5 to 7.5 $\mu$m for MFLUCC 10-0149). Therefore, we identify MFLUCC 22-0122 as the new host record of *P. dracontomelon*.

*(f) Pestalotiopsis hispanica* F. Liu, L. Cai, and Crous. Material examined: China, Guizhou Province, Guiyang City, Nanming District, Guiyang Medicinal Botanical Garden, on healthy leaves of *Peristrophe japonica* (*Acanthaceae*), 1 May 2022, Y.R. Sun, E53, living culture KUNCC 22-12595 = GUCC 21-0803; ibid., on healthy leaves of *Peristrophe japonica*, 1 May 2022, Y.R. Sun, E55, living culture KUNCC 22-12593 = GUCC 21-0802; ibid., on healthy leaves of *Peristrophe japonica*, 1 May 2022, Y.R. Sun, E52, living culture KUNCC 22-12594 = GUCC 21-0804.

Notes: *Pestalotiopsis hispanica* was originally reported on *Proteaceae* plants in Spain by Liu et al. (59). In this study, three strains were obtained from healthy leaves of *Peristrophe japonica* in China. These three strains grouped with *P. hispanica* (CBS 115391) with maximum support (ML-BS = 100% and BYPP = 1), and it has overlapping characteristics with *P. hispanica* (CBS 115391). Thus, we identify KUNCC 22-12594, KUNCC 22-12595, and KUNCC 22-12593 as *P. hispanica*, representing a new record.

*(g) Pestalotiopsis hydei* Huanraluek and Jayaward. Material examined: China, Guizhou Province, Qiannan Bouyei and Miao Autonomous Prefecture, Libo District, on dead twigs, 12 March 2022, J.E. Sun, L19-1, living culture, GUCC 21-0819.

Notes: *Pestalotiopsis hydei* was isolated from the leaf spots of *Litsea petiolate* in Thailand (60). Our isolate GUCC 21-0819 clustered together with *P. hydei* (MFLUCC 20-0135) in the phylogenetic tree. GUCC 21-0819 also has a similar conidial measurement to MFLUCC 20-0135 (19 to 26 × 4 to 6 $\mu$m in GUCC 21-0819, 18 to 35 × 3 to 6 $\mu$m in MFLUCC 20-0135). Additionally, there are only 3 bp different in the *ITS* and *tef1-α* genes. Therefore, we identify GUCC 21-0819 as a new geographical record of *P. hydei*.

***Pseudopestalotiopsis* Maharachch., K.D. Hyde, and Crous, Stud. Mycol. 79: 180 (2014). (i) Phylogenetic analyses.** The sequence data sets for *ITS*, *tef1-α*, and *tub2* were analyzed in combination to infer the interspecific relationships within *Pseudopestalotiopsis*. The aligned sequence matrix consisted of 27 sequences, including two outgroups *Pestalotiopsis trachycarpicola* (IFRDCC 2240) and *P. linearis* (MFLUCC 12-0271). Similar tree topologies were obtained by ML and BYPP methods, and the most likely tree (−ln = 66,531.894) is presented (Fig. 14). Our collection is clustered with the type species of *Pseudopestalotiopsis*, *P. theae*, in the phylogenetic tree (Fig. 14).

**(ii) Taxonomy.** *(a) Pseudopestalotiopsis theae* (Sawada) Maharachch., K.D. Hyde, and Crous. Material examined: Thailand, Suphan Buri Province, dead leaf of *Ceriops tagal* (*Rhizophoraceae*), 5 September 2020, S Wang, TN07 (MFLU 22-0169); living culture, MFLUCC 22-0128.

Notes: our isolate MFLUCC 22-0128 is phylogenetically grouped with the type species of *Pseudopestalotiopsis*, *Ps. theae*. Morphologically, our new collection MFLU 22-0169 resembles *Ps. theae* (MFLUCC 12-0055) in color and size of the conidiogenous cells, conidia, and appendages. Therefore, we report this isolate as a new host record of *Ps. theae* from *Ceriops tagal*.

## DISCUSSION

During research of microfungi on medicinal plants in southwest China and Thailand, 26 pestalotioid strains representing 17 species were isolated from 16 medicinal plants. Four new *Neopestalotiopsis* species, namely, *N. amomi*, *N. photiniae*, *N. suphanburiensis*, and *N. hyperici*, three new *Pestalotiopsis* species, namely, *P. chiangmaiensis*, *P. loeiana*, and *P. smilacicola*, and eight new records are introduced. Among them, 10 species are related to leaf diseases of medicinal plants, 3 species are saprobes, 2 species are endophytes, and 1 species has various lifestyles. A worldwide checklist of pestalotioid species

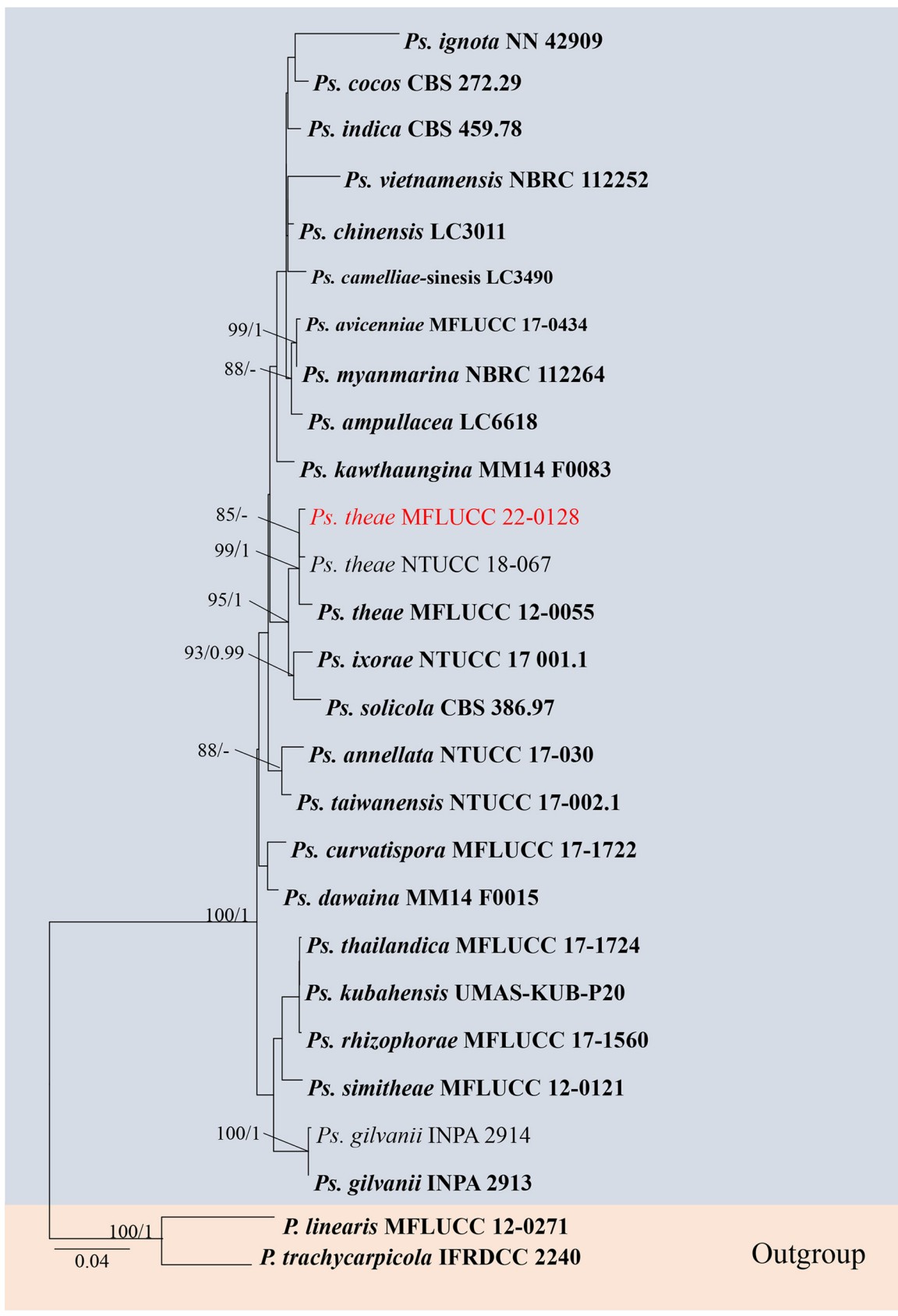

**FIG 14** Maximum likelihood (RAxML) tree for *Pseudopestalotiopsis* based on analysis of a combined data set of *ITS*, *tef1-α*, and *tub2* and sequence data. The tree is rooted with *P. trachycarpicola* (IFRDCC 2240) and *P. linearis* (MFLUCC 12-0271). Bootstrap support values for ML greater than 75% and Bayesian posterior probabilities greater than 0.95 are given near nodes, respectively. The new isolates are in red, and the ex-type strains are in bold.

associated with medicinal plants is provided (Table 1). Among them, most are related to diseases of medicinal plants, and 47 (a total of 79) were found in China and Thailand. The analysis showed that the majority of pestalotioid species are associated with dicotyledonous medicinal plants (Fig. 15).

In this study, *Neopestalotiopsis* sp. 1 was isolated from healthy leaves of *Pinellia ternata* and dead leaves of *Ceiba pentandra*, and it was previously isolated from the leaf spots of *Castanea mollissima* (45). *Pestalotiopsis chamaeropis* was found as an endophyte from an important medicinal plant here. However, it was reported as a serious phytopathogen in different countries (34, 49–51, 53–56). *Pestalotiopsis photiniae* was reported as an endophytic fungus on the branches of *Roystonea regia*, and it also has been isolated from the diseased leaves of blueberries (61, 62). These seem to indicate that one fungus can be endophytic, pathogenic, or saprophytic in different (or the same) plants or organs. Understanding the external factors that influence the fungal lifestyle could have major implications for agriculture, industrial development, and human health.

It is worth noting that two *Neopestalotiopsis* species, *N. amomi* and *N. hyperici* introduced in this paper, do not match the characteristics of versicolorous median cells described in *Neopestalotiopsis* (13). This phenomenon was also mentioned by Liu et al. (27). The reason for this phenomenon probably could be the type of medium or the time of cultivation. However, there are not enough data to explain this phenomenon.

*Neopestalotiopsis* and *Pseudopestalotiopsis* were separated from *Pestalotiopsis* by Maharachchikumbura et al. (13) based on phylogenetic analyses and conidial color. After that, the number of species in *Neopestalotiopsis* increased from 24 to 73, and the number of species in *Pseudopestalotiopsis* increased from 3 to 23 (http://www.indexfungorum.org/, 10 Nov 2022). In recent years, there have been more reports on diseases caused by *Neopestalotiopsis*; for example, *N. vitis* caused grapevine leaf spots in China, *N. rosicola* caused stem canker of *Rosa chinensis* in China, *N. clavispora* caused leaf spots and fruit rot of strawberry in India, *N. maddoxii* caused flower diseases of *Macadamia integrifolia* in Australia, *N. eucalyptorum* was associated with disease of *Eucalyptus* plants in Portugal, and *N. siciliana* caused stem lesion and dieback on avocado plants in Italy (22, 23, 26, 29, 30, 63). However, only six *Pseudopestalotiopsis* species were reported as plant pathogens (47, 64–67). In this study, there are nine *Neopestalotiopsis* species, and only one *Pseudopestalotiopsis* species was encountered. This seems to indicate that *Neopestalotiopsis* has richer species diversity, and the *Neopestalotiopsis* species are more likely to infect the plant and cause disease than *Pseudopestalotiopsis*. Comparing the differences between the two genera through whole-genome sequence analysis and finding related disease-causing genes would probably explain this phenomenon.

The interspecific morphological differences of pestalotioid species have been unclear. In their attempt to find a reliable criterion for interspecific differences, taxonomists have complicated the description of pestalotioid species (11, 13, 60, 68). In the past, the conidia were divided into apical cells, basal cells, and the three median cells when describing them. The three median cells were divided into the second cells from the base, third cells, and fourth cells, and every cell is measured. Obviously, the length of the three median cells is the sum length of the second, third, and fourth cells. Therefore, the descriptions are repeated. In addition, the characteristics of each cell were not treated as criteria for interspecies differences (11, 26, 27, 69). Therefore, we suggest removing the measurement for each cell to simplify the descriptions.

## MATERIALS AND METHODS

**Collection, examination, and isolation.** Fresh healthy leaves, diseased leaves, and twigs of different medicinal plants were collected from terrestrial habitats in southwest China and Thailand from 2019 to 2022. Samples were brought to the laboratory in Ziploc plastic bags or paper envelopes for examination. The fruiting bodies on natural substrates were observed and photographed using a stereomicroscope

**TABLE 1** Checklist of pestalotioid species associated with medicinal plants

| Species[a] | Life mode[b] | Disease (if any) | Host[c] | Location | Reference |
|---|---|---|---|---|---|
| *Neopestalotiopsis acrostichi* | P | Leaf spot | *Acrostichum aureum* (F) | Thailand | 67 |
| *Neopestalotiopsis alpapicalis* | E | | *Rhyzophora mucronate* (D) | Thailand | 96 |
| *Neopestalotiopsis amomi* | P | Leaf spot | *Amomum villosum* (M) | China | This study |
| *Neopestalotiopsis brachiata* | P | Leaf spot | *Rhizophora apiculate* (D) | Thailand | 67 |
| *Neopestalotiopsis clavispora* | P | Leaf spot, branch blight | *Dendrobium officinale, Taxus×media* (M/G) | China | 97, 98 |
| *Neopestalotiopsis cubana* | P | Leaf blight | *Ixora chinensis* (D) | Malaysia | 99 |
| *Neopestalotiopsis dendrobii* | E | | *Dendrobium cariniferum* (M) | Thailand | 41 |
| *Neopestalotiopsis ellipsospor* | Unknown | | *Ardisia crenata* (D) | Hong Kong, China | 13 |
| *Neopestalotiopsis eucalypticola* | Unknown | | *Eucalyptus globulus* (D) | Unknown | 13 |
| *Neopestalotiopsis eucalyptorum* | P | Leaf necrosis, stem basal cankers | *Eucalyptus globulus* (D) | Fundão/Guarda/ Portugal | 22 |
| *Neopestalotiopsis formicarum* | P | Leaf spot | *Photinia serratifolia* | China | This study |
| *Neopestalotiopsis haikouensis* | P | Leaf spot | *Ilex chinensis* (D) | China | 17 |
| *Neopestalotiopsis hispanica* | P | Leaves and stem necrosis | *Eucalyptus globulus* (D) | Fundão/Guarda/Spain | 22 |
| *Neopestalotiopsis hydeana* | P | Leaf spot | *Alpinia malaccensis* (M) | Thailand | 60 |
| *Neopestalotiopsis hyperici* | P | Leaf spot | *Hypericum monogynum* (D) | China | This study |
| *Neopestalotiopsis iberica* | P | Leaves and stem necrosis | *Eucalyptus globulus* (D) | Pegões/Portugal/Spain | 22 |
| *Neopestalotiopsis longiappendiculata* | P | Leaves and stem necrosis | *Eucalyptus globulus/E. nitens* (D) | Furadouro/Portugal | 22 |
| *Neopestalotiopsis lusitanica* | P | Leaves and stem necrosis | *Eucalyptus globulus* (D) | Pegões/Portugal | 22 |
| *Neopestalotiopsis pernambucana* | Unknown | | *Vismia guianensis* (D) | Brazil | 100 |
| *Neopestalotiopsis petila* | P | Leaf spot | *Rhizophora mucronate* (D) | Thailand | 67 |
| *Neopestalotiopsis photiniae* | P | Leaf spot | *Photinia serrulate* (D) | China | This study |
| *Neopestalotiopsis rhapidis* | P | Leaf spot | *Podocarpus macrophyllus* (G) | China | This study |
| *Neopestalotiopsis rhizophorae* | P | Leaf spot | *Rhizophora mucronate* (D) | Thailand | 67 |
| *Neopestalotiopsis rhododendri* | P | Leaf spot | *Dracaena fragrans* (M) | Thailand | This study |
| *Neopestalotiopsis rosae* | Unknown | | *Paeonia suffruticosa* (D) | United States | 13 |
| *Neopestalotiopsis rosicola* | P* | Stem canker | *Rosa chinensis* (D) | China | 63 |
| *Neopestalotiopsis saprophytica* | S | | *Litsea rotundifolia* (D) | Hong Kong, China | 13 |
| *Neopestalotiopsis surinamensis* | E | | *Scurrula atropurpurea* (D) | Indonesia | 101, 102 |
| *Neopestalotiopsis thailandica* | P | Leaf spot | *Rhizophora mucronate* (D) | Thailand | 67 |
| *Pestalotiopsis adusta* | E, P* | Leaf spot | *Clerodendrum canescens/ Sinopodophyllum hexandrum/ Rubus idaeus* (D) | China | 103–105 |
| *Pestalotiopsis affinis* | P | Leaf spot | *Taxus chinensis* (G) | China | 106 |
| *Pestalotiopsis alpiniae* | P | Leaf spot | *Alpinia galanga* (M) | China | 106 |
| *Pestalotiopsis antiaris* | P | Leaf spot | *Antiaris toxicaria* (M) | China | 106 |
| *Pestalotiopsis bicilia* | S | | *Viburnum opulus* (D) | Canada | 13 |
| *Pestalotiopsis biciliata* | P* | Stem canker | *Pistacia lentiscus* (D) | Tunisia | 107 |
| *Pestalotiopsis bicolor* | S | | *Smilax sp.* (M) | United States | 108 |
| *Pestalotiopsis bruguierae* | Unknown | | *Bruguiera gymnorhiza* (D) | India | 109 |
| *Pestalotiopsis bulbophylli* | S | | *Bulbophyllum thouars* (M) | China | 110 |
| *Pestalotiopsis chamaeropis* | E, P* | Leaf spot | *Eurya nitida, Peristrophe japonica* (D) | China | 55, this study |
| *Pestalotiopsis chiangmaiensis* | P | Leaf strip | Bamboo (M) | Thailand | This study |
| *Pestalotiopsis cruenta* | Unknown | | *Polygonum lasianthum* (D) | Japan | Index Fungorum (2022) |
| *Pestalotiopsis digitalis* | P | Leaf spot | *Digitalis purpurea* (D) | New Zealand | 58 |
| *Pestalotiopsis dilleniae* | P | Leaf spot | *Dillenia turbinate* (M) | China | 106 |
| *Pestalotiopsis diploclisiae* | Unknown | | *Diploclisia glaucescens* (D) | Hong Kong, China | 13 |
| *Pestalotiopsis dracaenae* | S | | *Dracaena fragrans* (M) | China | 111 |
| *Pestalotiopsis ellipsospora* | P* | Stem canker | *Acanthopanax divaricatus* (D) | Korea | 112 |
| *Pestalotiopsis gibbosa* | S | | *Gaultheria shallon* (D) | United States | 113 |
| *Pestalotiopsis heucherae* | Unknown | | *Heuchera parviflora* | United States | 114 |
| *Pestalotiopsis hispanica* | E | | *Peristrophe japonica* | China | This study |
| *Pestalotiopsis hughesii* | Unknown | | *Cyperus articulates* (M) | Ghana | 115 |
| *Pestalotiopsis japonica* | Unknown | | *Cedrela sinensis* (D) | Japan | 19 |
| *Pestalotiopsis jinchanghensis* | E | | *Vaccinium dunalianum* (D) | China | 116 |

**TABLE 1** (Continued)

| Species[a] | Life mode[b] | Disease (if any) | Host[c] | Location | Reference |
|---|---|---|---|---|---|
| *Pestalotiopsis kenyana* | P* | Leaf spot | *Zanthoxylum schinifolium* (D) | China | 117 |
| *Pestalotiopsis kunmingensis* | E | | *Podocarpus macrophyllus* (G) | China | 68 |
| *Pestalotiopsis kwangsiensis* | P | Leaf spot | *Sinopimelodendron kuwangsiensis* (D) | China | 106 |
| *Pestalotiopsis lespedezae* | Unknown | | *Lespedeza bicolor* (D) | Japan | 118 |
| *Pestalotiopsis linearis* | E | | *Trachelospermum sp.* (D) | China | 24 |
| *Pestalotiopsis lushanensis* | P* | Brown leaf spot, leaf blight | *Sarcandra glabra, Podocarpus macrophyllus* (G) | China | 119 |
| *Pestalotiopsis microspora* | S | | *Hedera helix* (D) | Argentina | 120 |
| *Pestalotiopsis moluccensis* | Unknown | | *Xylocarpus moluccensis* (D) | India | 109 |
| *Pestalotiopsis neolitseae* | P* | Leaf spot | *Neolitsea villosa* (D) | Taiwan, China | 51 |
| *Pestalotiopsis oenotherae* | Unknown | | *Oenothera laciniata* (D) | United States | 121 |
| *Pestalotiopsis pandani* | Unknown | | *Pandanus odoratissimus* (M) | Taiwan, China | 108 |
| *Pestalotiopsis paraguariensis* | Unknown | | *Ilex paraguariensis* (D) | Brazil | 122 |
| *Pestalotiopsis pestalozzioides* | Unknown | | *Clematis ligusticifolia* (D) | New Mexico | 11 |
| *Pestalotiopsis pipericola* | Unknown | | *Piper nigrum* (D) | India | 123 |
| *Pestalotiopsis quadriciliata* | Unknown | | *Vitis vulpine* (D) | Canada | 124 |
| *Pestalotiopsis rhodomyrtus* | Unknown | | *Rhodomyrtus tomentosa* (D) | China | 125 |
| *Pestalotiopsis smilacicola* | P | Leaf spot | *Smilax china, Dioscorea sp.* (M) | Thailand | This study |
| *Pestalotiopsis sinensis* | Unknown | | *Ginkgo biloba* (G) | China | 126 |
| *Pestalotiopsis tecomicola* | Unknown | | *Tecoma radicans* (D) | United States | 11 |
| *Pestalotiopsis thailandica* | P | Leaf spot | *Rhizophora apiculate* (D) | Thailand | 67 |
| *Pseudopestalotiopsisis ampullace* | E | | *Magnolia candolli* (D) | China | 76 |
| *Pseudopestalotiopsis curvatispora* | P | Leaf spot | *Rhizophora mucronate* (D) | Thailand | 67 |
| *Pseudopestalotiopsis gilvanii* | P* | Leaf spot | *Paullinia cupana* (D) | Brazil | 47 |
| *Pseudopestalotiopsis indica* | Unknown | | *Hibiscus rosa-sinensis* (D) | India | 13 |
| *Pseudopestalotiopsis simitheae* | S, E | | *Pandanus odoratissimus/ Magnolia candolli (M/D)* | Thailand/China | 76, 127 |
| *Pseudopestalotiopsis thailandica* | P | Leaf spot | *Rhizophora mucronate* (D) | Thailand | 67 |
| *Pseudoestalotiopsis theae* | S | | *Ceriops tagal* (D) | Thailand | This study |

[a]The checklist includes species names, life modes, disease names (if any), hosts, locations, and references. The current name is used according to Index Fungorum (2022).
[b]The mode of life is given as endophyte (E), pathogen (P), and saprobe (S). For the species, those with confirmed pathogenicity data are marked with an asterisk (*).
[c]The taxonomic status of the host is given as dicotyledons (D), ferns (F), gymnosperms (G), and monocotyledons (M).

(SteREO Discovery, V12, Carl Zeiss Microscopy GmBH, Germany; VHX-7000, Keyence, Japan). Morphological characteristics were observed using a Nikon Eclipse Ni compound microscope (Nikon, Japan) and photographed with a Nikon DS-Ri2 digital camera (Nikon, Japan) or using a Carl Zeiss compound microscope (Carl Zeiss AG, Germany) and an Axiocam 208 color digital camera (Carl Zeiss AG, Germany). The photo plates were made with Adobe Photoshop CS6 Extended v. 13.0 software. Measurements were obtained with the Tarosoft (R) Image Frame Work software.

For endophytes, materials were washed under running tap water and immersed in 70% ethanol for 1 min, followed by soaking in 4% NaOCl for 1 min, rinsing three times in sterile distilled water, and drying on sterile filter paper. For the control, the final sterile water rinse was plated and observed during the postincubation period. The absence of any fungal (microbial) growth indicated that the leaf surface was sterile (70). The sterilized materials were cut into 2- to 5-mm$^2$ segments and placed on PDA containing 50 $\mu$g/mL penicillin and 50 $\mu$g/mL streptomycin (71). The plates were observed daily, and the mycelial growth on the edge of the fungal colonies was transferred to fresh PDA dishes to obtain pure cultures. For other samples, single-spore isolations were used to obtain pure cultures following the methods described by Senanayake et al. (72). Germinated conidia were transferred to fresh PDA plates and incubated at 25°C for 4 weeks. The pure cultures were deposited in Mae Fah Luang University Culture Collection (MFLUCC), Chiang Rai, Thailand, the Culture Collection of Kunming Institute of Botany, the Chinese Academy of Sciences (KUNCC), Kunming, China, and the Culture Collection of the Department of Plant Pathology, Agriculture College, Guizhou University (GUCC), Guiyang, China. Specimens were deposited in the herbarium of Mae Fah Luang University (MFLU) Chiang Rai, Thailand, and the Herbarium of Cryptogams, Kunming Institute of Botany, Academia Sinica (HKAS), Kunming, China. Facesoffungi (FoF) numbers were acquired as described by Jayasiri et al. (73). Taxonomic descriptions and nomenclature were deposited at Fungal Names (https://nmdc.cn/fungalnames/registe) following the description in reference 74.

**DNA extraction, PCR amplification, and sequencing.** A Biomiga fungus genomic DNA extraction kit (Biomiga, USA) was used to extract DNA from fresh fungal mycelia, which were grown on PDA medium for 4 weeks at 25°C. PrepMan ultra sample preparation reagent (Thermo Fisher Scientific, Japan) was used to extract DNA directly from fruiting bodies. Three genes were selected in this study: the

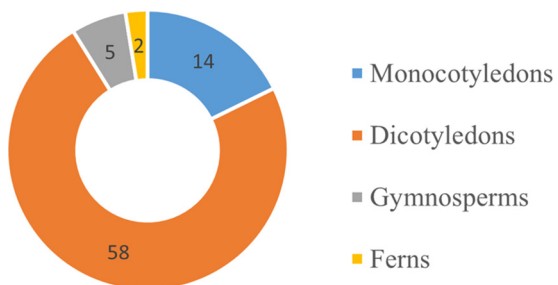

**FIG 15** Distribution of pestalotioid species in different medicinal plants.

internal transcribed spacers (*ITS*), the translation elongation factor 1 (*tef1-α*), and the partial *β*-tubulin region (*tub2*). PCR was carried out in a 20-$\mu$L reaction volume, which contained 10 $\mu$L of 2× PCR master mix, 7 $\mu$L of double-distilled water (ddH$_2$O), 1 $\mu$L of each primer, and 1 $\mu$L of template DNA. The PCR thermal cycle program and primers are given in Table 2. Purification and sequencing of PCR products were carried out at SinoGenoMax (Beijing) Co., China.

**Phylogenetic analyses.** Phylogenetic analyses and the preparatory work were carried out according to the method described in Dissanayake et al. (75). The sequences were compared against the NCBI GenBank nucleotide database using BLASTn to evaluate the closest phylogenetic matches. All sequences used in this study were downloaded from GenBank referring to previous studies (16, 23, 27, 30, 44, 65, 67, 76–79) (Table 3). The single-gene sequences were checked with BioEdit v. 7.0.9.0 (80). Alignments for each locus were generated using MAFFT v.7 (https://mafft.cbrc.jp/alignment/server/) and were manually improved using AliView (81). The final single-gene alignments were combined by SequenceMatrix 1.7.8 (82). For the final alignment, see Data S1 in the supplemental material.

The maximum likelihood (ML) analyses were carried out using IQ-TREE (83, 84) on the IQ-TREE web server (http://iqtree.cibiv.univie.ac.at, 1 Nov 2022) under partitioned models. The best-fit substitution models (Table 4) were determined by WIQ-TREE (85). Ultrafast bootstrap (BS) analyses were implemented with 1,000 replicates (86, 87).

The Bayesian posterior probability (BYPP) analyses were performed in CIPRES (88) with MrBayes on XSEDE 3.2.7a (89). MrModeltest 2.2 (90) was used to evaluate the best nucleotide substitution models (Table 4) for the data. The Markov chain Monte Carlo (MCMC) sampling approach was used to calculate posterior probabilities (PP) (91). Six simultaneous Markov chains were run for 10 million generations, and trees were sampled every 1,000th generation. The first 20% of trees, representing the burn-in phase of the analyses, were discarded, and the remaining trees were used for calculating posterior probabilities in the majority rule consensus tree.

The final phylogenetic trees were viewed with FigTree v1.4.0 (http://tree.bio.ed.ac.uk/software/figtree/) and were modified in Microsoft Office PowerPoint 2010.

**Genealogical concordance phylogenetic species recognition analysis.** Genealogical concordance phylogenetic species recognition was used to analyze the new species, their species boundaries, and their most closely related taxa, as described by Quaedvlieg et al. (92). The pairwise homoplasy index (PHI) test (93) was performed in SplitsTree4 (94, 95). If the PHI is above 0.05 threshold (Φw > 0.05), it indicates that there is no significant recombination present in the data set. The relationships between closely related species were visualized in splits graphs with both the Log-Det transformation and splits decomposition options.

**Data availability.** The sequences generated in this study can be found in GenBank. The accession numbers of the sequences deposited in GenBank are listed in Table 3.

## SUPPLEMENTAL MATERIAL

Supplemental material is available online only.
**SUPPLEMENTAL FILE 1**, PDF file, 0.5 MB.

**TABLE 2** Primers and PCR procedures used in this study

| Locus | Primers | | PCR procedures | Reference |
|---|---|---|---|---|
| | Name | Sequence (5′ to 3′) | | |
| *ITS* | ITS5 | GGAAGTAAAAGTCGTAACAAGG | 94°C 3 min; 94°C 30 s; 52°C 30 s; 72°C 1 min; repeat | 128 |
| | ITS4 | TCCTCCGCTTATTGATATGC | 2 to 4 for 35 cycles; 72°C 8 min; 4°C on hold | |
| *tef1-α* | EF1-728F | CAT CGA GAA GTT CGA GAA GG | 94°C 5 min; 94°C 30 s; 52°C 30 s; 72 °C 1 min; repeat | 129, 130 |
| | EF2 | GGA RGT ACC AGT SAT CAT GTT | 2 to 4 for 40 cycles; 72°C 8 min; 4°C on hold | |
| *tub2* | T1 | AACATGCGTGAGATTGTAAGT | 95°C 3 min; 94°C 30 s; 55°C 50 s; 72°C 1 min; repeat | 131, 132 |
| | Bt2b | ACCCTCAGTGTAGTGACCCTTGGC | 2 to 4 for 40 cycles; 72°C 8 min; 4°C on hold | |

**TABLE 3** Taxa of the three genera *Neopestalotiopsis*, *Pestalotiopsis*, and *Pseudopestalotiopsis* used in the phylogenetic analysis with the corresponding GenBank accession numbers

| Taxa[a] | Strain/voucher no. | tef1-α[b] | tub2[b] | ITS[b] |
|---|---|---|---|---|
| **Neopestalotiopsis acrostichi** | **MFLUCC 17-1754** | **MK764316** | **MK764338** | **MK764272** |
| *N. alpapicalis* | **MFLUCC 17-2544** | **MK463547** | **MK463545** | **MK357772** |
| <u>*N. amomi*</u> | <u>**HKAS 124563**</u> | **OP653489** | **OP752133** | **OP498012** |
| <u>*N. amomi*</u> | <u>**HKAS 124564**</u> | **OP753382** | **OP765913** | **OP498013** |
| *N. aotearoa* | **CBS 367.54** | **KM199526** | **KM199454** | **KM199369** |
| *N. asiatica* | **MFLUCC 12-0286** | **JX399049** | **JX399018** | **JX398983** |
| *N. australis* | **CBS 114159** | **KM199537** | **KM199432** | **KM199348** |
| *N. brachiata* | **MFLUCC 17-1555** | **MK764318** | **MK764340** | **MK764274** |
| *N. brasiliensis* | **COAD 2166** | **MG692402** | **MG692400** | **MG686469** |
| *N. camelliae-oleiferae* | **CSUFTCC81** | **OK507955** | **OK562360** | **OK493585** |
| *N. camelliae-oleiferae* | CSUFTCC82 | OK507956 | OK562361 | OK493586 |
| *N. cavernicola* | **KUMCC 20-0269** | **MW550735** | **MW557596** | **MW545802** |
| *N. cavernicola* | KUMCC 20-0332 | MW590327 | MW590328 | MW581238 |
| *N. chiangmaiensis* | **MFLUCC 18-0113** | **MH388404** | **MH412725** | **NA** |
| *N. chrysea* | **MFLUCC 12-0261** | **JX399051** | **JX399020** | **JX398985** |
| *N. clavispora* | **MFLUCC 12-0281** | **JX399045** | **JX399014** | **JX398979** |
| *N. cocoes* | **MFLUCC 15-0152** | **KX789689** | **NA** | **NR_156312** |
| *N. coffea-arabicae* | **HGUP4015** | **KF412644** | **KF412641** | **KF412647** |
| *N. coffea-arabicae* | HGUP4109 | KF412646 | KF412643 | KF412649 |
| *N. cubana* | **CBS 600.96** | **KM199521** | **KM199438** | **KM199347** |
| *N. dendrobii* | **MFLUCC 14-0106** | **MK975829** | **MK975835** | **MK993571** |
| *N. dendrobii* | MFLUCC 14-0132 | MK975830 | NA | MK993572 |
| *N. drenthii* | BRIP 72264a | MZ344172 | MZ312680 | MZ303787 |
| *N. drenthii* | BRIP 72263a | MZ344171 | MZ312679 | MZ303786 |
| *N. ellipsospora* | **MFLUCC 12-0283** | **JX399047** | **JX399016** | **JX398980** |
| *N. egyptiaca* | **CBS 140162** | **KP943748** | **KP943746** | **KP943747** |
| *N. eucalypticola* | **CBS 264.37** | **KM199551** | **KM199431** | **KM199376** |
| *N. eucalyptorum* | PE194 | MW805398 | MW802831 | MW794098 |
| *N. eucalyptorum* | **CBS 147684** | **MW805397** | **MW802841** | **MW794108** |
| *N. foedans* | **CGMCC 3.9123** | **JX399053** | **JX399022** | **JX398987** |
| *N. formicarum* | **CBS 362.72** | **KM199517** | **KM199455** | **KM199358** |
| *N. formicarum* | CBS 115.83 | KM199519 | KM199444 | KM199344 |
| <u>*N. formicarum*</u> | <u>GUCC 21-0809</u> | <u>OP753367</u> | <u>OP752132</u> | <u>OP498007</u> |
| *N. guajavae* | **FMB0026** | **MH460868** | **MH460871** | **MF783085** |
| *N. guajavicola* | **FMB0129** | **MH460870** | **MH460873** | **MH209245** |
| *N. haikouensis* | **SAUCC212271** | **OK104877** | **OK104870** | **OK087294** |
| *N. haikouensis* | SAUCC212272 | OK104878 | OK104871 | OK087295 |
| *N. hadrolaeliae* | VIC 47180 | MK465122 | MK465120 | MK454709 |
| *N. hispanica* | **CBS 147686** | **MW805399** | **MW802840** | **MW794107** |
| *N. honoluluana* | **CBS 114495** | **KM199548** | **KM199457** | **NR_145245** |
| *N. hydeana* | **MFLUCC 20-0132** | **MW251129** | **MW251119** | **MW266069** |
| *N. hypericin* | <u>**KUNCC 22-12597**</u> | **OP713768** | **OP765908** | **OP498010** |
| <u>*N. hypericin*</u> | KUNCC 22-12598 | OP737880 | OP737883 | OP498009 |
| *N. iberica* | **CBS 147688** | **MW805402** | **MW802844** | **MW794111** |
| *N. iraniensis* | **CBS 137768** | **KM074051** | **KM074057** | **KM074048** |
| *N. javaensis* | **CBS 257 31** | **KM199543** | **KM199437** | **NR_145241** |
| *N. keteleeria* | **MFLUCC 13-0915** | **KJ503822** | **KJ503821** | **KJ503820** |
| *N. longiappendiculata* | **MEAN 1315** | **MW805404** | **MW802845** | **MW794112** |
| *N. lusitanica* | MEAN 1317 | MW805406 | MW802843 | MW794110 |
| *N. lusitanica* | MEAN 1320 | MW805409 | MW802830 | MW794097 |
| *N. macadamiae* | **BRIP 63737c** | **KX186627** | **KX186654** | **NR_161002** |
| *N. maddoxii* | **BRIP 72266a** | **MZ344167** | **MZ312675** | **MZ303782** |
| *N. magna* | **MFLUCC 12-0652** | **KF582791** | **KF582793** | **KF582795** |
| *N. mesopotamica* | **CBS 336.86** | **KM199555** | **KM199441** | **KM199362** |
| *N. musae* | **MFLUCC 15-0776** | **KX789685** | **KX789686** | **NR_156311** |
| *N. natalensis* | **CBS 138.41** | **KM199552** | **KM199466** | **NR_156288** |
| *N. nebuloides* | **BRIP 66617** | **MK977633** | **MK977632** | **MK966338** |
| *N. olumideae* | **BRIP 72273a** | **MZ344175** | **MZ312683** | **MZ303790** |
| *N. pandanicola* | **KUMCC 17-0175** | **MH388389** | **MH412720** | **NA** |
| *N. pernambucana* | **GS 2014-RV01** | **KU306739** | **NA** | **KJ792466** |
| *N. petila* | **MFLUCC 17-1738** | **MK764319** | **MK764341** | **MK764275** |

**TABLE 3** (Continued)

| Taxa[a] | Strain/voucher no. | tef1-α[b] | tub2[b] | ITS[b] |
|---|---|---|---|---|
| **N. phangngaensis** | **MFLUCC 18-0119** | **MH388390** | **MH412721** | **MH388354** |
| **N. photiniae** | **MFLUCC 22-0129** | **OP753368** | **OP752131** | **OP498008** |
| N. photiniae | GUCC 21-0820 | OP828691 | OP896200 | OP806524 |
| N. perukae | FMB0127 | MH523647 | MH460876 | MH209077 |
| **N. piceana** | **CBS 394.48** | **KM199527** | **KM199453** | **KM199368** |
| **N. protearum** | **CBS 114178** | **KM199542** | **KM199463** | **JN712498** |
| **N. psidii** | **FMB0028** | **MH460874** | **MH477870** | **MF783082** |
| **N. rhapidis** | **GUCC21501** | **MW980442** | **MW980441** | **MW931620** |
| N. rhapidis | KUNCC 22-12590 | OP753369 | OP752134 | OP498004 |
| **N. rhizophorae** | **MFLUCC 17-1550** | **MK764321** | **MK764343** | **MK764277** |
| **N. rhododendri** | **GUCC 21504** | **MW980444** | **MW980443** | **MW979577** |
| N. rhododendri | MFLUCC 22-0130 | OP753370 | OP762671 | OP497995 |
| **N. rhododendricola** | **KUN-HKAS-123204** | **OK274148** | **OK274147** | **OK283069** |
| **N. rosae** | **CBS 101057** | **KM199523** | **KM199429** | **KM199359** |
| **N. rosicola** | **CFCC 51992** | **KY885243** | **KY885245** | **KY885239** |
| **N. samarangensis** | **MFLUCC 12-0233** | **JQ968611** | **JQ968610** | **JQ968609** |
| **N. saprophytica** | **MFLUCC 12-0282** | **KM199538** | **KM199433** | **KM199345** |
| N. saprophytica | GUCC 21506 | MW980449 | MW980447 | MW979578 |
| **N. sichuanensis** | **CFCC 54338** | **MW199750** | **MW218524** | **MW166231** |
| N. sichuanensis | SM15-1C | MW199751 | MW218525 | MW166232 |
| **N. siciliana** | **CBS 149117** | **ON107273** | **ON209162** | **ON117813** |
| **N. sonneratae** | **MFLUCC 17-1745** | **MK764323** | **MK764345** | **MK764279** |
| **N. steyaertii** | **IMI 192475** | **KF582792** | **KF582794** | **KF582796** |
| **N. suphanburiensis** | **MFLUCC 22-0126-** | **OP753372** | **OP752135** | **OP497994** |
| **N. surinamensis** | **CBS 450.74** | **KM199518** | **KM199465** | **KM199351** |
| **N. thailandica** | **MFLUCC 17-1730** | **MK764325** | **MK764347** | **MK764281** |
| **N. umbrinospora** | **MFLUCC 12-0285** | **JX399050** | **JX399019** | **JX398984** |
| **N. vheenae** | **BRIP 72293a** | **MZ344177** | **MZ312685** | **MZ303792** |
| **N. vitis** | **MFLUCC 15-1265** | **KU140676** | **KU140685** | **KU140694** |
| N. zakeelii | BRIP 72282a | MZ344174 | MZ312682 | MZ303789 |
| **N. zimbabwana** | **CBS 111495** | **KM199545** | **KM199456** | **MH554855** |
| Neopestalotiopsis sp. 1 | CFCC 54337 | MW199752 | MW218526 | MW166233 |
| Neopestalotiopsis sp. 1 | ZX12-1 | MW199753 | NA | MW166234 |
| Neopestalotiopsis sp. 1 | HKAS 124560 | OP753364 | OP752138 | OP498005 |
| Neopestalotiopsis sp. 1 | KUNCC 22-12592 | OP753365 | OP752140 | OP498006 |
| Neopestalotiopsis sp. 1 | GUCC 21-0808 | OP753366 | OP752139 | OP498011 |
| Neopestalotiopsis sp. 2 | CFCC 54340 | MW199754 | MW218528 | MW166235 |
| Neopestalotiopsis sp. 2 | ZX22B | MW199755 | MW218529 | MW166236 |
| Neopestalotiopsis sp. 2 | MFLUCC 22-0131 | OP753371 | OP752141 | OP497996 |
| Neopestalotiopsis sp. 2 | KUNCC 22-12596 | OP797834 | OP752142 | OP498003 |
| **Pestalotiopsis adusta** | **ICMP 6088** | **JX399070** | **JX399037** | **JX399006** |
| P. adusta | MFLUCC 10-0146 | JX399071 | JX399038 | JX399007 |
| **P. aggestorum** | **LC6301** | **KX895234** | **KX895348** | **KX895015** |
| **P. anacardiacearum** | **IFRDCC 2397** | **KC247156** | **KC247155** | **KC247154** |
| **P. arceuthobii** | **CBS 434.65** | **KM199516** | **KM199427** | **KM199341** |
| **P. appendiculata** | **CGMCC 3.23550** | **OP185509** | **OP185516** | **OP082431** |
| **P. arengae** | **CBS 331.92** | **KM199515** | **KM199426** | **KM199340** |
| **P. australasiae** | **CBS 114126** | **KM199499** | **KM199409** | **KM199297** |
| P. australasiae | CBS 11141 | KM199501 | KM199410 | KM199298 |
| **P. australis** | **CBS 114193** | **KM199475** | **KM199383** | **KM199332** |
| **P. biciliata** | **CBS 124463** | **KM199505** | **KM199399** | **KM199308** |
| P. biciliata | CAA1011 | MW959090 | MW934601 | MW969738 |
| **P. brachiata** | **LC2988** | **KX895150** | **KX895265** | **KX894933** |
| P. brachiata | LC8189 | KY464153 | KY464163 | KY464143 |
| **P. brassicae** | **CBS 170.26** | **KM199558** | **NA** | **KM199379** |
| **P. camelliae** | **MFLUCC 12-0277** | **JX399074** | **JX399041** | **JX399010** |
| **P. camelliae-oleiferae** | **CSUFTCC08** | **OK507963** | **OK562368** | **OK493593** |
| P. camelliae-oleiferae | CSUFTCC09 | OK507964 | OK562369 | OK493594 |
| **P. cangshanensis** | **CGMCC 3.23544** | **OP185510** | **OP185517** | **OP082426** |
| **P. chamaeropis** | **CBS 186.71** | **KM199473** | **KM199391** | **KM199326** |
| P. chamaeropis | KUNCC 22-12591 | OP753373 | OP752130 | OP497998 |
| **P. chiangmaiensis** | **MFLUCC 22-0127** | **OP753374** | **OP752137** | **OP497990** |
| P. chiaroscuro | BRIP 72970 | OK423753 | OK423752 | OK422510 |

**TABLE 3** (Continued)

| Taxa[a] | Strain/voucher no. | tef1-α[b] | tub2[b] | ITS[b] |
|---|---|---|---|---|
| *P. chinensis* | **MFLUCC 12-0273** | **NA** | **NA** | **NR_111786** |
| *P. clavata* | **MFLUCC 12-0268** | **JX399056** | **JX399025** | **JX398990** |
| *P. colombiensis* | **CBS 118553** | **KM199488** | **KM199421** | **KM199307** |
| *P. daliensis* | **CGMCC 3.23548** | **OP185511** | **OP185518** | **OP082429** |
| *P. digitalis* | **ICMP 5434** | **NA** | **KP781883** | **KP781879** |
| *P. diploclisiae* | **CBS 115587** | **KM199486** | **KM199419** | **KM199320** |
| *P. dilucida* | **LC3232** | **KX895178** | **KX895293** | **KX894961** |
| *P. diversiseta* | **MFLUCC 12-0287** | **JX399073** | **JX399040** | **NR_120187** |
| *P. dracaenae* | **HGUP4037** | **MT598644** | **MT598645** | **NA** |
| *P. dracaenicola* | **MFLUCC 18-0913** | **MN962732** | **MN962733** | **MN962731** |
| *P. dracontomelon* | **MFUCC 10-0149** | **KP781880** | **NA** | **KP781877** |
| _P. dracontomelon_ | MFLUCC 22-0122 | OP753375 | OP762672 | NA |
| *P. endophytica* | MFLUCC 18-0932 | MW417119 | NA | NR_172439 |
| *P. ericacearum* | **IFRDCC 2439** | **KC537814** | **KC537821** | **KC537807** |
| *P. etonensis* | BRIP 66615 | MK977635 | MK977634 | MK966339 |
| *P. formosana* | **NTUCC 17-009** | **MH809389** | **MH809385** | **MH809381** |
| *P. furcata* | **MFLUCC 12-0054** | **JQ683740** | **JQ683708** | **JQ683724** |
| *P. fusoidea* | **CGMCC 3.23545** | **OP185512** | **OP185519** | **OP082427** |
| *P. gaultheria* | **IFRD 411.014** | **KC537812** | **KC537819** | **KC537805** |
| *P. gibbosa* | **NOF 3175** | **LC311591** | **LC311590** | **LC311589** |
| *P. grevilleae* | **CBS 114127** | **KM199504** | **KM199407** | **KM199300** |
| *P. hawaiiensis* | **CBS 114491** | **KM199514** | **KM199428** | **KM199339** |
| *P. hispanica* | **CBS 115391** | **MH554399** | **MH554640** | **MH553981** |
| _P. peristrophes_ | KUNCC 22-12595 | OP753381 | OP765910 | OP498001 |
| _P. peristrophes_ | KUNCC 22-12593 | OP753378 | OP737882 | OP498000 |
| _P. peristrophes_ | KUNCC 22-12594 | OP753380 | OP765912 | OP498002 |
| *P. hydei* | **MFLUCC 20-0135** | **MW251113** | **MW251112** | **NR_172003** |
| _P. hydei_ | GUCC 21-0816 | OP753383 | OP765909 | OP753660 |
| *P. hollandica* | **CBS 265.33** | **KM199481** | **KM199388** | **KM199328** |
| *P. hollandica* | MEAN 1091 | MT374691 | MT374703 | MT374678 |
| *P. humus* | **CBS 336.97** | **KM199484** | **KM199420** | **KM199317** |
| *P. hunanensis* | **CSUFTCC15** | **OK507969** | **OK562374** | **OK493599** |
| *P. hunanensis* | CSUFTCC18 | OK507970 | OK562375 | OK493600 |
| *P. iberica* | **CAA1006** | **MW759039** | **MW759036** | **MW732249** |
| *P. inflexa* | **MFLUCC 12-0270** | **JX399072** | **JX399039** | **JX399008** |
| *P. intermedia* | **MFLUCC 12-0259** | **JX399059** | **JX399028** | **JX398993** |
| *P. italiana* | **MFLUCC 12-0657** | **KP781881** | **KP781882** | **KP781878** |
| *P. jesteri* | **CBS 109350** | **KM199554** | **KM199468** | **KM199380** |
| *P. jiangxiensis* | **LC4399** | **KX895227** | **KX895341** | **KX895009** |
| *P. jinchanghensis* | **LC6636** | **KX895247** | **KX895361** | **KX895028** |
| *P. kandelicola* | **NCYU 19-0355** | **MT563101** | **MT563099** | **MT560722** |
| *P. kaki* | **KNU-PT-1804** | **LC553555** | **LC552954** | **LC552953** |
| *P. kenyana* | **CBS 442.67** | **KM199502** | **KM199395** | **KM199302** |
| *P. kenyana* | CBS 911.96 | KM199503 | KM199396 | KM199303 |
| *P. knightiae* | **CBS 114138** | **KM199497** | **KM199408** | **KM199310** |
| *P. knightiae* | CBS 111963 | KM199495 | KM199406 | KM199311 |
| *P. licualacola* | **HGUP4057** | **KC481684** | **KC481683** | **KC492509** |
| *P. linearis* | **MFLUCC 12-0271** | **JX399058** | **JX399027** | **JX398992** |
| _P. loeiana_ | **MFLUCC 22-0123** | **OP737881** | **OP713769** | **OP497988** |
| *P. longiappendiculata* | **LC3013** | **KX895156** | **KX895271** | **KX894939** |
| *P. lushanensis* | **LC4344** | **KX895223** | **KX895337** | **KX895005** |
| *P. lushanensis* | LC8182 | KY464146 | KY464156 | KY464136 |
| *P. macadamiae* | **BRIP 63738B** | **KX186621** | **KX186680** | **KX186588** |
| *P. malayana* | **CBS 102220** | **KM199482** | **KM199411** | **KM199306** |
| *P. monochaeta* | **CBS 144.97** | **KM199479** | **KM199386** | **KM199327** |
| *P. monochaeta* | CBS 440.83 | KM199480 | KM199387 | KM199329 |
| *P. montellica* | **MFLUCC 12-0279** | **JX399076** | **JX399043** | **JX399012** |
| *P. nanjingensis* | **CSUFTCC16** | **OK507972** | **OK562377** | **OK493602** |
| *P. nanjingensis* | CSUFTCC20 | OK507973 | OK562378 | OK493603 |
| *P. nanningensis* | **CSUFTCC10** | **OK507966** | **OK562371** | **OK493596** |
| *P. nanningensis* | CSUFTCC11 | OK507967 | OK562372 | OK493597 |
| *P. neolitseae* | **NTUCC 17-011** | **MH809391** | **MH809387** | **MH809383** |
| *P. novae-hollandiae* | **CBS 130973** | **KM199511** | **KM199425** | **KM199337** |

**TABLE 3** (Continued)

| Taxa[a] | Strain/voucher no. | tef1-α[b] | tub2[b] | ITS[b] |
|---|---|---|---|---|
| **P. oryzae** | **CBS 353.69** | **KM199496** | **KM199398** | **KM199299** |
| P. oryzae | CL107 | MN022941 | MN015425 | MK156295 |
| **P. papuana** | **CBS 331.96** | **KM199491** | **KM199413** | **KM199321** |
| P. papuana | MFLU 19-2764 | MW192204 | MW296942 | MW114337 |
| **P. parva** | **CBS 265.37** | **KM199508** | **KM199404** | **KM199312** |
| **P. pallidotheae** | **MAFF 240993** | **LC311585** | **LC311584** | **NR_111022** |
| **P. photinicola** | **GZCC 16-0028** | **KY047662** | **KY047663** | **KY092404** |
| **P. pinisp** | **CBS 146841** | **MT374694** | **MT374706** | **MT374681** |
| **P. portugalica** | **CBS 393.48** | **KM199510** | **KM199422** | **KM199335** |
| **P. rhizophorae** | **MFLUCC 17-0416** | **MK764327** | **MK764349** | **MK764283** |
| P. rhizophorae | MFLUCC 17-0417 | MK764328 | MK764350 | MK764284 |
| **P. rhododendri** | **IFRDCC 2399** | **KC537811** | **KC537818** | **NR_120265** |
| **P. rhodomyrtus** | **HGUP4230** | **KF412645** | **KF412642** | **KF412648** |
| P. rhodomyrtus | MG7 | MZ126725 | MZ126718 | MZ089458 |
| **P. rosarioides** | **CGMCC 3.23549** | **OP185513** | **OP185520** | **OP082430** |
| **P. rosea** | **MFLUCC 12-0258** | **JX399069** | **JX399036** | **JX399005** |
| **P. scoparia** | **CBS 176.25** | **KM199478** | **KM199393** | **KM199330** |
| **P. sequoiae** | **MFLUCC 13-0399** | **NA** | **NA** | **NR_153271** |
| **P. shandogensis** | **JZB340038** | **MN626740** | **MN626729** | **MN625275** |
| **P. shorea** | **MFLUCC 12-0314** | **KJ503817** | **KJ503814** | **KJ503811** |
| <u>P. smilacicola</u> | <u>MFLUCC 22-0124</u> | <u>OP737879</u> | <u>OP762674</u> | <u>OP497989</u> |
| **<u>P. smilacicola</u>** | **<u>MFLUCC 22-0125</u>** | **OP753376** | **OP762673** | **OP497991** |
| **P. spathulata** | **CBS 356.86** | **KM199513** | **KM199423** | **KM199338** |
| **P. spathuliappendiculata** | **CBS 144035** | **MH554607** | **MH554845** | **MH554172** |
| **P. suae** | **CGMCC 3.23546** | **OP185514** | **OP185521** | **OP082428** |
| **P. telopeae** | **CBS 114161** | **KM199500** | **KM199403** | **KM199296** |
| P. telopeae | CBS 114137 | KM199559 | KM199469 | KM199301 |
| **P. thailandica** | **MFLUCC 17-1616** | **MK764329** | **MK764351** | **MK764285** |
| P. thailandica | MFLUCC 17-1617 | MK764329 | MK764351 | MK764285 |
| **P. trachycarpicola** | **IFRDCC 2240** | **JQ845946** | **JQ845945** | **NR_120109** |
| **P. unicolor** | **MFLUCC 12-0276** | **NA** | **JX399030** | **JX398999** |
| **P. verruculosa** | **MFLUCC 12-0274** | JX399061 | **NA** | **JX398996** |
| **P. yanglingensis** | **LC4553** | **KX895231** | **KX895345** | **KX895012** |
| Pestalotiopsis sp. | LC3637 | KX895210 | KX895324 | KX894993 |
| **Pseudopestalotiopsis ampullacea** | **LC6618** | **KX895244** | **KX895358** | **KX895025** |
| **Ps. annellata** | **NTUCC 17-030** | **MT321988** | **MT321889** | **MT322087** |
| **Ps. avicenniae** | **MFLUCC 17-0434** | **MK764331** | **MK764353** | **MK764287** |
| **Ps. camelliae-sinesis** | **LC3490** | **KX895202** | **KX895316** | **KX894985** |
| **Ps. chinensis** | **LC3011** | **KX895154** | **KX895269** | **KX894937** |
| **Ps. curvatispora** | **MFLUCC 17-1722** | **MK764332** | **MK764354** | **MK764288** |
| **Ps. cocos** | **CBS 272.29** | **KM199553** | **KM199467** | **KM199378** |
| **Ps. dawaina** | **MM14 F0015** | **LC324752** | **LC324751** | **LC324750** |
| **Ps. gilvanii** | **INPA 2913** | **MN385957** | **MN385954** | **MN385951** |
| Ps. gilvanii | INPA 2914 | MN385958 | MN385955 | MN385952 |
| **Ps. ignota** | **NN 42909** | **KU500016** | **NA** | **KU500020** |
| **Ps. indica** | **CBS 459.78** | **KM199560** | **KM199470** | **KM199381** |
| **Ps. ixorae** | **NTUCC 17-001.1** | **MG816336** | **MG816326** | **MG816316** |
| **Ps. kawthaungina** | **MM14 F0083** | **LC324755** | **LC324754** | **LC324753** |
| **Ps. kubahensis** | **UMAS-KUB-P20** | **NA** | **NA** | **KT006749** |
| **Ps. myanmarina** | **NBRC 112264** | **LC114065** | **LC114045** | **LC114025** |
| **Ps. rhizophorae** | **MFLUCC 17-1560** | **MK764335** | **MK764357** | **MK764291** |
| **Ps. simitheae** | **MFLUCC 12-0121** | **KJ503818** | **KJ503815** | **KJ503812** |
| **Ps. solicola** | **CBS 386.97** | **MH554474** | **MH554715** | **NR_161086** |
| **Ps. taiwanensis** | **NTUCC 17-002.1** | **MG816339** | **MG816329** | **MG816319** |
| **Ps. thailandica** | **MFLUCC 17-1724** | **MK764336** | **MK764358** | **MK764292** |
| **Ps. theae** | **MFLUCC 12-0055** | **JQ683743** | **JQ683711** | **JQ683727** |
| Ps. theae | NTUCC 18-067 | MT321987 | MT321888 | MT322086 |
| <u>Ps. theae</u> | <u>MFLUCC 22-0128</u> | OP753377 | OP752136 | OP497993 |
| **Ps. vietnamensis** | **NBRC 112252** | LC114074 | LC114054 | LC114034 |

[a]Ex-type strains are in bold, and the newly generated strains are indicated with underlining.
[b]NA, not available.

**TABLE 4** The best-fit evolutionary models used in our phylogenetic analyses

| Data set | Method | Model | | |
|---|---|---|---|---|
| | | tef1-α | tub2 | ITS |
| *Neopestalotiopsis* | ML | HKY+F+G4 | TNe+I+G4 | TIM2+F+I+G4 |
| | BYPP | GTR+I+G | GTR+I+G | GTR+I+G |
| *Pestalotiopsis* | ML | TN+F+I+G4 | K2P+I+G4 | TPM3u+F+I+G4 |
| | BYPP | GTR+I+G | GTR+I+G | GTR+I+G |
| *Pseudopestalotiopsis* | ML | TIM+F+I+G4 | GTR+F+I+G4 | TIM2+I+G4 |
| | BYPP | GTR+I+G | GTR+I+G | GTR+I+G |

## ACKNOWLEDGMENTS

We thank Shaun Pennycook for checking the nomenclature. We thank Abhaya Balasuriya, the Onsite Visiting Scholars for World Class Research Collaboration Program under the Reinventing University System Project sponsored by Ministry of Higher Education, Science, Research and Innovation, Thailand. Y.R.S. thanks Mae Fah Luang University for the award of a fee-less scholarship. Y.R.S. also thanks Jing-Yi Zhang and Song Wang for collecting the samples. The study was funded by Guizhou Science Technology Department International Cooperation Basic Project ([2018]5806), National Natural Science Foundation of China (31972222 and 31560489), Program of Introducing Talents of Discipline to Universities of China (111 Program, D20023), and Talent project of Guizhou Science and Technology Cooperation Platform ([2017]57885, [2019]5641, and [2020]5001).

We declare no conflicts of interest.

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
