## [Reviewer comments · Microbiology Spectrum]

Microbiology Spectrum

Pestalotioid species associated with medicinal plants in southwest China and Thailand

Ya-Ru Sun, Ruvishika Jayawardena, Jing-E Sun, and Yong Wang

Corresponding Author(s): Yong Wang, Guizhou University

Review Timeline:

Submission Date:	September 30, 2022
Editorial Decision:	October 24, 2022
Revision Received:	November 21, 2022
Accepted:	November 23, 2022

Editor: Florian Freimoser

Reviewer(s): Disclosure of reviewer identity is with reference to reviewer comments included in decision letter(s). The following individuals involved in review of your submission have agreed to reveal their identity: Ishara Manawasinghe (Reviewer #2)

Transaction Report:

DOI: <https://doi.org/10.1128/spectrum.03987-22>

October 24, 2022

Prof. Yong Wang
Guizhou University
Department of Plant Pathology, Agriculture College, Guizhou University
Guiyang 550025
China

Re: Spectrum03987-22 (Pestalotioid species associated with medicinal plants in southwest China and Thailand)

Dear Prof. Yong Wang:

Thank you for submitting your manuscript to Microbiology Spectrum. Your manuscript has been reviewed by two specialists in the field who find your working interesting but also identified questions and concerns. Their comments are given below and are marked in the attached documents. Please also make sure to submit gene sequences to genbank and to deposit new isolates in a culture collection (please provide culture collection numbers).

Link Not Available

Sincerely,

Florian Freimoser

Journals Department
Reviewer comments:

Reviewer #1 (Comments for the Author):

Dear Authors,

This is an important work for fungi associated with medicinal plants and this study enrich the species diversity for Pestalotioid fungal group. However, there are still some mistakes need to be correct, especially, for the genus Pestalotiopsis, there is a newly published paper which introduced 6 new species, you should check this paper and redo the morphology and phylogeny for your paper. Some other comments and suggestions are marked in the attached file.

Reviewer #2 (Comments for the Author):

Dear Authors,

The manuscript entitled "Pestalotioid species associated with medicinal plants in southwest China and Thailand" describes new Pestalotioid species from various hosts got several serious concerns

1. Authors did not follow the recent taxonomic conflicts on *Neopestalotiopsis* and introduced several new species, I would like to get a clear answer why some studies prefer to introduce new species as *Neopestalotiopsis* sp. while you choose to introduce them as new species
2. As I can see in the material and methods, the authors have not chosen 2022 pulsations to construct a phylogenetic tree, thus I am not sure still your species are new. Therefore, both trees required an update following the 2022 papers
3. Add PHI analysis for all new species, if do not have species in the same clade, you can choose species from nearby clades.
4. Did you isolate pathogens? Or do you only observe fruiting bodies from leaf spots? If isolated, please mention it clearly in the methods
5. In descriptions, I recommend rounding off small values, because 0.5 μm is beyond our eye capacity yet your software for measuring is automated. Also, descriptions are consists of different type font types while considering the avg symbol.
6. Since there are a lot of new species, I do not think we required descriptions of new hosts, you can only include material examined and notes.
7. I have several problems related to PHI analysis; in this case, I would prefer to see the alignment. However, the final alignment of the paper is not available for revision.

All other minor comments are given in the file.

Staff Comments:

Preparing Revision Guidelines

Please return the manuscript within 60 days; if you cannot complete the modification within this time period, please contact me. If you do not wish to modify the manuscript and prefer to submit it to another journal, please notify me of your decision immediately so that the manuscript may be formally withdrawn from consideration by Microbiology Spectrum.

Pestalotioid species associated with medicinal plants in southwest China and Thailand

Sun YR^{1,2,3}, Jayawardena RS^{2,3}, Sun JE¹, Wang Y¹

¹Department of Plant Pathology, College of Agriculture, Guizhou University, Guiyang, Guizhou 550025, China.

²Center of Excellence in Fungal Research, Mae Fah Luang University, Chiang Rai 57100, Thailand.

³School of Science, Mae Fah Luang University, Chiang Rai 57100, Thailand.

Corresponding author: Yong Wang (yongwangbis@aliyun.com)

Abstract

In this paper, a total of 27 pestalotioid isolates associated with different medicinal plants from southwest China and Thailand were studied. Based on the morphological examination and multi-gene analyses of three gene loci (TEF, TUB, and ITS), these 27 isolates represent 17 species distributed in three genera; including six new *Neopestalotiopsis* species, four new *Pestalotiopsis* species, and six new records. Genealogical Concordance Phylogenetic Species Recognition (GCPSR) with a pairwise homoplasy index (PHI) test was also carried out to provide additional evidence for the placements of the ten new species. Further, simplifying the description of pestalotioid species is discussed, and a checklist for pestalotioid species associated with medicinal plants worldwide is provided.

Importance

Pestalotioid species are an important fungal group, occurring commonly as important plant pathogens, endophytes, and saprophytes. The study of pestalotioid species associated with medicinal plants is significant for agriculture, industry, and pharmaceutical industry but remains poorly studied. In this study, we reported 10 new pestalotioid species and 6 new records based on morphology and molecular analyses. Our study significantly enriches the species richness of pestalotioid and provides a basis for follow-up studies.

Keywords – 10 new species – 6 new records – Diversity – Endophytes – Plant pathogens – Phylogeny – Saprophytes – Taxonomy

Introduction

Medicinal plants play a crucial role in the development of human cultures. They are a rich source of natural products with both biological and chemical properties. They play a health care or treatment role in various ways and have been used since prehistoric times across the world (Rasool-Hassan 2012, Rasool et al. 2020). It is estimated that more than 70% of the world's population relies on medicinal plants

(David et al. 2015). Microfungi can affect the growth and quality of medicinal plants. Some endophytes isolated from medicinal plants have broad developmental prospects (Guo 2016, Jia et al. 2016). Microfungi associated with medicinal plants have always been among the researched hotspots (Weber et al. 2004, Sun et al. 2008, Abtahi and Nourani 2017, Keshri et al. 2021).

Pestalotioid species is a very common group of fungi, which form important associations with different plants as pathogens, endophytes, or saprophytes, and are widely distributed in tropical and temperate regions (Guba 1961, Barr 1975, Nag Raj 1993, Maharachchikumbura et al. 2014, Hyde et al. 2014, Jayawardena et al. 2019, 2021, Norphanphoun et al. 2019, Ul Haq et al. 2021, Yang et al. 2021). Traditionally taxonomy of pestalotioid species mainly depended on their hosts and color intensities of the median conidial cell (Moreau 1949, Steyaert 1949, Guba 1961, Sutton 1980). With the development of DNA-based phylogenetic analyses, the traditional classification system has been proved unreliable. The use of molecular data in resolving pestalotioid species was revisited by Maharachchikumbura et al. (2014) and they separated this group into three genera, viz. *Neopestalotiopsis*, *Pestalotiopsis*, and *Pseudopestalotiopsis*. *Neopestalotiopsis* differs from *Pseudopestalotiopsis* and *Pestalotiopsis* by its versicolourous (two upper median cells darker than the lowest median cell) median cells and indistinct conidiophores, while *Pseudopestalotiopsis* can be easily distinguished from *Pestalotiopsis* by darker colored concolourous (for those possessing equally pigmented median cells) median cells (Maharachchikumbura et al. 2014).

As important plant pathogens, pestalotioid species are almost ubiquitous in agricultural and non-cultivated ecosystems, causing multiple diseases and huge economic losses (Zhang et al. 2012, Maharachchikumbura et al. 2013, Jayawardena et al. 2016, Liu et al. 2017, Yang et al. 2017, Mahapatra et al. 2018, Diogo et al. 2021, Prasannath et al. 2021). For example, grey blight disease of tea plants is caused by *Pseudopestalotiopsis* spp. and *Pestalotiopsis* spp. and accounts for at least 17 % production damage in southern India (Joshi et al. 2009) and 10–20 % yield loss in Japan (Horikawa 1986). *Neopestalotiopsis clavispora* caused the leaf blight of *Elettaria cardamomu* in India (Biju et al. 2018), and leaf spot of *Taxus chinensis* in China (Wang et al. 2019). Diogo et al. (2021) reported that pestalotioid fungi caused stem girdling and dieback in young eucalyptus plants in Portugal. Li et al. (2021) identified five new pestalotioid species associated with symptomatic leaves of *Camellia oleifera* in China. Thus, it is necessary to study the pathogenic pestalotioid species related to medicinal plants, which could provide the research foundation for the prevention and treatment of diseases and reduce economic losses.

The study of endophytic fungi in medicinal plants is of great significance for elucidating their distribution, growth and developmental characteristics and resource regeneration (Weber et al. 2004, Huang et al. 2008, Sun et al. 2008, Schafhauser et al. 2019, Keshri et al. 2021). Many pestalotioid fungi have been found as endophytes from different medicinal plants with richer secondary metabolites (Huang et al. 2008, Jia et al. 2016, Xu et al. 2010, 2014, Reddy et al. 2016, Ma et al. 2019). For example,

the endophytic fungus, *Pestalotiopsis versicolor*, was isolated from the healthy leaves of *Taxus cuspidata* and it is an excellent candidate for an alternate source of Taxol supply (Kumaran et al. 2010). Therefore, the study of endophytic pestalotioid species related to medicinal plants could be of very importance to pharmaceuticals and therapeutic medicine.

This study aims to identify the pestalotioid fungi associated with medicinal plants in southwest China and Thailand based on morphology, phylogenetic analyses, and GCPSR. This paper describes, illustrates, and compares ten new species and six new records with allied species. In addition, we provided a checklist for pestalotioid species associated with medicinal plants worldwide.

Materials and methods

Collection, examination and isolation

The fresh samples of different medicinal plants were collected in southwest China and Thailand from 2019 to 2022. Samples were brought to the laboratory in Ziplock plastic bags or paper envelopes for examination. The fruiting bodies on natural substrates were observed and photographed using a stereo-microscope (SteREO Discovery, V12, Carl Zeiss Microscopy GmbH, Germany; VHX-7000, Keyence, Japan). Morphological characters were observed using a Nikon ECLIPSE Ni compound microscope (Nikon, Japan) and photographed with a Nikon DS-Ri2 digital camera (Nikon, Japan); Carl Zeiss compound microscope (Carl Zeiss AG, Germany) and photographed with an Axiocam 208 color digital camera (Carl Zeiss AG, Germany). The photo plates were made by the Adobe Photoshop CS6 Extended v. 13.0 software. Measurements were obtained with the Tarosoft (R) Image Frame Work software.

For pathogens and saprophytes, single spore isolations were used to obtain pure cultures, following the methods described by Senanayake et al. (2020). Germinated conidia were transferred to fresh potato dextrose agar (PDA) plates and incubated at 25°C for four weeks. For endophytes, materials were washed under running tap water, immersed in 70% ethanol for 1 min, followed by soaking in 4% NaOCl for 1 min, rinsed three times in sterile distilled water, and dried on a sterile filter paper. For the control, the final sterile water rinsed was plated and observed during the post-incubation period. The absence of any fungal (microbial) growth indicated that the leaf surface was sterile (Kjer et al. 2010). The sterilized materials were cut into 2-5 mm² segments and placed on PDA containing 50 µg/ml penicillin and 50 µg/ml streptomycin (Otero et al. 2002). The plates were observed daily and the mycelial growth on the edge of fungal colonies were transferred to fresh PDA dishes to obtain pure cultures. The pure cultures were deposited in Mae Fah Luang University Culture Collection (MFLUCC), Chiang Rai, Thailand and the Culture Collection of the Department of Plant Pathology, Agriculture College, Guizhou University (GUCC), Guiyang, China. Specimens were deposited in the herbarium of Mae Fah Luang University (MFLU) Chiang Rai, Thailand, the herbarium of the Department of Plant Pathology, Agricultural College, Guizhou University (HGUP), Guiyang, China, and

the Herbarium of Cryptogams, Kunming Institute of Botany, Academia Sinica (HKAS), Kunming, China. Facesoffungi (FoF) and Index Fungorum numbers were acquired as described in Jayasiri et al. (2015) and in Index Fungorum (2022).

DNA extraction, PCR amplification and sequencing

BIOMIGA Fungus Genomic DNA Extraction Kit (GD2416, BIOMIGA, San Diego, California, USA) was used to extract DNA from fresh fungal mycelia, which were grown on PDA medium for 4 weeks at 25 °C. Three genes were selected in this study: the internal transcribed spacers (ITS), the translation elongation factor 1 (TEF), and the partial β -tubulin region (TUB). Polymerase chain reaction (PCR) was carried out in 20 μ L of reaction volume which contained 10 μ L 2 \times PCR Master Mix, 7 μ L of ddH₂O, 1 μ L of each primer, and 1 μ L of template DNA. The PCR thermal cycle program and primers are given (Table 1). Purification and sequencing of PCR products were carried out at SinoGenoMax (Beijing) Co., China.

Table 1. Primers and PCR procedures used in this study.

Locus	Primers		PCR procedures	Reference
	Name	Sequence (5'-3')		
ITS	ITS5	GGAAGTAAAAGTCGTAACAAGG	94 °C 3 min; 94 °C 30 s; 52 °C 30 s; 72 °C 1 min; Repeat 2–4	White et al. (1990)
	ITS4	TCCTCCGCTTATTGATATGC	for 35 cycles; 72 °C 8 min; 4 °C on hold	
TEF	EF1-728F	CAT CGA GAA GTT CGA GAA GG	94 °C 5 min; 94 °C 30 sec; 52 °C 30 sec; 72 °C 1 min;	Carbone & Kohn (1999), O'Donnell et al. (1998)
	EF2	GGA RGT ACC AGT SAT CAT GTT	Repeat 2–4 for 40 cycles; 72 °C 8 min; 4 °C on hold	
TUB	T1	AACATGCGTGAGATTGTAAGT	95 °C 3 min; 94 °C 30 sec; 55 °C 50 sec; 72 °C 1 min;	Glass & Donaldson (1995), O'Donnell & Cigel'nik (1997)
	Bt2b	ACCCTCAGTGTAGTGACCCTTGGC	Repeat 2–4 for 40 cycles; 72 °C 8 min; 4 °C on hold	

Table 2. The best-fit evolutionary models used in our phylogenetic analyses

Data set	Method	Model		
		TEF	TUB	ITS
Neopestalotiopsis	ML	HKY+F+G4	TNe+I+G4	TIM2+F+I+G4
	BYPP	GTR+I+G	GTR+I+G	GTR+I+G TPM3u+F+I+G
Pestalotiopsis	ML	TN+F+I+G4	K2P+I+G4	4
	BYPP	GTR+I+G	GTR+I+G	GTR+I+G
Pseudopestalotiopsis	ML	TIM+F+I+G4	GTR+F+I+G4	TIM2+I+G4
	BYPP	GTR+I+G	GTR+I+G	GTR+I+G

Phylogenetic analyses

Phylogenetic analysis and its preparatory work were carried out according to the method described in Dissanayake et al. (2020). The sequences were compared against the NCBI GenBank nucleotide database using BLASTn to evaluate the closest phylogenetic matches. All sequences used in this study were downloaded from GenBank referring to previous studies (Maharachchikumbura et al. 2016b, Liu et al. 2017, Nozawa et al. 2017, Norphanphoun et al. 2019, Prasannath et al. 2021) (Table 3). The single gene sequences were checked with BioEdit v. 7.0.9.0 (Hall 1999). Alignments for each locus were generated using MAFFT v.7 (<https://mafft.cbrc.jp/alignment/server/>) and manually improved using AliView (Larsson 2014). The final single-gene alignments were combined by SequenceMatrix 1.7.8 (Vaidya et al. 2011).

The maximum likelihood (ML) analyses were carried out using IQ-TREE (Nguyen et al. 2015, Trifinopoulos et al. 2016) on the IQ-TREE web server (<http://iqtree.cibiv.univie.ac.at>, 1 May 2022) under partitioned models. The best-fit substitution models (Table 2) were determined by jModelTest (Chernomor et al. 2016). Ultrafast bootstrap analyses were implemented with 1,000 replicates (Minh et al. 2013, Hoang et al. 2018).

The Bayesian posterior probability (BYPP) analyses were performed in CIPRES (Miller et al. 2010) with MrBayes on XSEDE 3.2.7a (Ronquist et al. 2012). MrModeltest 2.2 (Nylander 2004) was used to evaluate the best nucleotide substitution models (Table 2) for each data. The Markov chain Monte Carlo (MCMC) sampling approach was used to calculate posterior probabilities (PP) (Rannala and Yang 1996). Six simultaneous Markov chains were run for 10 million generations and trees were sampled every 1,000th generation. The first 20% of trees, representing the burn-in phase of the analyses, were discarded and the remaining trees were used for calculating posterior probabilities in the majority rule consensus tree.

The final phylogenetic trees were viewed and changed format with FigTree v1.4.0 (Rambaut and Drummond 2008) and modified in Microsoft Office PowerPoint 2010. The new sequences used in this study were deposited in GenBank.

Genealogical Concordance Phylogenetic Species Recognition Analysis

The Genealogical Concordance Phylogenetic Species Recognition (GCPSR) was used to analyze the new species, their species boundaries, and their most closely related taxa as described by Quaedvlieg et al. (2014). The pairwise homoplasy index (PHI) test (Bruen et al. 2006) was performed in SplitsTree4 (Huson 1998, Huson and Bryant 2006). If the PHI is above 0.05 threshold ($\Phi_w > 0.05$), it indicates there is no significant recombination present in the dataset. The relationships between closely related species were visualized in splits graphs with both, the Log-Det transformation and splits decomposition options.

Table 3. Taxa of the three genera; *Neopestalotiopsis*, *Pestalotiopsis*, and *Pseudopestalotiopsis* used in the phylogenetic analysis with the corresponding GenBank accession numbers.

Taxa	Strain_number	TEF	TUB	ITS
Neopestalotiopsis acrostichi	MFLUCC 17-1754	MK764316	MK764338	MK764272
N. alpapicalis	MFLUCC 17-2544	MK463547	MK463545	MK357772
N. amomumica	HKAS 124563			
N. amomumica	HKAS 124564			
N. aotearoa	CBS 367.54	KM199526	KM199454	KM199369
N. asiatica	MFLUCC 12-0286	JX399049	JX399018	JX398983
N. australis	CBS 114159	KM199537	KM199432	KM199348
N. brachiata	MFLUCC 17-1555	MK764318	MK764340	MK764274
N. brasiliensis	COAD 2166	MG692402	MG692400	MG686469
N. camelliae-oleiferae	CSUFTCC81	OK507955	OK562360	OK493585
N. camelliae-oleiferae	CSUFTCC82	OK507956	OK562361	OK493586
N. cavernicola	KUMCC 20-0269	MW550735	MW557596	MW545802
N. cavernicola	KUMCC 20-0332	MW590327	MW590328	MW581238
N. Chiangmaiensis	MFLUCC 18-0113	MH388404	MH412725	N/A
N. chinensis	HKAS 124565			
N. chinensis	HKAS 124565			
N. chinensis	GUCC 808			
N. chrysea	MFLUCC 12-0261	JX399051	JX399020	JX398985
N. clavispora	MFLUCC 12-0281	JX399045	JX399014	JX398979
N. cocoas	MFLUCC 15-0152	KX789689	N/A	NR_156312
N. coffea arabicae	HGUP4015	KF412644	KF412641	KF412647
N. cubana	CBS 600.96	KM199521	KM199438	KM199347
N. dendrobii	MFLUCC 14-0106	MK975829	MK975835	MK993571
N. dendrobii	MFLUCC 14-0132	MK975830	N/A	MK993572
N. drethii	BRIP 72264a	MZ344172	MZ312680	MZ303787
N. drethii	BRIP 72263a	MZ344171	MZ312679	MZ303786
N. ellipospora	MFLUCC 12-0283	JX399047	JX399016	JX398980
N. egyptiaca	CBS 140162	KP943748	KP943746	KP943747
N. eucalypticola	CBS 264.37	KM199551	KM199431	KM199376
N. eucalyptorum	PE194	MW805398	MW802831	MW794098
N. eucalyptorum	CBS 147684	MW805397	MW802841	MW794108
N. foedans	CGMCC 2-0123	JX399053	JX399022	JX398987
N. formicarum	CBS 362.72	KM199517	KM199455	KM199358
N. formicarum	CBS 115.83	KM199519	KM199444	KM199344
N. formicarum	GUCC 809			
N. guajavae	FMB0026	MH460868	MH460871	MF783085
N. guajavicola	FMB0129	MH460870	MH460873	MH209245
N. haikouensis	SAUCC212272	OK104877	OK104870	OK087294
N. haikouensis	SAUCC212272	OK104878	OK104871	OK087295
N. hadrolaeliae	VIC 47180	MK465122	MK465120	MK454709
N. hispanica	CBS 147686	MW805399	MW802840	MW794107
N. honoluluana	CBS 114495	KM199548	KM199457	NR_145245
N. hydeana	MFLUCC 20-0132	MW251129	MW251119	MW266069

N. hyperica	HKAS 124561			
N. hyperica	GUCC 811			
N. iberica	CBS 147688	MW805402	MW802844	MW794111
N. iraniensis	CBS 137768	KM074051	KM074057	KM074048
N. javaensis	CBS 257 31	KM199543	KM199437	NR_145241
N. keteleeria	MFLUCC 13-0915	KJ503822	KJ503821	KJ503820
N. longiappendiculata	MEAN 1315	MW805404	MW802845	MW794112
N. lusitanica	MEAN 1317	MW805406	MW802843	MW794110
N. lusitanica	MEAN 1320	MW805409	MW802830	MW794097
N. macadamiae	BRIP 63737c	KX186627	KX186654	NR_161002
N. maddoxii	BRIP 72266a	MZ344167	MZ312675	MZ303782
N. magna	MFLUCC 12-0652	KF582791	KF582793	KF582795
N. mesopotamica	CBS 336.86	KM199555	KM199441	KM199362
N. musae	MFLUCC 15-0776	KX789685	KX789686	NR_156311
N. natalensis	CBS 138.41	KM199552	KM199466	NR_156288
N. nebuloides	BRIP 66617	MK977633	MK977632	MK966338
N. olumideae	BRIP 72273a	MZ344175	MZ312683	MZ303790
N. pandanicola	KUMCC 17-0175	MH388389	MH412720	N/A
N. pernambucana	GS 2014-RV01	KU306739	N/A	KJ792466
N. petila	MFLUCC 17-1738	MK764319	MK764341	MK764275
N. phangngaensis	MFLUCC 18-0119	MH388390	MH412721	MH388354
N. photiniana	GUCC 810			
N. perukae	FMB0127	MH523647	MH460876	MH209077
N. piceana	CBS 394.48	KM199527	KM199453	KM199368
N. protearum	CBS 114178	KM199542	KM199463	JN712498
N. psidii	FMB0028	MH460874	MH477870	MF783082
N. rhapsidis	GUCC21501	MW980442	MW980441	MW931620
N. rhapsidis	HKAS 124559			
N. rhizophorae	MFLUCC 17-1550	MK764321	MK764343	MK764277
N. rhododendri	GUCC 2150	MW980444	MW980443	MW979577
N. rhododendri	MFLUCC 22-			
N. rosae	CBS 101057	KM199523	KM199429	KM199359
N. rosicola	CFCC 51992	KY885243	KY885245	KY885239
N. samarangensis	MFLUCC 12-0233	JQ968611	JQ968610	JQ968609
N. saprophytica	MFLUCC 12-0282	KM199538	KM199433	KM199345
N. sichuanensis	CFCC 54338	MW199750	MW218524	MW166231
N. sichuanensis	SM15-1C	MW199751	MW218525	MW166232
N. sonneratae	MFLUCC 17-1745	MK764323	MK764345	MK764279
N. steyaertii	IMI 192475	KF582792	KF582794	KF582796
N. subtropicalis	GUCC 805			
N. subtropicalis	MFLUCC 22-			
N. suphanburica	MFLUCC 22-			
N. surinamensis	CBS 450 74	KM199518	KM199465	KM199351
N. thailandica	MFLUCC 17-1730	MK764325	MK764347	MK764281

N. umbrinospora	MFLUCC 12-0285	JX399050	JX399019	JX398984
N. vheena	BRIP 72293a	MZ344177	MZ312685	MZ303792
N. vitis	MFLUCC 15-1265	KU140676	KU140685	KU140694
N. zakeelii	BRIP 72282a	MZ344174	MZ312682	MZ303789
N. zimbabwana	CBS 111495	KM199545	KM199456	MH554855
Neopestalotiopsis sp1	CFCC 54337	MW199752	MW218526	MW166233
Neopestalotiopsis sp1	ZX12-1	MW199753	N/A	MW166234
Neopestalotiopsis sp2	CFCC 54340	MW199754	MW218528	MW166235
Neopestalotiopsis sp2	ZX22B	MW199755	MW218529	MW166236
Pestalotiopsis adusta	ICMP 6088	JX399070	JX399037	JX399006
P. adusta	MFLUCC 10-0146	JX399071	JX399038	JX399007
P. aggestorum	LC6301	KX895234	KX895348	KX895015
P. anacardiacearum	IFRDCC 2397	KC247156	KC247155	KC247154
P. arceuthobii	CBS 434.65	KM199516	KM199427	KM199341
P. arengae	CBS 331.92	KM199515	KM199426	KM199340
P. australasiae	CBS 114126	KM199499	KM199409	KM199297
P. australasiae	CBS 11141	KM199501	KM199410	KM199298
P. australis	CBS 114193	KM199475	KM199383	KM199332
P. biciliata	CBS 124463	KM199505	KM199399	KM199308
P. biciliata	CAA1011	MW959090	MW934601	MW969738
P. brachiata	LC2988	KX895150	KX895265	KX894933
P. brachiata	LC8189	KY464153	KY464163	KY464143
P. brassicae	CBS 170.26	KM199558	N/A	KM199379
P. camelliae	MFLUCC 12-0277	JX399074	JX399041	JX399010
P. camelliae-oleiferae	CSUFTCC08	OK507963	OK562368	OK493593
P. camelliae-oleiferae	CSUFTCC09	OK507964	OK562369	OK493594
P. chamaeropis	CBS 186.71	KM199473	KM199391	KM199326
P. chamaeropis	GUCC 800			
P. chiangmaiensis	MFLUCC 22-			
P. chiangmaiensis	MFLUCC 22-			
P. chinensis	MFLUCC 12-0273	N/A	N/A	NR_111786
P. clavata	MFLUCC 12-0268	JX399056	JX399025	JX398990
P. colombiensis	CBS 118553	KM199488	KM199421	KM199307
P. digitalis	ICMP 5434	N/A	KP781883	KP781879
P. diploclisiae	CBS 115587	KM199486	KM199419	KM199320
P. dilucida	LC3232	KX895178	KX895293	KX894961
P. diversiseta	MFLUCC 12-0287	JX399073	JX399040	NR_120187
P. dracaenae	HGUP4037	MT598644	MT598645	
P. dracaenicola	MFLUCC 18-0913	MN962732	MN962733	MN962731
P. dracontomelon	MFUCC 10-0149	KP781880	N/A	KP781877
P. dracontomelon	MFLUCC 22-			
P. endophytica	MFLUCC 18-0932	MW417119	N/A	NR_172439
P. ericacearum	IFRDCC 2439	KC537814	KC537821	KC537807
P. etonensis	BRIP 66615	MK977635	MK977634	MK966339

P. formosana	NTUCC 17-009	MH809389	MH809385	MH809381
P. furcata	MFLUCC 12-0054	JQ683740	JQ683708	JQ683724
P. gaultheria	IFRD 411.014	KC537812	KC537819	KC537805
P. gibbosa	NOF 3175	LC311591	LC311590	LC311589
P. grevilleae	CBS 114127	KM199504	KM199407	KM199300
P. hawaiiensis	CBS 114491	KM199514	KM199428	KM199339
P. hydei	MFLUCC 20-0135	MW251113	MW251112	NR_172003
P. hydei	GUCC 816			
P. hollandica	CBS 265.33	KM199481	KM199388	KM199328
P. hollandica	MEAN 1091	MT374691	MT374703	MT374678
P. humus	CBS 336.97	KM199484	KM199420	KM199317
P. hunanensis	CSUFTCC15	OK507969	OK562374	OK493599
P. hunanensis	CSUFTCC18	OK507970	OK562375	OK493600
P. iberica	CAA1006	MW759039	MW759036	MW732249
P. inflexa	MFLUCC 12-0270	JX399072	JX399039	JX399008
P. intermedia	MFLUCC 12-0259	JX399059	JX399028	JX398993
P. italiana	MFLUCC 12-0657	KP781881	KP781882	KP781878
P. jesteri	CBS 109350	KM199554	KM199468	KM199380
P. jiangxiensis	LC4399	KX895227	KX895341	KX895009
P. jinchanghensis	LC6636	KX895247	KX895361	KX895028
P. kandelicola	NCYU 19-0355	MT563101	MT563099	MT560722
P. kaki	KNU-PT-1804	LC553555	LC552954	LC552953
P. kenyana	CBS 442.67	KM199502	KM199395	KM199302
P. kenyana	CBS 911.96	KM199503	KM199396	KM199303
P. knightiae	CBS 114138	KM199497	KM199408	KM199310
P. knightiae	CBS 111963	KM199495	KM199406	KM199311
P. licualacola	HGUP4057	KC481684	KC481683	KC492509
P. linearis	MFLUCC 12-0271	JX399058	JX399027	JX398992
P. longiappendiculata	LC3013	KX895156	KX895271	KX894939
P. lushanensis	LC4344	KX895223	KX895337	KX895005
P. lushanensis	LC8182	KY464146	KY464156	KY464136
P. macadamiae	BRIP 63738B	KX186621	KX186680	KX186588
P. malayana	CBS 102220	KM199482	KM199411	KM199306
P. monochaeta	CBS 144.97	KM199479	KM199386	KM199327
P. monochaeta	CBS 440.83	KM199480	KM199387	KM199329
P. montellica	MFLUCC 12-0279	JX399076	JX399043	JX399012
P. nanjingensis	CSUFTCC16	OK507972	OK562377	OK493602
P. nanjingensis	CSUFTCC20	OK507973	OK562378	OK493603
P. nanningensis	CSUFTCC10	OK507966	OK562371	OK493596
P. nanningensis	CSUFTCC11	OK507967	OK562372	OK493597
P. neolitseae	NTUCC 17-011	MH809391	MH809387	MH809383
P. novae-hollandiae	CBS 130973	KM199511	KM199425	KM199337
P. oryzae	CBS 353.69	KM199496	KM199398	KM199299
P. oryzae	CL107	MN022941	MN015425	MK156295

P. papuana	CBS 331.96	KM199491	KM199413	KM199321
P. papuana	MFLU 19-2764	MW192204	MW296942	MW114337
P. parva	CBS 265.37	KM199508	KM199404	KM199312
P. pallidotheae	MAFF 240993	LC311585	LC311584	NR_111022
P. peristrophidis	GUCC 803			
P. peristrophidis	GUCC 804			
P. peristrophidis	GUCC 802			
P. peristrophidis	GUCC 802			
P. photinicola	GZCC 16-0028	KY047662	KY047663	KY092404
P. pinisp	CBS 146841	MT374694	MT374706	MT374681
P. portugalia	CBS 393.48	KM199510	KM199422	KM199335
P. rhizophorae	MFLUCC 17-0416	MK764327	MK764349	MK764283
P. rhizophorae	MFLUCC 17-0417	MK764328	MK764350	MK764284
P. rhododendri	IFRDCC 2399	KC537811	KC537818	NR_120265
P. rhodomyrtus	HGUP4230	KF412645	KF412642	KF412648
P. rhodomyrtus	MG7	MZ126725	MZ126718	MZ089458
P. rosea	MFLUCC 12-0258	JX399069	JX399036	JX399005
P. scoparia	CBS 176.25	KM199478	KM199393	KM199330
P. sequoiae	MFLUCC 13-0399	N/A	N/A	NR_153271
P. shandogensis	JZB340038	MN62674	MN626729	MN625275
P. shorea	MFLUCC 12-0314	KJ503817	KJ503814	KJ503811
P. smilaxe	MFLUCC 22-			
P. smilaxe	MFLUCC 22-			
P. spathulata	CBS 356.86	KM199513	KM199423	KM199338
P. spathuliappendiculata	CBS 144035	MH554607	MH554845	MH554172
P. telopeae	CBS 114161	KM199500	KM199403	KM199296
P. telopeae	CBS 114137	KM199559	KM199469	KM199301
P. thailandica	MFLUCC 17-1616	MK764329	MK764351	MK764285
P. thailandica	MFLUCC 17-1617	MK764329	MK764351	MK764285
P. trachycarpicola	IFRDCC 2240	JQ845946	JQ845945	NR_120109
P. unicolor	MFLUCC 12-0276	N/A	JX399030	JX398999
P. verruculosa	MFLUCC 12-0274	JX399061	N/A	JX398996
P. yanglingensis	LC4553	KX895231	KX895345	KX895012
Pestalotiopsis sp	LC3637	KX895210	KX895324	KX894993
Pseudopestalotiopsis				
ampullacea	LC6618	KX895244	KX895358	KX895025
Ps. annellata	NTUCC 17-030	MT321988	MT321889	MT322087
Ps. avicenniae	MFLUCC 17-0434	MK764331	MK764353	MK764287
Ps. camelliae-sinesis	LC3490	KX895202	KX895316	KX894985
Ps. chinensis	LC3011	KX895154	KX895269	KX894937
Ps. curvatispora	MFLUCC 17-1722	MK764332	MK764354	MK764288
Ps. cocos	CBS 272.29	KM199553	KM199467	KM199378
Ps. dawaina	MM14 F0015	LC324752	LC324751	LC324750

Ps. gilvanii	INPA 2913	MN385957	MN385954	MN385951
Ps. gilvanii	INPA 2914	MN385958	MN385955	MN385952
Ps. ignota	NN 42909	KU500016	N/A	KU500020
Ps. indica	CBS 459.78	KM199560	KM199470	KM199381
Ps. ixorae	NTUCC 17-001.1	MG816336	MG816326	MG816316
Ps. kawthaungina	MM14 F0083	LC324755	LC324754	LC324753
Ps. kubahensis	UMAS-KUB-P20	N/A	N/A	KT006749
Ps. myanmarina	NBRC 112264	LC114065	LC114045	LC114025
Ps. rhizophorae	MFLUCC 17-1560	MK764335	MK764357	MK764291
Ps. simitheae	MFLUCC 12-0121	KJ503818	KJ503815	KJ503812
Ps. solicola	CBS 386.97	MH554474	MH554715	NR_161086
Ps. taiwanensis	NTUCC 17-002.1	MG816339	MG816329	MG816319
Ps. thailandica	MFLUCC 17-1724	MK764336	MK764358	MK764292
Ps. theae	MFLUCC 12-0055	JQ683743	JQ683711	JQ683727
Ps. theae	MFLUCC 22-			
Ps. vietnamensis	NBRC 112252	LC114074	LC114054	LC114034

Ex-type strains are in bold. The newly generated strains are indicated in red. N/A: Not available.

Results

Part1: *Neopestalotiopsis*

Phylogenetic analyses

The combined datasets consist of 98 *Neopestalotiopsis* strains along with the outgroup *Pestalotiopsis diversiseta* (MFLUCC 12-0287) and *P. spathulata* (CBS 356.86), which were analyzed to infer the interspecific relationships within *Neopestalotiopsis*. The aligned sequence matrix comprised TEF (1–500), TUB (501–1,015), and ITS (1,016–1,453), sequence data for a total of 1,305 characters, including coded alignment gaps. Similar tree topologies were obtained by ML and BYPP methods, and the most likely tree ($-\ln = 7569.2805$) is presented in Figure 1. The phylogenetic tree which analyzed the 14 *Neopestalotiopsis* isolates from medicinal plants indicated six novel species and two new records.

Figure 1. Cont.

Figure 1 – Maximum likelihood (RAxML) tree for *Neopestalotiopsis*, based on the analysis of a combined dataset of TEF, TUB and ITS sequence data. The tree is rooted with *Pestalotiopsis diversiseta* (MFLUCC 12-0287) and *P. spathulata* (CBS 356.86). Bootstrap support values for ML greater than 75% and Bayesian posterior probabilities greater than 0.95 are given near nodes, respectively. The new isolates are in red.

Genealogical Concordance Phylogenetic Species Recognition Analysis

The PHI test revealed that there is no significant recombination ($\Phi_w = 1.0$), between *N. photiniana* and its closely related taxa *N. chinensis* (HKAS 124560), *N. formicarum* (CBS 362.72), *N. sichuanensis* (CFCC 54338) and *N. vheenae* (BRIP 72293a) (Fig. 2a). *Neopestalotiopsis hyperica* based on the PHI test resulted that there is no significant recombination ($\Phi_w = 1.0$), between *N. hyperica* and its closely related taxa *N. acrostichi* (MFLUCC 17-1754), *N. olumideae* (BRIP 72273a), *N. protearum* (CBS 114178) and *N. rhododendri* (GUCC 2150) (Fig. 2b). Similar result

also occurs in *N. subtropicalis* ($\Phi_w = 1.0$) (Fig. 2c) and *N. suphanburica* ($\Phi_w = 1.0$) (Fig. 2d), indicating there is no significant recombination between them and their closely related taxa.

Figure 2 – Split graphs showing the results of PHI test of new *Neopestalotiopsis* species with their most closely related species using Log-Det transformation and splits decomposition options. The new taxon in each graph is shown in red font.

Taxonomy

Neopestalotiopsis amomumica Y.R. Sun & Yong Wang bis, sp. nov. Fig. 3

Index Fungorum number: IFXX; Facesoffungi number: FoF XX

Etymology: Refers to the name of the host plant from which the fungus was isolated.

Holotype – HKAS 124563

Associated with leaf blight of *Amomum villosum*. *Symptoms* irregular shape, pale to brown, slightly sunken spots appear on the leaves of *Amomum villosum*, which later expand outwards. **Sexual morph:** Not observed. **Asexual morph:** *Conidiomata* solitary, subglobose to globose, unilocular, brown, semi-immersed on leaves. *Conidiophores* indistinct, often reduced to conidiogenous cells. *Conidiogenous cells* 1.2–2.0 μm wide, subcylindrical, ampulliform, hyaline. *Conidia* 18.5–30.0 \times 4.5–7.5 μm , $\bar{x} \pm \text{SD} = 25 \pm 2.5 \times 6.0 \pm 0.68 \mu\text{m}$ (n = 40), L/W ratio = 4.2, fusiform, straight to slightly curved, 4-septate; basal cell obconic with truncate base, hyaline, smooth-walled, 3.2–6.5 μm long ($\bar{x} \pm \text{SD} = 4.7 \pm 0.75 \mu\text{m}$); three median cells 12.0–19.0 μm long ($\bar{x} \pm \text{SD} = 15.7 \pm 1.6 \mu\text{m}$), pale brown to brown, concolourous, wall rugose, septa darker than the rest of the cell; second cell from base pale brown to

brown, 3.5–8.0 μm long ($\bar{x} \pm \text{SD} = 5.6 \pm 0.93 \mu\text{m}$); third cell pale brown to brown, 3.5–6.8 μm long ($\bar{x} \pm \text{SD} = 5.0 \pm 0.84 \mu\text{m}$); fourth cell pale brown to brown, 3.5–6.5 μm long ($\bar{x} \pm \text{SD} = 5.0 \pm 0.7 \mu\text{m}$); apical cell 1.9–4.6 μm long ($\bar{x} \pm \text{SD} = 3.5 \pm 0.8 \mu\text{m}$), hyaline, conic to acute; with 2–3 tubular appendages on apical cell, inserted at different loci in a crest at the apex of the apical cell, unbranched, 7.0–17.0 μm long ($\bar{x} \pm \text{SD} = 12.5 \pm 2.8 \mu\text{m}$); single basal appendage, unbranched, tubular, centric, 2.5–5.0 μm long ($\bar{x} \pm \text{SD} = 3.8 \pm 0.7 \mu\text{m}$).

Culture characteristics – Conidia germinated on PDA within 12 hours from single-spore isolation. Colony diameter reached 8 cm after two weeks at 25 °C on PDA media, circular, surface rough, flat, white from above and below.

Material examined – China, Guizhou Province, Qiannan Buyei and Miao Autonomous Prefecture, Luodian District, leaf blight of *Amomum villosum* Lour. (Zingiberaceae), 3 September 2021, Y.R. Sun, L8 (HKAS 124563, holotype); ex-type-living cultures, GUCC 814, *ibid.*, on leaf blight of *Amomum villosum*, 3 September 2021, Y.R. Sun, L8-1 (HKAS 124564); living cultures, GUCC 815.

Notes – *Neopestalotiopsis amomumica* was isolated from the diseased leaves of *Amomum villosum* in China. Two strains GUCC 814 and GCUU 815 clustered together with good support (ML-BS = 99%, BYPP = 1) and formed a sister clade to *N. zingiberis* (GUCC 21001) which was also isolated from a Zingiberaceae plant (He et al. 2022). The former differs in producing thinner conidia with basal appendages (4.5–7.5 μm in *N. amomumica* vs. 6–9.5 μm in HN89-1, without basal appendage). In addition, there are 4 base pairs differences between GUCC 814 and GUCC 21001 in the ITS gene and 10 base pairs differences in the TEF gene. *Neopestalotiopsis amomumica* also differs by smaller conidia (18.5–30.0 \times 4.5–7.5 μm vs. 42–46 \times 9.5–12 μm) from *N. magna*. Thus, we introduce *N. amomumica* as a new species. Due to the lack of enough closely related species, the GCSPP was not used to evaluate its placement.

Figure 3 – *Neopestalotiopsis amomumica* (MFLU 22-xx, holotype). a Host. b Leaf blight on *Acrostichum aureum*. c Close up view of conidiomata. d conidiogenous cells. e Immature conidia attached to conidiogenous cells. f–i Conidia. j Germinated conidium. Scale bars: d, e = 10 μ m, f–j = 20 μ m.

Neopestalotiopsis chinensis Y.R. Sun & Yong Wang bis, sp. nov.

Fig. 4

Index Fungorum number: IFXX; Facesoffungi number: FoF XX

Etymology: The specific epithet is referring to China, the country from where the taxa were isolated.

Holotype – HKAS 124560

Associated with leaf spot of *Cyrtomium fortune*, *Lithocarpus* sp. and *Smilax scobinicaulis*. Symptoms irregular shape, pale brown, small spots gradually enlarged, changing to brown circular ring spots with a dark brown border. **Sexual morph**: Not observed. **Asexual morph**: *Conidiomata* solitary, subglobose to globose, unilocular,

dark brown, semi-immersed on leaves. *Conidiophores* indistinct, often reduced to conidiogenous cells. *Conidiogenous cells* subcylindrical or ampulliform, hyaline. *Conidia* 21.0–31.0 × 4.0–7.0 μm, $\bar{x} \pm SD = 26.5 \pm 2.3 \times 6.0 \pm 0.7$ μm (n = 30), L/W ratio = 4.4, fusiform, straight to slightly curved, 4-septate; basal cell obconic with a truncate base, hyaline, 3.0–6.8 μm long ($\bar{x} \pm SD = 5.2 \pm 0.85$ μm); three median cells doliiform to cylindrical, 11.5–18.0 μm long ($\bar{x} \pm SD = 15.5 \pm 1.4$ μm), yellow to brown, concolourous, septa darker than the rest of the cell; second cell from base yellow to brown, 3.5–6.5 μm long ($\bar{x} \pm SD = 5.0 \pm 0.6$ μm); third cell yellow to brown, 3.5–6.5 μm long ($\bar{x} \pm SD = 5.0 \pm 0.64$ μm); fourth cell yellow to brown, 4.0–6.5 μm long ($\bar{x} \pm SD = 5.0 \pm 0.9$ μm); apical cell 2.5–6.0 μm long ($\bar{x} \pm SD = 4.0 \pm 0.7$ μm), hyaline, conic to acute; with 1–4 tubular appendages on apical cell, inserted at different loci in a crest at the apex of the apical cell, unbranched, 13.0–26.0 μm long ($\bar{x} \pm SD = 19.5 \pm 3.2$ μm); single basal appendage, unbranched, tubular, centric, 2.5–7.0 μm long ($\bar{x} \pm SD = 4.5 \pm 1.1$ μm).

Culture characteristics – *Conidia* germinated on PDA within 12 hours at 25 °C from single-spore isolation. Apical cells produced germ tubes. Colony diameter reached 80 mm after two weeks at 25 °C on PDA media, circular, surface rough, flat, white from above, yellow from below.

Material examined – China, Guizhou Province, Qiannan Buyi and Miao Autonomous Prefecture, Libo District, leaf spot of *Smilax china* L.(Liliaceae), 12 March 2022, Y.R. Sun, bb1 (HKAS 124560, holotype); ex-type-living cultures, GUCC 807; China, Guizhou Province, Tongren City, Jiangkou District, Yamugou Parkland, leaf spot of *Lithocarpus* sp.(Fagaceae), 20 May 2022, Y.R. Sun, JK15-2 (HKAS 124565, Paratype); living cultures, GUCC 808; China, Guizhou Province, Guiyang City, Baiyun District, Changpoling National Forest Park, leaf spot of *Dryopteris crassirhizoma* Nakai (Dryopteridaceae), 20 August 2021, Y.R. Sun, CL1-2 (HGUP 22-xxx, dried culture); living cultures, GUCC 813.

Notes – Three strains of *Neopestalotiopsis chinensis* (GUCC 807, GUCC 813 and GUCC 808) have identical ITS, TEF, and TUB sequences to isolates CFCC-54337 and ZX12-1, which were previously provided by Jiang et al. (2021). However, they did not introduce it as a new species due to the lack of neighboring species to compare the morphology. In this study, GUCC 807, GUCC 808 and GUCC 813 have the same morphology. However, they have longer conidia than CFCC-54337 and ZX12-1 (21.0–31.0 × 4.0–7.0 vs. 19.9–23 × 5.8–7.6). In phylogenetic analyses, *N. chinensis* is close to *N. formicarum*, *N. photiniana*, *N. sichuanensi* and *N. vheena*. The PHI test on *N. chinensis* indicated that there is no significant recombination ($\Phi_w = 1.0$) between *N. chinensis* and its closely related taxa. Thus, we introduce *N. chinensis* as a new species and assign GUCC 807 as the holotype, due to CFCC-54337 and ZX12-1 are invalid. *Neopestalotiopsis chinensis* appears to be a common phytopathogen as it has been found in leaf spots on different plants.

Figure 4 – *Neopestalotiopsis chinensis* (HUGP 22-xx, holotype). a Host. b Leaf spot on *Smilax scobinicaulis*. c, d Close up view of conidiomata. e–g Conidia attached to conidiogenous cells. h–k Conidia. Scale bars: e–k = 20 μ m.

Neopestalotiopsis hyperica Y.R. Sun & Yong Wang bis, sp. nov.

Fig. 5

Index Fungorum number: IFXX; Facesoffungi number: FoF XX

Etymology: The specific epithet is referring to *Hypericum*, the host plant from which the fungus was isolated.

Holotype – HKAS 124561

Associated with leaf spots of *Hypericum monogynum*. Symptoms irregular shape, pale to brown, slightly sunken spots appear on the leaves of *Hypericum* sp., which later expand outwards. **Sexual morph**: Not observed. **Asexual morph**: Conidiomata solitary, unilocular, dark. Conidiophores often reduced to conidiogenous cells. Conidiogenous cells indistinct. Conidia 17.0–22.0(–24.2)×5.0–8.2 μ m $\bar{x} \pm SD = 19.0 \pm 1.3 \times 6.8 \pm 0.9 \mu$ m (n = 30), L/W ratio = 2.8, fusoid, sub-cylindrical, straight to

slightly curved, 4-septate; basal cell conic to obconic with a truncate base, hyaline to sub-hyaline, 2.0–4.0 (–4.6) μm long; three median cells 10.5–14.5(–16.5) μm long ($\bar{x} \pm \text{SD} = 12.5 \pm 1.0 \mu\text{m}$), wall rugose, concolorous; second cell from base pale brown to brown, 3.0–4.7 μm long; third cell pale brown to brown, 3.0–5.2(–6.1) μm long; fourth cell pale brown to brown, 2.5–5.5 μm long; apical cell 1.5–3.5 μm long, hyaline, rugose and thin-walled; with 2–3 tubular apical appendages, arising from the apical crest, unbranched, filiform, 11.5–22.5 μm long; single basal appendage 3.7–6.8 μm long, unbranched, tubular, centric.

Culture characteristics – Colonies on PDA reaching up to 10 cm after 2 weeks, dense mycelium on the surface, white from above and below. Fruiting bodies were observed after 14 days.

Material examined – China, Guizhou Province, Guiyang City, Baiyun District, Changpoling National Forest Park, leaf spot of *Hypericum monogynum* L.(Clusiaceae), 20 August 2021, Y.R. Sun, CL5-1 (HKAS 124561, holotype); ex-type-living cultures, GUCC 812; *ibid.*, on leaf spots of *Hypericum monogynum*, 20 August 2021, Y.R. Sun, CL5-1-1 (HGUP 22-xxx), living culture GUCC 811.

Notes – *Neopestalotiopsis hyperica* is related to *N. rhododendri* and *N. protearum* in the phylogenetic analyses (Fig 1), but they can be distinct from concolorous conidia and the size of their median cells (10.5–14.5 μm in *N. hyperica* vs. 13.5–19.5 μm in *N. rhododendri* vs. 16–17 μm in *N. protearum* (Maharachchikumbura et al. 2014, Yang et al. 2021). In addition, there are 13 base pairs differences between *N. hyperica* and *N. rhododendri* in the TEF region. Moreover, the PHI test on *N. hyperica* indicated that there is no significant recombination ($\Phi_w = 1.0$) between *N. hyperica* and its closely related taxa (Fig 2b). Thus, we introduce *N. hyperica* as a new species.

Figure 5 – *Neopestalotiopsis hyperica* (MFLU 22-xx, holotype). a Culture. b Close up view of conidiomata. c–f Conidia. Scale bars: c–f = 10 μm .

Neopestalotiopsis photiniana Y.R. Sun & Yong Wang bis, sp. nov.

Fig. 6

Index Fungorum number: IFXX; Facesoffungi number: FoF XX

Etymology: Referring to the host plant from which the fungus was isolated.

Holotype – MFLU 22-0XXX

Associated with leaf spots of *Photinia serrulata*. *Symptoms* irregular shape, pale to brown, slightly sunken spots appear on the leaves of *Photinia serrulata*, which later expand outwards. Small spots gradually enlarged, changing to brown circular ring spots with a dark brown border. **Sexual morph:** Not observed. **Asexual morph:** *Conidiomata* solitary, subglobose to globose, unilocular, dark brown, semi-immersed on leaves. *Conidiophores* indistinct, often reduced to conidiogenous cells. *Conidiogenous cells* sub-cylindrical, ampulliform, hyaline. *Conidia* 20.0–28.5 \times 5.5–11.5 μm , $\bar{x} \pm \text{SD} = 23.0 \pm 1.9 \times 8.5 \pm 0.89 \mu\text{m}$ (n = 40), L/W ratio = 2.7, broadly fusiform, straight to slightly curved, 4-septate; basal cell obconic with a truncate base, hyaline to pale brown, 1.5–5.5 μm long ($\bar{x} \pm \text{SD} = 3.5 \pm 0.89 \mu\text{m}$); three median cells 13.0–19.5 μm long ($\bar{x} \pm \text{SD} = 15.5 \pm 1.5 \mu\text{m}$), brown to dark, wall rugose, versicolourous; second cell from base pale brown to brown, 3.5–6.5 μm long ($\bar{x} \pm \text{SD} = 4.8 \pm 0.6 \mu\text{m}$); the third and fourth cells dark brown to **dark** are not easily distinguished, septate indistinct, 9.5–13.0 μm long ($\bar{x} \pm \text{SD} = 11.5 \pm 1.0 \mu\text{m}$); apical cell 2.0–4.0 μm long ($\bar{x} \pm \text{SD} = 3.0 \pm 0.5 \mu\text{m}$), hyaline, conic to acute; with 2–3 tubular appendages on apical cell, inserted at different loci in a crest at the apex of the apical cell, unbranched, 17.0–32.5 μm long ($\bar{x} \pm \text{SD} = 24.5 \pm 4.2 \mu\text{m}$); single basal appendage, unbranched, tubular, centric, 1.5–5.5 μm long ($\bar{x} \pm \text{SD} = 3.0 \pm 1.2 \mu\text{m}$).

Culture characteristics – Conidia germinated on PDA within 12 hours at 25 °C from single-spore isolation. Apical cells produced germ tubes. Colony diameter reached 80 mm after three weeks at 25 °C on PDA media, circular, surface rough, flat, white from above and below.

Material examined – China, Guizhou Province, Guiyang City, Nanming District, xiaochuhe road, Guiyang Ahahu National Wetland Park, leaf spots of *Photinia serrulata* Lindl.(Rosaceae), 21 September 2019, Y.R. Sun, AH9 (MFLU 22-xxx, holotype); ex-type-living culture GUCC 810.

Notes – *Neopestalotiopsis photiniana* is phylogenetically related to *N. sichuanensis* and *N. vheena* (Fig. 1). *Neopestalotiopsis photiniana* differs by its thinner conidia (L/W ratio = 2.7 vs. L/W ratio = 4.1) from *N. sichuanensis* (Jiang et al. 2021). *Neopestalotiopsis photiniana* is morphologically indistinguishable from *N. vheena* (Prasannath et al. 2021). However, they have 13 base pairs (without gap, 473 bp) in the TEF region. The result of the PHI test showed there is no obvious recombination ($\Phi_w = 1.0$) between *N. photiniana* and its closely related taxa (Fig 2a). Therefore, *N. photiniana* is introduced as a new species.

Figure 6 – *Neopestalotiopsis photiniana* (MFLU 22-xx, holotype). a Host. b Leaf spot on *Photinia serrulata*. c Close up view of conidiomata. d, e conidia attached to conidiogenous cells. f–i Conidia. Scale bars: d = 1000 μm , c = 200 μm , d = 20 μm , f–i = 10 μm .

Neopestalotiopsis subtropicalis Y.R. Sun & Yong Wang bis, sp. nov.

Fig. 7

Index Fungorum number: IFXX; Facesoffungi number: FoF XX

Etymology: Referring to the subtropical regions in which the collections were encountered.

Holotype – MFLU 22-0XXX

Saprobic on *Ceiba pentandra* leaves and endophytic from *Pinellia ternata*.

Sexual morph: Not observed. **Asexual morph:** *Conidiomata* solitary, unilocular, dark, immersed on stems. *Conidiophores* indistinct, often reduced to conidiogenous cells. *Conidiogenous cells* indistinct. *Conidia* 19–24.5 \times 6.0–8.0 μm $\bar{x} \pm \text{SD} =$

21.5±1.2 × 7.0 ± 0.6 μm (n = 30), L/W ratio = 3.1, fusoid, ellipsoid to subcylindrical, straight to slightly curved, 4-septate; basal cell conic to obconic with a truncate base, hyaline to subhyaline, 3–5.5 μm long; three median cells 13.0–15.2 μm long ($\bar{x} \pm SD = 14.0 \pm 0.6 \mu\text{m}$), wall rugose, versicolourous, septa darker than the rest of the cell; second cell from base pale brown to brown, 3.5–5.0 μm long; third cell brown, 3.5–5.5 μm long; fourth cell brown, 3.0–5.5 μm long; apical cell 2.5–4.0 μm long, hyaline, rugose and thin-walled; with 2 (seldom 3) tubular apical appendages, arising from the apical crest, unbranched, filiform, 11.5–20.0 μm long; single basal appendage 2.0–5.0 μm long, unbranched, tubular, centric.

Culture characteristics – Colonies on PDA reaching up to 8 cm in two weeks, dense aerial mycelium on the surface with undulate edge, white. Fruiting bodies were observed after 14 days.

Material examined – Thailand, Chiang Rai Province, dead leaves of *Ceiba pentandra* (L.) Gaertn. (Bombacaceae), 16 Jan 2020, Y.R. Sun, CR20 (MFLU 22-xxx, holotype); ex-type living cultures, MFLUCC 22-xxx. China, Guizhou Province, Guiyang City, Nanming District, Guiyang Medicinal Botanical Garden, on healthy leaves of *Pinellia ternata* (Thunb.) Breit. (Araceae), 1 May 2022, Y.R. Sun, E2 (HGUP 22-xxxx, paratype); living culture GUCC 805.

Notes – Our isolates GUCC 22-xxxx and MFLUCC 22-xxx clustered together with *Neopestalotiopsis* sp2 (CFCC 54340 and ZX22B) and these four isolates formed a distinct clade in the phylogenetic tree (Fig. 1). Four isolates have similar characteristics. The PHI test on *N. subtropicalis* indicated that there is no significant recombination ($\Phi_w = 1.0$), between *N. subtropicalis* and its closely related taxa (Fig 2c). Hence, we identify *N. subtropicalis* as a new species. Interestingly, GUCC 22-xxxx, MFLUCC 22-xxx, ZX22B and CFCC 54340 have different habitats. GUCC 22-xxxx was endophytic in healthy leaves of *Pinellia ternata*, MFLUCC 22-xxx was saprobic on decaying leaves of *Ceiba pentandra*, and CFCC 54340, ZX22B were isolated from leaf spots of *Castanea mollissima*. This seems to indicate that the same species has different lifestyles in different hosts or regions.

Figure 7 – *Neopestalotiopsis subtropicalis* (MFLU 22-xx, holotype). a, b Conidiomata on the host. c–i Conidia. j Germinated conidium. Scale bars: c–j = 10 μ m.

Neopestalotiopsis suphanburica Y.R. Sun & Yong Wang bis, sp. nov. Fig. 8

Index Fungorum number: IFXX; Facesoffungi number: FoF XX

Etymology – refers to the province where the fungus was collected, Suphan Buri Province.

Holotype – MFLU 22-xxx

Saprobic on stems of an unidentified plant. **Sexual morph:** Not observed. **Asexual morph:** *Conidiomata* solitary, subglobose to globose, unilocular, brown to dark, immersed on stems. *Conidiophores* indistinct, often reduced to conidiogenous cells. *Conidiogenous cells* sub-cylindrical, ampulliform, hyaline. *Conidia* 18.5–29.0 \times 3.7–6.8 μ m, $\bar{x} \pm SD = 24.5 \pm 2.8 \times 5.0 \pm 0.6 \mu$ m (n = 40), L/W ratio = 4.9, fusiform, straight to slightly curved, 4-septate; basal cell obconic with a truncate base, hyaline, smooth-walled, 2.8–6.7 μ m long ($\bar{x} \pm SD = 4.5 \pm 1.0 \mu$ m); three median cells 12.0–19.0 μ m long ($\bar{x} \pm SD = 15.7 \pm 2.0 \mu$ m), pale brown to brown, wall rugose, concolor, septa darker than the rest of the cell, versicolourous; second cell from base pale brown to brown, 3.0–7.3 μ m long ($\bar{x} \pm SD = 5.5 \pm 0.9 \mu$ m); third cell pale brown to brown, 3.3–6.0 μ m long ($\bar{x} \pm SD = 4.6 \pm 0.7 \mu$ m); fourth cell pale brown to brown,

3.5–6.5 μm long ($\bar{x} \pm \text{SD} = 5.0 \pm 0.6 \mu\text{m}$); apical cell 3.0–6.5 μm long ($\bar{x} \pm \text{SD} = 4.2 \pm 0.8 \mu\text{m}$), hyaline, conic to acute; with 2–3 tubular appendages on apical cell, inserted at different loci in a crest at the apex of the apical cell, unbranched, 9.0–21.0 μm long ($\bar{x} \pm \text{SD} = 13.5 \pm 3.5 \mu\text{m}$); single basal appendage, unbranched, tubular, centric, 2.3–11.0 μm long ($\bar{x} \pm \text{SD} = 5.5 \pm 1.9 \mu\text{m}$).

Culture characteristics – Colony diameter reached 8 cm after two weeks at 25 °C on PDA media, circular, surface rough, flat, white from above, white to pale gray from below.

Material examined – Thailand, Suphan Buri Province, dead stem of an unidentified plant, 5 September 2020, S Wang, TN01 (MFLU 22-xxx, holotype); ex-type-living cultures, MFLUCC 22-xxx.

Notes – *Neopestalotiopsis suphanburica* is phylogenetically sister to *N. eucalyptorum* which was isolated from leaves and stems of *Eucalyptus globulus* (Fig. 1). In morphology, *N. suphanburica* differs from *N. eucalyptorum* in having thinner conidia (3.7–6.8 μm vs. 7.6–8.1 μm). In addition, there are 10 base pair differences (without gap, 445 bp) in the TEF region. The PHI test on *N. suphanburica* also indicated there is no significant recombination ($\Phi_w = 1.0$) between *N. suphanburica* and its closely related taxa (Fig 2d). We thus introduce *N. suphanburica* as a new species.

Figure 8 – *Neopestalotiopsis suphanburica* (MFLU 22-xx, holotype). a, b Cultures. c

Colony on culture. d, e Conidia attached to conidiogenous cells, f–i Conidia. Scale bars: d–i = 10 µm.

Neopestalotiopsis rhapsidis Qi Yang & Yong Wang bis, Biodiversity Data Journal 9(no. e70446): 9 (2021) Fig. 9

Index Fungorum number: IF 840065; Facesoffungi number: FoF XX

Associated with leaf spot on leaves tip of *Podocarpus macrophyllus*. *Symptoms* irregular shape, brown, dry, slightly sunken on leaves tip. **Sexual morph:** Not observed. **Asexual morph:** *Conidiomata* solitary, subglobose to globose, unilocular, dark brown to dark, semi-immersed on leaves. *Conidiophores* indistinct, often reduced to conidiogenous cells. *Conidiogenous cells* subcylindrical or ampulliform, short, hyaline. *Conidia* 17.5–25.0 × 5.0–8.0 µm, $\bar{x} \pm SD = 22.0 \pm 1.4 \times 6.5 \pm 0.7$ µm (n = 40), L/W ratio = 3.4, fusiform, straight to slightly curved, 4-septate; basal cell obconic with a truncate base, hyaline, 3.0–6.2 µm long ($\bar{x} \pm SD = 4.3 \pm 0.65$ µm); three median cells doliiiform to cylindrical, 11.5–17.0 µm long ($\bar{x} \pm SD = 14.5 \pm 1.0$ µm), pale to brown, versicolourous, septa darker than the rest of the cell; second cell from base pale to brown, 3.0–6.0 µm long ($\bar{x} \pm SD = 4.5 \pm 0.65$ µm); third cell pale to brown, 3.3–6.5 µm long ($\bar{x} \pm SD = 5.0 \pm 0.6$ µm); fourth cell pale to brown, 2.5–5.5 µm long ($\bar{x} \pm SD = 4.2 \pm 0.7$ µm); apical cell 2.0–5.0 µm long ($\bar{x} \pm SD = 3.0 \pm 0.6$ µm), hyaline, conic to acute; with 2–4 (mostly 3) tubular appendages on apical cell, inserted at different loci in a crest at the apex of the apical cell, unbranched, 12.0–28.0 µm long ($\bar{x} \pm SD = 20.5 \pm 3.4$ µm); single basal appendage, unbranched, tubular, centric, 2.5–7.0 µm long ($\bar{x} \pm SD = 4.8 \pm 1.2$ µm).

Culture characteristics – Conidia germinated on PDA within 12 hours at 25 °C from single-spore isolation. Apical cells produced germ tubes. Colony diameter reached 80 mm after 7 days at 25 °C on PDA media, circular, surface rough, flat, white from above, yellow from below.

Material examined – China, Guizhou Province, Qiannan Buyi and Miao Autonomous Prefecture, Libo District, leaf spots of *Podocarpus macrophyllus* (Thunb.) D. Don (Podocarpaceae), 12 March 2022, Y.R. Sun, ML3 (HKAS 124559); living cultures, GUCC 806.

Notes – *Neopestalotiopsis rhapsidis* was introduced by Yang et al. (2021) from leaf spot of *Rhapis excelsa* (Arecaceae) in China. Our isolate GUCC 806 clustered together with *N. rhapsidis* (GUCC 21501) in the phylogenetic tree. These two species have overlapping conidial measurements (17.5–25.0 × 5.0–8.0 µm for GUCC 806 vs. (22–)25.5 × 4(–6) µm for GUCC 21501) (Yang et al. 2021). Both isolates were associated with leaf spots in China. Therefore, we identify GUCC 806 and GUCC 21501 to be conspecific species, and GUCC 806 represents a new host record.

Figure 9 – *Neopestalotiopsis rhapsidis* (MFLU 22-0XXX, new host record). a, b Host. c Leaf blight on *Podocarpus macrophyllus*. d, e Close up view of conidiomata. f, g Conidia attached to conidiogenous cells. h–k Conidia. Scale bars: f = 10 μ m, g–k = 20 μ m.

Neopestalotiopsis rhododendri Qi Yang & Yong Wang bis, Biodiversity Data Journal 9(no. e70446): 9 (2021) Fig. 10

Index Fungorum number: IF 840066; Facesoffungi number: FoF **XX**

Associated with leaf spots of *Dracaena fragrans*. *Symptoms* subcircular to circular, yellow to brown, slightly sunken spots appear on the leaves of *Dracaena* sp, which later expand outwards. Small auburn spots appeared initially and then gradually enlarged. **Sexual morph**: Not observed. **Asexual morph**: *Conidiomata* on PDA pycnidial, subglobose to cylindrical, solitary or aggregated, dark, semi-immersed or

partly erumpent; exuding black conidial masses. *Conidiophores* indistinct. *Conidiogenous cells* cylindrical to sub-cylindrical or ampulliform to lageniform, hyaline. *Conidia* pale brown to dark brown, fusiform, straight to slightly curved, 4-septate, $19.0\text{--}28.0 \times 5\text{--}7 \mu\text{m}$, $\bar{x} \pm \text{SD} = 24.0 \pm 2.0 \times 6.0 \pm 0.5 \mu\text{m}$ ($n = 40$), L/W ratio = 4; basal cell obconic with a truncate base, hyaline or pale brown, $3.0\text{--}6 \mu\text{m}$ long ($\bar{x} \pm \text{SD} = 4.3 \pm 0.6 \mu\text{m}$, $n = 40$); three median cells $12.0\text{--}18.0\text{--}(18.5) \mu\text{m}$ long ($\bar{x} \pm \text{SD} = 12.8 \pm 1.5 \mu\text{m}$, $n = 40$), pale brown to dark brown, wall rugose, septa darker than the rest of the cell, somewhat constricted at the septa, versicolourous; second cell from base pale brown, $3.5\text{--}6.5 \mu\text{m}$ long ($\bar{x} \pm \text{SD} = 4.8 \pm 0.7 \mu\text{m}$, $n = 40$); third cell brown, $3.0\text{--}6.0 \mu\text{m}$ long ($\bar{x} \pm \text{SD} = 4.5 \pm 0.7 \mu\text{m}$, $n = 40$); fourth cell brown, $3.5\text{--}6 \mu\text{m}$ long ($\bar{x} \pm \text{SD} = 4.5 \pm 0.66 \mu\text{m}$, $n = 40$); apical cell $2.0\text{--}5.5 \mu\text{m}$ long ($\bar{x} \pm \text{SD} = 3.8 \pm 0.7 \mu\text{m}$, $n = 40$), hyaline, conic to acute; with 3–5 tubular appendages on apical cell, inserted at different loci in a crest at the apex of the apical cell, unbranched, $10.0\text{--}26.0 \mu\text{m}$ long ($\bar{x} \pm \text{SD} = 15.5 \pm 4.8 \mu\text{m}$); 1–3 basal appendage, unbranched, tubular, centric, $3.5\text{--}12.0 \mu\text{m}$ long ($\bar{x} \pm \text{SD} = 7.5 \pm 2.6 \mu\text{m}$).

Culture characteristics – Colony diameter reached 8 cm after two weeks at $25 \text{ }^\circ\text{C}$ on PDA medium, circular, surface rough, flat, white from above, yellow from below.

Material examined – Thailand, Chiang Mai Province, Mae Taeng District, Mushroom Research Center, leaf spots of *Dracaena fragrans* (L.) Ker Gawl. (Liliaceae), 15 September 2020, S Wang, LD1 (MFLU 22-xxx, holotype); ex-type-living cultures, MFLUCC 22-xxx.

Notes – *Neopestalotiopsis rhododendri* was introduced by Yang et al. (2021) from the diseased leaf of *Rhododendron simsii* (Ericaceae) in China. Based on our phylogenetic analysis of combined TEF, TUB and ITS sequence data, our isolate MFLUCC 22-xxx clustered with the type species, *N. rhododendri* (GUCC 21504) with maximum support (ML-BS = 100%, BYPP = 1). Our collection also shares similar morphological features with the holotype of *N. rhododendri* (GUCC 21504). Both isolates were associated with leaf spots. Therefore, we identify our collection as *N. rhododendri*, which represents a new host and geographical record.

Figure 10 – *Neopestalotiopsis rhododendri* (MFLU 22-xx, new record). a Host. b Leaf spot of *Dracaena fragrans*. c Cultures. d Colonies on PDA. e–g Conidiogenous cells and developing conidia. h–l Conidia. Scale bars: d = 500 µm, e–l = 20 µm.

Part 2: *Pestalotiopsis*

Phylogenetic analyses

The phylogenetic tree (*Pestalotiopsis*) comprised 112 ingroups and two outgroups, *Neopestalotiopsis protearum* (CBS 114178) and *N. cubana* (CBS 600.96). A total of 1,389 characters including gaps (498 for TEF, and 412 for TUB, 479 for ITS) were included in the phylogenetic analysis. Similar tree topologies were obtained by ML and BYPP methods, and the most likely tree ($-\ln = 66,531.894$) is presented (Figure 11). The phylogenetic tree analyzed 12 *Pestalotiopsis* taxa isolated from medicinal plants and indicated four novel species and three new records of *Pestalotiopsis*.

Figure 11. Cont.

Figure 11 – Maximum likelihood (RAxML) tree for *Pestalotiopsis*, based on analysis of a combined dataset of TEF, TUB and ITS sequence data. The tree is rooted with *Neopestalotiopsis protearum* (CBS 114178) and *N. cubana* (CBS 600.96). Bootstrap support values for ML greater than 75% and Bayesian posterior probabilities greater than 0.95 are given near nodes, respectively. The new isolates are in red.

Genealogical Concordance Phylogenetic Species Recognition Analysis

The PHI test revealed that there is no significant recombination ($\Phi_w = 0.2563$), between *P. chiangmaiensis* and its closely related taxa *P. smilaxe* (MFLUCC 22-xxx), *P. dracontomelon* (MFLUCC 10-0149) and *P. rhizophorae* (MFLUCC 17-0416) (Fig 12a). The *P. loeiensis* based PHI test confirmed that there is no significant recombination ($\Phi_w = 0.1318$), between *P. loeiensis* and its closely related taxa *P. chiangmaiensis* (MFLUCC 22-xxx), *P. nanningensis* (CSUFTCC10), *P. rhizophorae* (MFLUCC 17-0416) and *P. thailandica* (MFLUCC 17-1616) (Fig 12b). The pairwise homoplasy index (PHI) test revealed that there is no significant recombination ($\Phi_w = 5.827$), between *P. peristrophidis* and its closely related taxa *P. biciliate* (CBS 124463), *P. brachiata* (LC2988), and *P. camelliae-oleiferae* (CSUFTCC09) (Fig 12c).

Figure 12 – Split graphs showing the results of PHI test of new *Pestalotiopsis* species with their most closely related species, using Log-Det transformation and split decomposition options. The new taxon in each graph is shown in red font.

Taxonomy

Pestalotiopsis chiangmaiensis Y.R. Sun & Yong Wang bis, sp. nov. Fig. 13

Index Fungorum number: IFXX; Facesoffungi number: FoF XX

Etymology – Refers to the location where the fungus was encountered.

Holotype – MFLU 22-0XXX

Associated with leaf strips of *Phyllostachys edulis*. **Sexual morph:** Not observed.

Asexual morph: *Conidiomata* on PDA pycnidial, subglobose to globose, solitary or aggregated, dark, semi-immersed or partly erumpent; exuding black conidial masses.

Conidiophores hyaline, smooth, simple, reduced to conidiogenous cells. *Conidiogenous cells* 5.0–11.0 × 1.5–3.0 µm, cylindrical to sub-cylindrical or ampulliform to lageniform, hyaline, smooth. *Conidia* pale brown, fusiform, straight to slightly curved, (3)4-septate, 15.5–26.0 × 4.0–6.5 µm, $\bar{x} \pm SD = 21.0 \pm 2.2 \times 5 \pm 0.5$ µm (n = 40), L/W ratio = 4.2; basal cell obconic with a truncate base, hyaline or sometimes pale brown, smooth-walled, (2.5–)3.5–6 µm long ($\bar{x} \pm SD = 4.5 \pm 0.7$ µm); three median cells 10.0–16.0 µm long ($\bar{x} \pm SD = 14.0 \pm 1.4$ µm), pale brown, concolourous, wall rugose, septa darker than the rest of the cell, somewhat constricted at the septa; second cell from base pale brown, 3.0–6.0 µm long ($\bar{x} \pm SD = 4.3 \pm 0.6$ µm); third cell brown, 2.5–6.5 µm long ($\bar{x} \pm SD = 4.0 \pm 0.7$ µm); fourth cell brown, 2.5–5.5 µm long ($\bar{x} \pm SD = 4 \pm 0.7$ µm); apical cell 2.5–5.0 µm long ($\bar{x} \pm SD = 3.6 \pm 0.5$ µm), hyaline, conic to acute; with 2(–3) tubular appendages on apical cell, inserted at different loci in a crest at the apex of the apical cell, unbranched, 8.5–13.0 µm long ($\bar{x} \pm SD = 10.0 \pm 1.7$ µm); single basal appendage, unbranched, tubular, centric, 2.5–6.5 µm long ($\bar{x} \pm SD = 5.0 \pm 1.7$ µm).

Culture characteristics – Colonies on PDA reaching 5–6 cm diam after 7 d at 25 °C, colonies filamentous to circular, medium dense, aerial mycelium on surface flat or raised, with filiform (curle) margin, fluffy, white from above and below; fruiting bodies black.

Material examined – Thailand, Chiang Mai Province, Mae Taeng District, Mushroom Research Center, leaf strip of *Phyllostachys edulis* (Carriere) J. Houzeau (Poaceae), 15 July 2020, Y.R. Sun, M18 (MFLU 22-xxx, holotype); ex-type-living cultures, MFLUCC 22-xxx; *ibid.*, on leaf strip of *Phyllostachys edulis*, 15 July 2020, Y.R. Sun, M18-1 (MFLU 22-xxx), living cultures, MFLUCC 22-xxx.

Notes – *Pestalotiopsis chiangmaiensis* formed a distinct lineage and was sister to *P. smilaxe* (GUCC 22-xxx) and *P. dracontomelon* (MFLUCC 10-0149) in the phylogenetic tree (Fig. 11). It differs by visible, longer conidiophores from *P. smilaxe* and shorter apical appendages (8.5–13 µm vs. 10.0–22 µm) from *P. dracontomelon*. In addition, there are 14 base pair differences (without gap, 474bp) in the TEF region between MFLUCC 22-xxx and MFLUCC 22-M13, and 15 base pair differences (without gap, 464bp) between MFLUCC 22-xxx and MFLUCC 10-0149. The PHI test on *P. chiangmaiensis* also showed that there is no significant recombination ($\Phi_w = 0.2563$), between *P. chiangmaiensis* and its closely related taxa (Fig 12a). Therefore, we introduce *P. chiangmaiensis* as a new species.

Figure 13 – *Pestalotiopsis chiangmaiensis* (MFLU 22-0XXX, holotype). a Host. b, c Cultures. d Colonies on PDA. e–g Conidiogenous cells and developing conidia. h–m Conidia. Scale bars: e–m = 20 μ m.

Pestalotiopsis loeiensis Y.R. Sun & Yong Wang bis, sp. nov. Fig. 14

Index Fungorum number: IFXX; Facesoffungi number: FoF XX

Etymology – Refers to the collected site, Loei province.

Holotype – MFLU 22-0XXX

Saprobic on dead leaves. **Sexual morph:** Not observed. **Asexual morph:** *Conidiomata* solitary, black, semi-immersed on leaves. *Conidiophores* indistinct and *conidiogenous cells* indistinct. *Conidia* 17.0–22.0 \times 4.0–6.0 μ m, $\bar{x} \pm SD = 19.0 \pm 1.2 \times 5.2 \pm 0.5 \mu$ m (n = 40), L/W ratio = 3.7, fusiform, straight to slightly curved, 4-septate; basal cell obconic with a truncate base, hyaline or sometimes pale brown, rugose walled, 2.8–5.5 μ m long ($\bar{x} \pm SD = 3.6 \pm 0.6 \mu$ m), with 1–3 basal appendages, unbranched, tubular, centric, 3.0–13.0 μ m long ($\bar{x} \pm SD = 8.7 \pm 2.8 \mu$ m); three median cells 10.0–14.0 μ m long ($\bar{x} \pm SD = 12.0 \pm 0.8 \mu$ m), doliiform to cylindrical, brown, concolourous, wall rugose, septa darker than the rest of the cell, somewhat constricted at the septa; second cell from base brown, 3.0–5.5 μ m long ($\bar{x} \pm SD = 4.1$

$\pm 0.5 \mu\text{m}$); third cell brown, 3.0–5.0 μm long ($\bar{x} \pm \text{SD} = 3.9 \pm 0.4 \mu\text{m}$); fourth cell brown, 2.0–5.5 μm long ($\bar{x} \pm \text{SD} = 4.0 \pm 0.6 \mu\text{m}$); apical cell 2.8–4.5 μm long ($\bar{x} \pm \text{SD} = 3.8 \pm 0.4 \mu\text{m}$), hyaline, conic to acute; with 1–3 tubular appendages on apical cell, inserted at different loci in a crest at the apex of the apical cell, unbranched, 13–24 μm long ($\bar{x} \pm \text{SD} = 18.0 \pm 2.7 \mu\text{m}$).

Culture characteristics – Colonies on PDA reaching 8 cm diam after two weeks at 25 °C, colonies filamentous to circular, medium dense, mycelium on surface flat or raised, with filiform margin, fluffy, yellow circle in the middle surrounded by white mycelium from above, light yellow to pale brown from the reverse.

Material examined – Thailand, Loei Province, dead leaves of an identified plant, 27 February 2020, J.Y. Zhang, JY1 (MFLU 22-xxx, holotype); ex-type-living cultures, MFLUCC 22-xxx.

Notes –*Pestalotiopsis Loeiensis* (MFLUCC 22-xxx) is phylogenetically related to *P. rhizophorae* and *P. thailandica*, which were isolated from leaf spots of mangroves (Fig. 11). *P. loeiensis* has a similar conidial size to *P. rhizophorae* and *P. thailandica*. However, *P. loeiensis* is distinguishable by its more than one basal appendage. The result of the PHI test ($\Phi_w = 0.2881$) also showed there is no significant recombination between *P. loeiensis* and its closely related taxa (Fig 12b). Therefore, we introduce *P. loeiensis* as a new species.

Figure 14 – *Pestalotiopsis loeiensis* (MFLU 22-xx, holotype). a, b Conidiomata on the host. c–h Conidia. i Germinated conidium. j, k Colonies on PDA. Scale bars: a = 500 μ m, b = 200 μ m, c–i = 10 μ m.

Pestalotiopsis peristrophidis Y.R. Sun & Yong Wang bis, sp. nov.

Fig. 15

Index Fungorum number: IFXX; Facesoffungi number: FoF XX

Etymology – Name reflects the host's name, *Peristrophe japonica*

Holotype – HUGP 801

Endophytic from *Peristrophe japonica*. Colonies on PDA white with grey mycelia in the center, felty, black conidiomata, lobed edge; reverse white with pale to brown concentric circles. **Sexual morph:** Not observed. **Asexual morph:** *Conidiomata* on PDA pycnidial, solitary or aggregated, subglobose to globose, dark, semi-immersed or erumpent; exuding black conidial masses. *Conidiophores* hyaline, smooth, simple, reduced to conidiogenous cells. *Conidiogenous cells* 1.2–3.0 μ m wide, cylindrical to sub-cylindrical or ampulliform to lageniform, hyaline, smooth. *Conidia* pale to brown, fusiform, straight to slightly curved, 4-septate, 20.0–29.0 \times

4.5–6.2 μm , $\bar{x} \pm \text{SD} = 24.5 \pm 2.0 \times 5.2 \pm 0.4 \mu\text{m}$ ($n = 40$), L/W ratio = 4.7; basal cell obconic with a truncate base, hyaline, smooth walled, 4.5–7.0 μm long ($\bar{x} \pm \text{SD} = 5.5 \pm 0.6 \mu\text{m}$); three median cells 12.0–17.0 μm long ($\bar{x} \pm \text{SD} = 14.0 \pm 1.1 \mu\text{m}$), pale to brown, wall rugose, septa darker than the rest of the cell, somewhat constricted at the septa; second cell from base pale to brown, 3.5–6.0 μm long ($\bar{x} \pm \text{SD} = 5.0 \pm 0.4 \mu\text{m}$); third cell pale to brown, 3.5–6.0 μm long ($\bar{x} \pm \text{SD} = 4.5 \pm 0.4 \mu\text{m}$); fourth cell pale to brown, 3.5–6.0 μm long ($\bar{x} \pm \text{SD} = 4.5 \pm 0.5 \mu\text{m}$); apical cell 2.7–5.7 μm long ($\bar{x} \pm \text{SD} = 4.0 \pm 0.6 \mu\text{m}$), hyaline, conic to acute; with 1–4 (mostly 2) tubular appendages on apical cell, inserted at different loci in a crest at the apex of the apical cell, unbranched, 7.5–17.5 μm long ($\bar{x} \pm \text{SD} = 12.2 \pm 2.5 \mu\text{m}$); single basal appendage, unbranched, tubular, centric, 2.5–5.0 μm long ($\bar{x} \pm \text{SD} = 3.8 \pm 0.5 \mu\text{m}$).

Material examined – China, Guizhou Province, Guiyang City, Nanming District, Guiyang Medicinal Botanical Garden, on healthy leaves of *Peristrophe japonica* (Thunb.) Bremek. (Acanthaceae), 1 May 2022, Y.R. Sun, E53 (HUGP 801, died culture, holotype); ex-type living culture GUCC 803; China, Guizhou Province, Guiyang City, Nanming District, Guiyang Medicinal Botanical Garden, on healthy stems of *Peristrophe japonica*, 1 May 2022, Y.R. Sun, E56 (HUGP 802, dried culture, paratype; living culture GUCC 801).

Notes – An endophyte, *P. peristrophidis* was isolated from the medicinal plant, *Peristrophe japonica* in China. Our four strains GUCC 801, GUCC 802, GUCC 803 and GUCC 804 clustered together and formed a distinct inner clade, which is sister to *P. brachiata* (LC 2988 and LC 8189) with strong support (ML-BS = 99%, BYBB = 1.00) in the phylogenetic tree (Fig 12). In morphology, *P. peristrophidis* differs from *P. brachiata* by the numbers and size of the basal appendages (1 basal appendage, unbranched, 2.5–5.0 μm long in *P. peristrophidis* vs. 1–4 basal appendages, branched or unbranched, 5.5–9.5 μm long in *P. brachiata*) (Liu et al. 2017). The PHI test on *P. peristrophidis* revealed there is no significant recombination ($\Phi_w = 0.05827$) between *P. peristrophidis* and its closely related taxa (Fig 12c). Therefore, we introduce *P. peristrophidis* as a new species.

Figure 15 – *Pestalotiopsis peristrophidis* (HUGP 22-xx, Holotype). a Host. b Close up view of conidiomata, c, d Conidia attached to conidiogenous cells, e–i Conidia. Scale bars: c–i = 10 μ m.

Pestalotiopsis smilaxe Y.R. Sun & Yong Wang bis, sp. nov.

Fig. 16

Index Fungorum number: IFXX; Facesoffungi number: FoF XX

Etymology – Refers to the host plant from which the fungus was first isolated.

Holotype – MFLU 22-0XXX

Associated with leaf spots of *Smilax china* and *Dioscorea sp.*. Symptoms subcircular to irregular shape, brown, slightly sunken spots appear on the leaves of *Smilax china*, which later expand outwards. Small auburn spots appeared initially and then gradually enlarged. **Sexual morph:** Not observed. **Asexual morph:** *Conidiomata* solitary, subglobose, unilocular, black, semi-immersed on leaves. *Conidiomatal wall* 7.0–10.0 μ m wide, thin walled, pale brown. *Conidiophores* indistinct. *Conidiogenous cells* subcylindrical to ampulliform, hyaline, smooth, 1.4–2.6 μ m wide. *Conidia* 18.0–21.5 \times 4.5–6.5 μ m, $\bar{x} \pm SD = 20 \pm 0.94 \times 5.3 \pm 0.6 \mu$ m (n = 40), L/W ratio =

3.8, fusiform, straight to slightly curved, 4-septate; basal cell obconic with a truncate base, hyaline or sometimes pale brown, smooth walled, 2.8–5.3 μm long ($\bar{x} \pm \text{SD} = 4.0 \pm 0.6 \mu\text{m}$); three median cells 9.5–14.5 μm long ($\bar{x} \pm \text{SD} = 12.0 \pm 0.9 \mu\text{m}$), pale brown to brown, concolourous, wall rugose, septa darker than the rest of the cell, somewhat constricted at the septa; second cell from base pale brown to brown, 2.5–4.5 μm long ($\bar{x} \pm \text{SD} = 3.8 \pm 0.5 \mu\text{m}$); third cell brown, 2.5–5.5 μm long ($\bar{x} \pm \text{SD} = 4 \pm 0.5 \mu\text{m}$); fourth cell brown, 3.0–5.5 μm long ($\bar{x} \pm \text{SD} = 4 \pm 0.5 \mu\text{m}$); apical cell 2.5–4.5 μm long ($\bar{x} \pm \text{SD} = 3.3 \pm 0.4 \mu\text{m}$), hyaline, conic to acute; with 2–3 tubular appendages on apical cell, inserted at different loci in a crest at the apex of the apical cell, unbranched, 6–14 μm long ($\bar{x} \pm \text{SD} = 10.0 \pm 2.0 \mu\text{m}$); single basal appendage, unbranched, tubular, centric, (1.5–)2–5(–6) μm long ($\bar{x} \pm \text{SD} = 3.0 \pm 0.7 \mu\text{m}$).

Culture characteristics – Colonies on PDA reaching 10 cm diam after two weeks at 25 °C, colonies filamentous to circular, medium dense, aerial mycelium on surface flat or raised, with filiform margin, fluffy, white from above and reverse.

Material examined – Thailand, Chiang Mai Province, Mae Taeng District, Mushroom Research Center, leaf spots of *Smilax china* L. (Liliaceae), 15 July 2020, Y.R. Sun, M13 (MFLU 22-xxx, holotype); ex-type-living cultures, MFLUCC 22-xxx. Thailand, Chiang Mai Province, Mae Taeng District, Mushroom Research Center, leaf spots of *Dioscorea sp.* (Dioscoreaceae), 16 July 2020, Y.R. Sun, M26 (MFLU 22-xxx, paratype), living cultures, MFLUCC 22-xxx.

Notes – Two collections MFLU 22-xxx and MFLU 22-xxx share similar morphology. These two isolates clustered together and formed a sister clade to *P. dracontomelon* (MFLUCC 10-0149) in the phylogenetic tree. There are only 1 base pair difference in TEF and TUB genes and 3 base pair differences in the ITS gene between these two isolates. For the differences between *P. smilaxe* and its related species see the notes of *P. chiangmaiensis* (this study). Therefore, these two isolates are identified as conspecific, representing a new species.

Figure 16 – *Pestalotiopsis smilaxe* (MFLU 22-0XXX, holotype). a Host. b, c Close up view of conidiomata. d Section through conidioma. e Section through pycnidial wall. f–h Immature conidia attached to conidiogenous cells. i–m Conidia. n Germinated conidium. Scale bars: b = 1000 μ m, c = 200 μ m, d = 50 μ m, e = 20 μ m, f–n = 10 μ m.

Pestalotiopsis chamaeropsis Maharachch., K.D. Hyde & Crous, *Studies in Mycology*. 79: 158 (2014) Fig.17

Index Fungorum number: IF809735; Facesoffungi number: FoFxx

Endophytic from *Peristrophe japonica*. Colonies on PDA white with grey mycelia in the center, felty, black conidiomata, lobed edge. **Sexual morph:** Not observed. **Asexual morph:** *Conidiomata* on PDA pycnidial, solitary or aggregated, subglobose to globose, dark, semi-immersed or erumpent; exuding black conidial masses. *Conidiophores* hyaline, smooth, simple, reduced to conidiogenous cells. *Conidiogenous cells* 1.2–2.0 μm wide, cylindrical to sub-cylindrical, hyaline, smooth. *Conidia* brown, fusiform, straight to slightly curved, 4-septate, 20.0–27.5 \times 3.7–5.8 μm , $\bar{x} \pm \text{SD} = 24.5 \pm 1.3 \times 5.0 \pm 0.4 \mu\text{m}$ ($n = 40$), L/W ratio = 4.9; basal cell obconic with a truncate base, hyaline to pale brown, smooth walled, 3.2–6.0 μm long ($\bar{x} \pm \text{SD} = 4.8 \pm 0.5 \mu\text{m}$); three median cells 12.5–17.5 μm long ($\bar{x} \pm \text{SD} = 14.5 \pm 0.9 \mu\text{m}$), brown, septa darker than the rest of the cell, somewhat constricted at the septa; second cell from base brown, 4.0–5.8 μm long ($\bar{x} \pm \text{SD} = 5.0 \pm 0.4 \mu\text{m}$); third cell brown, 3.5–6.0 μm long ($\bar{x} \pm \text{SD} = 4.7 \pm 0.4 \mu\text{m}$); fourth cell brown, 4.0–5.8 μm long ($\bar{x} \pm \text{SD} = 4.7 \pm 0.4 \mu\text{m}$); apical cell 3.0–5.4 μm long ($\bar{x} \pm \text{SD} = 4.1 \pm 0.5 \mu\text{m}$), hyaline, conic to acute; with 2–3 tubular appendages on apical cell, inserted at different loci in a crest at the apex of the apical cell, unbranched, 8.5–18.0 μm long ($\bar{x} \pm \text{SD} = 13.8 \pm 2.1 \mu\text{m}$); single basal appendage, unbranched, tubular, centric, 4.0–9.7 μm long ($\bar{x} \pm \text{SD} = 5.6 \pm 0.7 \mu\text{m}$).

Material examined – China, Guizhou Province, Guiyang City, Nanming District, Guiyang Medicinal Botanical Garden, on healthy leaves of *Peristrophe japonica* (Thunb.) Bremek. (Acanthaceae), 1 May 2022, Y.R. Sun, E33 (HUGP 803, dried culture; living culture GUCC 800).

Notes – *Pestalotiopsis chamaeropsis* was originally reported on leaves of *Chamaerops humilis* in Italy by Maharachchikumbura et al. (2014). Subsequently, many studies have proven that *P. chamaeropsis* is a serious phytopathogen, which can cause diseases of *Camellia sinensis*, *Camellia oleifera* and *Eurya nitida* (in China), *Erica arborea* (in Tunisia), *Japanese andromeda* (in Japan) and *Prostanthera rotundifolia* (in Australia) (Moslemi and Taylor 2015, Jiang et al. 2017, Ariyawansa and Hyde 2018, Hlaiem et al. 2018, Nozawa et al. 2019, Wang et al. 2019, Chen et al. 2021, Qiu et al. 2022, Santos et al. 2022). Park et al. (2017) reported *P. chamaeropsis* as an endophyte, from the leaves of woody plants in Korea. In this study, our strain GUCC 800 is phylogenetically clustered with *P. chamaeropsis* CBS 186.71 with maximum support (ML-BS = 100%, BYPP = 1), and it has overlapping characteristics with *P. chamaeropsis* (CBS 186.71). Thus, we identify GUCC 800 as *P. chamaeropsis*, representing a new host record.

Figure 17 – *Pestalotiopsis chamaerapis* (HUGP 803, new record). a, b Host. c Close up view of conidiomata, d–f Conidia attached to conidiogenous cells, g–k Conidia. Scale bars: d–k = 10 μ m.

Pestalotiopsis dracontomelon Maharachch & K.D. Hyde, Fungal Diversity. 15: (2015)
Fig.18

Index Fungorum number: IF550943; Facesoffungi number: FoF00457

Associated with dead leaves of *Podocarpus* sp.. **Sexual morph:** Not observed. **Asexual morph:** *Conidiomata* solitary, subglobose, unilocular, black, semi-immersed on leaves. *Conidiophores* indistinct, often reduced to conidiogenous cells. *Conidiogenous cells* sub-cylindrical to cylindrical, hyaline, rugose walled, 1.5–3.0 μ m wide. *Conidia* 19.0–26.0 \times 5.5–8.0 μ m, \bar{x} \pm SD = 23.0 \pm 1.7 \times 7.0 \pm 0.6 μ m (n = 30), L/W ratio = 3.3, fusiform, ellipsoid, straight to slightly curved, 4-septate; basal cell

obconic with a truncate base, hyaline or sometimes pale brown, rugose walled, 3.3–5.0 μm long ($\bar{x} \pm \text{SD} = 4.0 \pm 0.5 \mu\text{m}$); three median cells 11.0–16.5 μm long ($\bar{x} \pm \text{SD} = 14.5 \pm 1.6 \mu\text{m}$), doliiform to cylindrical, pale brown to brown, concolourous, wall rugose, septa darker than the rest of the cell, somewhat constricted at the septa; second cell from base pale brown to brown, 4.0–6.0 μm long ($\bar{x} \pm \text{SD} = 5.0 \pm 0.6 \mu\text{m}$); third cell pale brown to brown, 3.0–5.5 μm long ($\bar{x} \pm \text{SD} = 4.5 \pm 0.6 \mu\text{m}$); fourth cell pale brown to brown, 3.5–5.5 μm long ($\bar{x} \pm \text{SD} = 4.5 \pm 0.5 \mu\text{m}$); apical cell 3.0–5.5 μm long ($\bar{x} \pm \text{SD} = 4.3 \pm 0.6 \mu\text{m}$), hyaline, conic to acute; with 2–4 tubular appendages on apical cell, inserted at different loci in a crest at the apex of the apical cell, unbranched, 10.0–22 μm long ($\bar{x} \pm \text{SD} = 13.5 \pm 4 \mu\text{m}$); single basal appendage, unbranched, tubular, centric, 2.0–6.0 μm long ($\bar{x} \pm \text{SD} = 3.8 \pm 1.3 \mu\text{m}$).

Culture characteristics – Colonies on PDA reaching 2.5 cm diam after 7 days at 25 °C, colonies filamentous to circular, medium dense, aerial mycelium on the surface flat or raised, fluffy, white from above and below.

Material examined – Thailand, Chiang Rai Province, Mae Fah Luang University, leaf spots of *Podocarpus sp.* (Podocarpaceae), 15 January 2019, Y.R. Sun, S18 (MFLU 22-xxx); living cultures, MFLUCC 22-xxx.

Notes – *Pestalotiopsis dracontomelon* was isolated from diseased leaves of *Dracontomelon mangifera* (Anacardiaceae) in Thailand (Liu et al. 2015). Our isolate MFLUCC 22-xxx was grouped with *P. dracontomelon* (MFLUCC 10-0149) in the phylogenetic tree. Morphologically, they have overlapping conidial measurements (19–26 \times 5.5–8 μm for MFLUCC 22-xxx vs. 18–23 \times 5.5–7.5 μm for MFLUCC 10-0149). Therefore, we identify MFLUCC 22-xxx as the new host record of *P. dracontomelon*.

Figure 18 – *Pestalotiopsis dracontomelon* (MFLU 22-xx, new host record). a Host. b, c Close up view of conidiomata, d Section through conidioma, e Conidia attached to conidiogenous cells, f–j Conidia, k Germinated conidium. l Colony. Scale bars: b = 1000 μ m, d = 50 μ m, e, k = 20 μ m, f–j = 10 μ m.

Pestalotiopsis hydei Huanraluek & Jayaward, Phytotaxa. 479(1): 35 (2021) Fig. 19

Index Fungorum number: IF558100; Facesoffungi number: FoF 09460

Saprobic on dead twigs. **Sexual morph:** Not observed. **Asexual morph:** *Conidiomata* solitary, unilocular, dark, immersed on stems. *Conidiophores* hyaline, smooth, simple, reduced to conidiogenous cells. *Conidiogenous cells* cylindrical to subcylindrical, hyaline, 1.8–2.5 μm wide. *Conidia* 19.5–26.0 \times 4.5–5.8 μm $\bar{x} \pm \text{SD} = 22.5 \pm 1.1 \times 5.0 \pm 0.3 \mu\text{m}$ (n = 40), L/W ratio = 4.5, fusiform, straight to slightly curved, 4-septate; basal cell conic to obconic with a truncate base, hyaline to subhyaline, 2.8–6.0 μm long; three median cells 11.5–16.0 μm long ($\bar{x} \pm \text{SD} = 13.5 \pm 0.7 \mu\text{m}$), wall rugose, pale to brown, concolourous, septa darker than the rest of the cell; second cell from base pale to brown, 3.2–5.5 μm long; third cell pale to brown, 3.0–5.0 μm long; fourth cell pale to brown, 4.0–5.5 μm long; apical cell 3.0–5.0 μm long, hyaline, rugose and thin-walled; with 1–3 tubular apical appendages, arising from the apical crest, unbranched, filiform, 7.0–17.5 μm long; single basal appendage 3.0–8.5 μm long, unbranched, tubular, centric.

Culture characteristics – Colonies on PDA reaching 8 cm diam after 2 weeks at 25 °C, colonies filamentous to circular, medium dense, aerial mycelium on the surface flat or raised, fluffy, white from above and reverse.

Material examined – China, Guizhou Province, Qiannan Buyi and Miao Autonomous Prefecture, Libo District, on dead twigs, 12 March 2022, J.E. Sun, L19-1 (HGUP 804); living cultures, GUCC 819.

Notes – *Pestalotiopsis hydei* was isolated from the leaf spots of *Litsea petiolate* in Thailand (Huanaluek et al. 2021). Our collection HGUP 804 clustered together with *P. hydei* (MFLUCC 20-0135) in the phylogenetic tree. HGUP 804 also has a similar conidial measurement to MFLUCC 20-0135 (19.5–26.0 \times 4.5–5.8 μm in HGUP 804, 18–35 \times 3–6 μm in MFLUCC 20-0135). Besides, there are only 3 base pair differences in the ITS and TEF gene. Therefore, we identify GUCC 819 as a new geographical record of *P. hydei*.

Figure 19 – *Pestalotiopsis hydei* (HGUP 804, new record). a Host. b Close up view of conidiomata, c, d Conidiogenous cells, e–i Conidia. Scale bars: c–i = 10 μ m.

Part3: *Pseudopestalotiopsis*

Phylogenetic analyses

The sequence datasets for TEF, TUB and ITS, were analyzed in combination to infer the interspecific relationships within *Pseudopestalotiopsis*. The aligned sequence matrix consisted of 27 sequences, including two outgroups *Pestalotiopsis trachycarpicola* (IFRDCC 2240) and *P. linearis* (MFLUCC 12-0271). Similar tree topologies were obtained by ML and BI methods, and the most likely tree ($-\ln = 66,531.894$) is presented (Figure 20). Our collection is clustered with the type species of *Pseudopestalotiopsis*, *P. theae*, in the phylogenetic tree (Fig 20).

Fig. 20 Maximum likelihood (rAxML) tree for *Pseudopezalotiopsis*, based on analysis of a combined dataset of TEF, TUB and ITS sequence data. The tree is rooted with *P. trachycarpicola* (IFRDCC 2240) and *P. linearis* (MFLUCC 12-0271). Bootstrap support values for ML greater than 75% and Bayesian posterior probabilities greater than 0.95 are given near nodes, respectively. The new isolates are in red.

Taxonomy

Pseudopestalotiopsis theae (Sawada) Maharachch., K.D. Hyde & Crous, Studies in Mycology. 79: 183 (2014) Fig. 21

Index Fungorum number: IF 631991; Facesoffungi number: FoF XX

Associated with dead leaves of *Ceriops tagal*. **Sexual morph:** Not observed. **Asexual morph:** *Conidiomata* solitary, subglobose, unilocular or bilocular, brown, semi-immersed on leaves. *Conidiophores* indistinct, often reduced to conidiogenous cells. *Conidiogenous cells* sub-cylindrical to cylindrical, hyaline, rugose walled. *Conidia* 22.0–31.0 × 5.0–7.0 μm, $\bar{x} \pm SD = 26.0 \pm 1.9 \times 6 \pm 0.5$ μm (n = 40), L/W ratio = 4.3, fusiform, straight to slightly curved, 4-septate; basal cell obconic with a truncate base, hyaline or sometimes pale brown, rugose walled, 3.0–7.0 μm long ($\bar{x} \pm SD = 5.2 \pm 0.7$ μm); three median cells 14.0–20.0 μm long ($\bar{x} \pm SD = 15.8 \pm 1.3$ μm), doliiform to cylindrical, pale brown to brown, concolourous, wall rugose, septa darker than the rest of the cell; second cell from base pale brown to brown, 4.0–7.5 μm long ($\bar{x} \pm SD = 5.5 \pm 0.7$ μm); third cell pale brown to brown, 3.5–6.5 μm long ($\bar{x} \pm SD = 5.0 \pm 0.7$ μm); fourth cell pale brown to brown, 2.5–7.0 μm long ($\bar{x} \pm SD = 5.0 \pm 0.9$ μm); apical cell 2.5–5.5 μm long ($\bar{x} \pm SD = 4.0 \pm 0.7$ μm), hyaline, conic to acute; with 2–3 tubular appendages on apical cell, inserted at different loci in a crest at the apex of the apical cell, unbranched, 18.0–29.0 μm long ($\bar{x} \pm SD = 24.5 \pm 2.7$ μm); single basal appendage, unbranched, tubular, centric, 3.0–6.0 μm long ($\bar{x} \pm SD = 4.5 \pm 0.6$ μm).

Culture characteristics – Colonies on PDA reaching 10 cm diam after two weeks at 25 °C, colonies filamentous to circular, medium dense, aerial mycelium on the surface flat or raised, fluffy, white from above and below.

Material examined – Thailand, Suphan Buri Province, dead leaf of *Ceriops tagal* (perr.) C. B. Rob. (Rhizophoraceae), 5 September 2020, S Wang, TN07 (MFLU 22-xxx); living cultures, MFLUCC 22-xxx.

Notes – Our isolate MFLUCC 22-xxx is phylogenetically grouped with the type species of *Pseudopestalotiopsis*, *Ps. theae*. Morphologically, our new collection MFLU 22-xx resembles *Ps. theae* (MFLUCC 12-0055) in color and size of the conidiogenous cells, conidia and appendages. Therefore, we report this isolate as a new host record of *Ps. theae* from the *Ceriops tagal*.

Figure 21 – *Pseudopestalotiopsis theae* (MFLU 22-0xx). a Host. b, c Close up view of conidiomata. d, e Section through conidioma. e Section through pycnidial wall. f, h Conidia attached to conidiogenous cells. h–l Conidia. Scale bars: d, e = 100 μ m, f–h = 10 μ m, i–l = 20 μ m.

Discussion

During the research of microfungi on medicinal plants in southwest China and Thailand, 27 pestalotioid strains representing 17 species were isolated from 16 medicinal plants. Six new *Neopestalotiopsis* species, namely, *N. amomumica*, *N. ceibania*, *N. chinensis*, *N. photiniana*, *N. suphanburica* and *N. hyperica*, four new *Pestalotiopsis* species, namely, *P. Chiangmaiensis*, *P. loeiensis*, *P. smilaxe*, and *P. peristrophidis*, and five new records are introduced with descriptions and illustrations. Among them, ten species are related to leaf diseases of medicinal plants, three species are saprobes, two species are endophytes, and one species has various lifestyles. A worldwide checklist of pestalotioid species associated with medicinal plants is provided (Table 4). Among them, most are related to diseases of medicinal plants, and 51 (a total of 87) were found in China and Thailand. By analysis, the majority of pestalotioid species are associated with dicotyledonous medicinal plants (Fig 22).

In this study, *N. subtropicalis* was isolated from healthy leaves of *Pinellia ternata* and dead leaves of *Ceiba pentandra*, and it was previously isolated from the leaf spots of *Castanea mollissima* (Jiang et al. 2021). *Pestalotiopsis chamaeropsis* was found as an endophyte from an important medicinal plant here. However, it was reported as a serious phytopathogen in different countries (Moslemi and Taylor 2015, Jiang et al. 2017, Ariyawansa and Hyde 2018, Nozawa et al. 2019, Wang et al. 2019, Chen et al. 2021, Qiu et al. 2022, Santos et al. 2022). *Pestalotiopsis photiniae* was reported as an endophytic fungus on the branches of *Roystonea regia*, and it also has been isolated from the disease leaves of blueberry (Chen et al. 2011, Ding et al. 2012). These seem to indicate that one fungus can be endophytic, pathogenic, or saprophytic in different (or the same) plants or organs. Understanding the external factors that influence the fungal lifestyle, could have major implications for agriculture, industrial development and human health.

It is worth noting that three *Neopestalotiopsis* species, *N. amomumica*, *N. chinensis* and *N. hyperica* introduced in this paper do not match the characteristics of versicolourous median cells described in *Neopestalotiopsis* (Maharachchikumbura et al. 2014). In addition, *N. chinensis* also has obvious conidiophores which do not match the characteristics of indistinct conidiophores described in *Neopestalotiopsis*. This phenomenon was also mentioned by Liu et al. (2017). The reason for this phenomenon probably could be the type of medium or the time of cultivation. However, there is not enough data to explain this phenomenon.

Neopestalotiopsis and *Pseudopestalotiopsis* were separated from *Pestalotiopsis* by Maharachchikumbura et al. (2014) based on phylogenetic analyses and conidial color. After that, the number of species in *Neopestalotiopsis* increased from 24 to 70 and the number of species in *Pseudopestalotiopsis* increased from three to 23 (<http://www.indexfungorum.org/>, 28 May 2022). In recent years, there have been more reports on diseases caused by *Neopestalotiopsis*, for example; *N. vitis* caused grapevine leaf spots in China, *N. rosicola* caused stem canker of *Rosa chinensis* in China, *N. clavispora* caused leaf spots and fruit rot of strawberry in India, *N. maddoxii* caused flower diseases of *Macadamia integrifolia* in Australia, *N. eucalyptorum* was associated with disease of *Eucalyptus* plant in Portugal

(Jayawardena et al. 2016, Jiang et al. 2018, Mahapatra et al. 2018, Diogo et al. 2021, Prasannath et al. 2021). However, only six *Pseudopestalotiopsis* species were reported as plant pathogens (Maharachchikumbura et al. 2016a, Nozawa et al. 2017, Nozawa et al. 2018, Norphanphoun et al. 2019, Gualberto et al. 2021). In this study, there are nine *Neopestalotiopsis* species and only one *Pseudopestalotiopsis* species was encountered. This seems to indicate that *Neopestalotiopsis* has richer species diversity, and the *Neopestalotiopsis* species are more likely to infect the plant and cause disease than the *Pseudopestalotiopsis*. By comparing the differences between the two genera through whole-genome sequence analysis, and finding related disease-causing genes probably explain this phenomenon.

The interspecific morphological differences of pestalotioid species have been unclear. In their attempt to find a reliable criterion for interspecific differences, taxonomists have complicated the description of pestalotioid species (Nag Raj 1993, Wei and Xu 2004, Maharachchikumbura et al. 2014, Huanaluek et al. 2021). In the past, the conidia were divided into apical cells, basal cells and the three median cells, describing them. The three median cells were divided into the second cells from the base, the third cells, and the fourth cells. And every cell is measured. Obviously, the length of the three median cells is the sum length of the second, the third, and the fourth cells. Therefore, the descriptions are repeated. In addition, the characteristics of each cell were not treated as a criterion for interspecies differences (Nag Raj 1993, Maharachchikumbura et al. 2012, Jayawardena et al. 2016, Liu et al. 2017). Therefore, we suggest removing the measurement description for each cell.

Figure 22 – Distribution of pestalotioid species in different medicinal plants

Table 4 Checklist of pestalotioid species associated with medicinal plants.

Species	Life mode	Disease (if any)	Host	Location	Reference
Neopestalotiopsis acrostichi	P	Leaf spot	Acrostichum aureum (F)	Thailand	Norphanphoun et al. (2019)
Neopestalotiopsis alpapicalis	E		Rhizophora mucronate (D)	Thailand	Kumar et al. (2019)
Neopestalotiopsis amomumica	P	Leaf spot	Amomum villosum (M)	China	This study
Neopestalotiopsis brachiata	P	Leaf spot	Rhizophora apiculata (D)	Thailand	Norphanphoun et al. (2019)
Neopestalotiopsis chinensis	P	Leaf spot	Smilax china , Cyrtomium fortune (M/F)	China	This study Cao et al. (2022), Li et al. (2022)
Neopestalotiopsis clavispora	P	Leaf spot, branch blight	Dendrobium officinale , Taxus×media (M/G)	China	(2022)
Neopestalotiopsis cubana	P	Leaf blight	Ixora chinensis (D)	Malaysia	Khoo et al. (2022)
Neopestalotiopsis dendrobii	E		Dendrobium cariniferum (M)	Thailand	Ma et al. (2019) Maharachchikumbura et al. (2014)
Neopestalotiopsis ellipsospor	Unknown		Ardisia crenata (D)	Hong Kong, China	(2014) Maharachchikumbura et al. (2014)
Neopestalotiopsis eucalypticola	Unknown		Eucalyptus globulus (D)	Unknown	(2014)
Neopestalotiopsis eucalyptorum	P	Leaf necrosis, stem basal cankers	Eucalyptus globulus (D)	Fundão/Guarda/Portugal	Diogo et al. (2021)
Neopestalotiopsis haikouensis	P	Leaf spot	Ilex chinensis (D)	China	Zhang et al. (2022)
Neopestalotiopsis hispanica	P	Leaves and stem necrosis	Eucalyptus globulus (D)	Fundão/Guarda/Spain	Diogo et al. (2021)
Neopestalotiopsis hydeana	P	Leaf spot	Alpinia malaccensis (M)	Thailand	Huanaluek et al. (2021)
Neopestalotiopsis hyperica	P	Leaf spot	Hypericum monogynum (D)	China	This study
Neopestalotiopsis iberica	P	Leaves and stem necrosis	Eucalyptus globulus (D) Eucalyptus globulus /	Pegões/Portugal/Spain	Diogo et al. (2021)
Neopestalotiopsis longiappendiculata	P	Leaves and stem necrosis	E. nitens (D)	Furadouro/Portugal	Diogo et al. (2021)

Neopestalotiopsis lusitanica	P	Leaves and stem necrosis	Eucalyptus globulus (D)	Pegões/Portugal	Diogo et al. (2022)
Neopestalotiopsis pernambucana	Unknown		Vismia guianensis (D)	Brazil	Silvério et al. (2016)
Neopestalotiopsis petila	P	Leaf spot	Rhizophora mucronate (D)	Thailand	Norphanphoun et al. (2019)
Neopestalotiopsis photiniana	P	Leaf spot	Photinia serrulate (D)	China	This study
Neopestalotiopsis rhapsidis	P	Leaf spot	Podocarpus macrophyllus (G)	China	This study
Neopestalotiopsis rhizophorae	P	Leaf spot	Rhizophora mucronate (D)	Thailand	Norphanphoun et al. (2019)
Neopestalotiopsis rhododendri	P	Leaf spot	Dracaena fragrans (M)	Thailand	This study Maharachchikumbura et al. (2014)
Neopestalotiopsis rosae	Unknown		Paeonia suffruticosa (D)	U.S.A	(2014)
Neopestalotiopsis rosicola	P*	Stem canker	Rosa chinensis (D)	China	Jiang et al. (2018) Maharachchikumbura et al. (2014)
Neopestalotiopsis saprophytica	S		Litsea rotundifolia (D)	Hong Kong, China	(2014)
Neopestalotiopsis subtropicalis	E, S		Ceiba pentandra , Pinellia ternate (D/M)	China	This study
Neopestalotiopsis surinamensis	E		Scurrula atropurpurea (D)	Indonesia	Elfita et al. (2020 a,b)
Neopestalotiopsis thailandica	P	Leaf spot	Rhizophora mucronate (D) Clerodendrum canescens/Sinopodophyllum	Thailand	Norphanphoun et al. (2019) Xu et al. (2016), Xiao et al. (2018), Yan et al. (2019)
Pestalotiopsis adusta	E, P*	Leaf spot	hexandrum/Rubus idaeus (D)	China	(2018), Yan et al. (2019)
Pestalotiopsis affinis	P	Leaf spot	Taxus chinensis (G)	China	Chen et al. (2002)
Pestalotiopsis alpiniae	P	Leaf spot	Alpinia galanga (M)	China	Chen et al. (2002)
Pestalotiopsis antenniformis	P	Stem canker	Rubus cissoides , Rubus australis (D)	New Zealand	Index fungorum (2022)
Pestalotiopsis antiaris	P	Leaf spot	Antiaris toxicaria (M)	China	Chen et al. (2002)
Pestalotiopsis apiculata	Unknown		Cunninghamia lanceolata (G)	China	Index fungorum (2022)
Pestalotiopsis arborei	Unknown		Rhododendron arboretum (D)	India	Index fungorum (2022)
Pestalotiopsis bicilia	S		Viburnum opulus (D)	Canada	Index fungorum (2022)

Pestalotiopsis biciliata	P*	Stem canker	Pistacia lentiscus (D)	Tunisia	Hlaiem et al. (2022)
Pestalotiopsis bicolor	S		Smilax sp. (M)	U.S.A	Index fungorum (2022)
Pestalotiopsis brideliae	Unknown		Bridelia monoica (D)	China	Index fungorum (2022)
Pestalotiopsis bruguierae	Unknown		Bruguiera gymnorhiza (D)	India	Index fungorum (2022)
			Bulbophyllum		
Pestalotiopsis bulbophylli	S		thouars (M)	China	Wang et al. (2017)
Pestalotiopsis canarii	Unknown		Canarium album (D)	China	Index fungorum (2022)
Pestalotiopsis caroliniana	Unknown		Euonymus japonicus (D)	U.S.A	Index fungorum (2022)
Pestalotiopsis chamaeropsis	E, P*	Leaf spot	Eurya nitida , Peristrophe japonica (D)	China	Qiu et al. (2022), this study
Pestalotiopsis Chiangmaiensis	P	Leaf strip	Bamboo (M)	Thailand	This study
Pestalotiopsis cruenta	Unknown		Polygonum lasianthum (D)	Japan	Index fungorum (2022)
Pestalotiopsis digitalis	p	Leaf spot	Digitalis purpurea (D)	New Zealand	Liu et al. (2015)
Pestalotiopsis dilleniae	p	Leaf spot	Dillenia turbinata (M)	China	Chen et al. (2002) Maharachchikumbura et al. (2014)
Pestalotiopsis diploclisiae	Unknown		Diploclisia glaucescens (D)	Hong Kong, China	(2014)
Pestalotiopsis dracaenae	S		Dracaena fragrans (M)	China	Ariyawansa et al. (2015)
Pestalotiopsis ellipsospora	p*	Stem canker	Acanthopanax divaricatus (D)	Korea	Yun et al. (2015)
Pestalotiopsis gibbosa	S		Gaultheria shallon (D)	U.S.A	Watanabe et al. (2018)
Pestalotiopsis heucherae			Heuchera parviflora (D)	U.S.A	Index fungorum (2022)
Pestalotiopsis hughesii	Unknown		Cyperus articulatus (M)	Ghana	Index fungorum (2022)
Pestalotiopsis japonica	Unknown		Cedrela sinensis (D)	Japan	Index fungorum (2022)
Pestalotiopsis jinchanghensis	E		Vaccinium dunalianum (D)	China	Fan et al. (2020)
Pestalotiopsis kenya	P*	Leaf spot	Zanthoxylum schinifolium (D)	China	Liu et al. (2021)
Pestalotiopsis kunmingensis	E		Podocarpus macrophyllus (G)	China	Wei and Xu (2004)
Pestalotiopsis kwangsiensis	P	Leaf spot	Sinopimelodendron kwangsiensis (D)	China	Chen et al. (2002)
Pestalotiopsis lawsoniae	Unknown		Lawsonia alba (D)	India	Index fungorum (2022)

Pestalotiopsis lespedezae	Unknown		Lepedeza bicolor (D)	Japan	Index fungorum (2022) Maharachchikumbura et al. (2012)
Pestalotiopsis linearis	E		Trachelospermum sp. (D)	China	(2012)
Pestalotiopsis lushanensis	P*	Brown leaf spot, leaf blight	Sarcandra glabra , Podocarpus macrophyllus (G)	China	Zhang et al. (2021)
Pestalotiopsis microspora	S		Hedera helix (D)	Argentina	Index fungorum (2022)
Pestalotiopsis moluccensis	Unknown		Xylocarpus moluccensis (D)	India	Index fungorum (2022)
Pestalotiopsis neolitsea	P*	Leaf spot	Neolitsea villosa (D)	Taiwan, China	Ariyawansa and Hyde (2018)
Pestalotiopsis oenotherae	Unknown		Oenothera laciniata (D)	U.S.A	Venkatasubbaiah et al. (1991)
Pestalotiopsis pandani	Unknown		Pandanus odoratissimus (M)	Taiwan, China	Index fungorum (2022)
Pestalotiopsis paraguariensis	Unknown		Ilex paraguariensis (D)	Brazil	Index fungorum (2022)
Pestalotiopsis peristropheidis	E		Peristrophe japonica (D)	China	This study
Pestalotiopsis pestalozzioides	Unknown		Clematis ligusticifolia (D)	New Mexico	Index fungorum (2022)
Pestalotiopsis pipericola	Unknown		Piper nigrum (D)	India	Index fungorum (2022)
Pestalotiopsis quadriciliata	Unknown		Vitis vulpina (D)	Canada	Index fungorum (2022)
Pestalotiopsis rhodomyrtus	Unknown		Rhodomyrtus tomentosa (D)	China	Song et al. (2013)
Pestalotiopsis smilaxe	P	Leaf spot	Smilax china , Dioscorea sp. (M)	Thailand	This study
Pestalotiopsis sinensis	Unknown		Ginkgo biloba (G)	China	Index fungorum (2022)
Pestalotiopsis tecomicola	Unknown		Tecoma radicans (D)	U.S.A	Index fungorum (2022)
Pestalotiopsis thailandica	P	Leaf spot	Rhizophora apiculata (D)	Thailand	Norphanphoun et al. (2019)
Pseudopestalotiopsis ampullace	E		Magnolia candolli (D)	China	de Silva et al. (2021)
Pseudopestalotiopsis curvatispora	P	Leaf spot	Rhizophora mucronata (D)	Thailand	Norphanphoun et al. (2019)
Pseudopestalotiopsis gilvanii	P*	Leaf spot	Paullinia cupana (D)	Brazil	Gualberto et al. (2021) Maharachchikumbura et al. (2014)
Pseudopestalotiopsis indica	Unknown		Hibiscus rosa-sinensis (D)	India	(2014) Song et al. (2014), de Silva et al. (2021)
Pseudopestalotiopsis simitheae	S, E		Pandanus odoratissimus / Magnolia candolli (M/D)	Thailand/China	al. (2021)

Pseudopestalotiopsis thailandica	P	Leaf spot	Rhizophora mucronate (D)	Thailand	Norphanphoun et al. (2019)
Pseudoestalotiopsis theae	S		Ceriops tagal (D)	Thailand	This study

^aThe checklist includes species names, life modes, disease names (if any), hosts, locations and references. The current name is used according to Index Fungorum (2022).

^bThe mode of life is given as (E) endophyte, (P) pathogen and (S) saprobe. For the species, those with confirmed pathogenicity data are marked with an asterisk (*).

^cThe taxonomic status of the host is given as (D) Dicotyledons, (F) Ferns, (G) Gymnosperms and (M) Monocotyledons.

Acknowledgements

We would like to thank Dr. Shaun Pennycook for checking the nomenclature. We would like to thank Professor Abhaya Balasuriya, the Onsite Visiting Scholars for World Class Research Collaboration Programme under the Reinventing University System Project sponsored by Ministry of Higher Education, Science, Research and Innovation, Thailand. Ya-Ru Sun thanks Mae Fah Luang University for the award of a fee-less scholarship. Ya-Ru Sun also thanks Jing-Yi Zhang and Song Wang for collecting the samples. The study was funded by Guizhou Science Technology Department International Cooperation Basic Project ([2018]5806), National Natural Science Foundation of China (No.31972222, 31560489), Program of Introducing Talents of Discipline to Universities of China (111 Program, D20023), and Talent project of Guizhou Science and Technology Cooperation Platform ([2017]57885, [2019]5641 and [2020]5001).

References

- Abtahi F, Nourani SL (2017) The Most Important Fungal Diseases Associated with Some Useful Medicinal Plants. In: Ghorbanpour M, Varma A (eds) Medicinal Plants and Environmental Challenges. Springer International Publishing, Cham, pp. 279–293. https://doi.org/10.1007/978-3-319-68717-9_16
- Ariyawansa HA, Hyde KD (2018) Additions to *Pestalotiopsis* in Taiwan. *Mycosphere* 9 (5):999–1013.
- Ariyawansa HA, Hyde KD, Jayasiri SC, Buyck B, Chethana KWT, Dai DQ, Dai YC, Daranagama DA, Jayawardena RS, Lücking R, Ghobad-Nejhad M, Niskanen T, Thambugala KM, Voigt K, Zhao RL, Li GJ, Doilom M, Boonmee S, Yang ZL, Cai Q, Cui YY, Bahkali AH, Chen J, Cui BK, Chen JJ, Dayarathne MC, Dissanayake AJ, Ekanayaka AH, Hashimoto A, Hongsanan S, Jones EBG, Larsson E, Li WJ, Li QR, Liu JK, Luo ZL, Maharachchikumbura SSN, Mapook A, McKenzie EHC, Norphanphoun C, Konta S, Pang KL, Perera RH, Phookamsak R, Phukhamsakda C, Pinruan U, Randrianjohany E, Singtripop C, Tanaka K, Tian CM, Tibpromma S, Abdel-Wahab MA, Wanasinghe DN, Wijayawardene NN, Zhang JF, Zhang H, Abdel-Aziz FA, Wedin M, Westberg M, Ammirati JF, Bulgakov TS, Lima DX, Callaghan TM, Callac P, Chang CH, Coca LF, Dal-Forno M, Dollhofer V, Fliegerová K, Greiner K, Griffith GW, Ho HM, Hofstetter V, Jeewon R, Kang JC, Wen TC, Kirk PM, Kytövuori I, Lawrey JD, Xing J, Li H, Liu ZY, Liu XZ, Liimatainen K, Lumbsch HT, Matsumura M, Moncada B, Nuankaew S, Parnmen S, de Azevedo Santiago ALCM, Sommai S, Song Y, de Souza CAF, de Souza-Motta CM, Su HY, Suetrong S, Wang Y, Wei SF, Wen TC, Yuan HS, Zhou LW, Réblová M, Fournier J, Camporesi E, Luangsa-ard JJ, Tasanathai K, Khonsanit A, Thanakitpipattana D, Somrithipol S, Diederich P, Millanes AM, Common RS, Stadler M, Yan JY, Li X, Lee HW, Nguyen TTT, Lee HB, Battistin E, Marsico O, Vizzini A, Vila J, Ercole E, Eberhardt U, Simonini G, Wen H-A, Chen XH, Miettinen O, Spirin V, Hernawati

- (2015) Fungal diversity notes 111–252—taxonomic and phylogenetic contributions to fungal taxa. *Fungal Diversity* 75 (1):27–274.
- Barr ME (1975). *Pestalospaeria*, a new genus in the Amphisphaeriaceae. *Mycologia* 67: 187–194.
- Biju CN, Peeran MF, Gowri R (2018) Identification and characterization of *Neopestalotiopsis clavispora* associated with leaf blight of small cardamom (*Elettaria cardamomum* Maton). *Journal of Phytopathology* 166, (7-8):532–546.
- Cao P, Fang YH, Zheng ZK, Han X, Zou HX, Yan XF (2022) Occurrence of *Neopestalotiopsis clavispora* Causing Leaf Spot on *Dendrobium officinale* in China. *Plant Disease* 11–21.
- Chen CQ, Zhang B, Yang L, Gao J (2011) Identification and biological characteristics of round leaf spot on blueberry caused by *Pestalotiopsis photiniae*. *Journal of Northeast Forestry University* 39 (1):95–98.
- Chen YJ, Wan YH, Zeng L, Meng Q, Yuan LY, Tong HR (2021) Characterization of *Pestalotiopsis chamaeropsis* causing gray blight disease on tea leaves (*Camellia sinensis*) in Chongqing, China. *Canadian Journal of Plant Pathology* 43 (3):413–420.
- Chen YX, Wei G, Chen WP (2002) New species of *Pestalotiopsis*. *Mycosystema* 21 (3):316–323.
- Chernomor O, Von Haeseler A, Minh BQ (2016) Terrace aware data structure for phylogenomic inference from supermatrices. *Systematic biology* 65 (6):997–1008.
- David B, Wolfender JL, Dias DA (2015) The pharmaceutical industry and natural products: historical status and new trends. *Phytochemistry Reviews* 14 (2):299–315.
- de Silva N, Maharachchikumbura SSN, Thambugala KM, Bhat DJ, Karunarathna SC, Tennakoon DS, Phookamsak R, Jayawardena RS, Lumyong S, Hyde KD (2021) Morpho-molecular taxonomic studies reveal a high number of endophytic fungi from *Magnolia candolli* and *M. garrettii* in China and Thailand. *Mycosphere* 12 (1):163–237.
- Ding G, Qi Y, Liu S, Guo L, Chen X (2012) Photopyrones A and B, new pyrone derivatives from the plant endophytic fungus *Pestalotiopsis photiniae*. *The Journal of Antibiotics* 65 (5):271–273.
- Diogo E, Gonçalves CI, Silva AC, Valente C, Bragança H, Phillips AJL (2021) Five new species of *Neopestalotiopsis* associated with diseased *Eucalyptus* spp. in Portugal. *Mycological Progress* 20 (11):1441–1456. <https://doi.org/10.1007/s11557-021-01741-5>
- Dissanayake AJ, Bhunjun CS, Maharachchikumbura SSN, Liu JK (2020) Applied aspects of methods to infer phylogenetic relationships amongst fungi. *Mycosphere* 11 (1):2652–2676.
- Elfita E, Muharni M, Mardiyanto M, Fitrya F, Fera F, Widjajanti H (2020a) Antibacterial activity of traditional medicine *Scurrula atropurpurea* (BL) DANS and their endophytic fungi. In: *Key Engineering Materials, Trans Tech Publ*, pp 205–213.

- Elfita E, Muharni M, Mardiyanto M, Fitrya F, Nurmawati E, Widjajanti H (2020b) Triacylglycerols produced by biomass of endophytic fungus *Neopestalotiopsis surinamensis* from the *Scurrula atropurpurea*. IJFAC (Indonesian Journal of Fundamental and Applied Chemistry) 5 (3):95–100.
- Fan M, Chen X, Luo X, Zhang H, Liu Y, Zhang Y, Wu J, Zhao C, Zhao P (2020) Diversity of endophytic fungi from the leaves of *Vaccinium dunalianum*. Letters in Applied Microbiology 71 (5):479–489.
- Gualberto GF, CATARINO AdM, Sousa TF, da CRUZ J, Hanada RE, Caniato FF, da SILVA G (2021) *Pseudopestalotiopsis gilvanii* sp. nov. and *Neopestalotiopsis formicarum* leaves spot pathogens from guarana plant: a new threat to global tropical hosts. Embrapa Amazônia Ocidental-Artigo em periódico indexado (ALICE).
- Guba EF (1956). *Monochaetia* and *Pestalotia* vs. *Truncatella*, *Pestalotiopsis* and *Pestalotia*. Annals of Microbiology 7:74–76.
- Guba EF (1961). Monograph of *Pestalotia* and *Monochaetia*. Harvard University Press, Cambridge.
- Guo SX (2016) Endophytic fungi biology of medicinal plant. Science Press, Beijing. (Chinese)
- Hall TA BioEdit: a user-friendly biological sequence alignment editor and analysis program for Windows 95/98/NT. In, 1999. pp 95–98.
- Hlaiem S, Zouaoui-Boutiti M, Ben Jemâa M, Della Rocca G, Barberini S, Danti R (2018) Identification and pathogenicity of *Pestalotiopsis chamaeropsis*, causal agent of white heather (*Erica arborea*) dieback, and in vitro biocontrol with the antagonist *Trichoderma* sp. Tunisian Journal of Plant Protection 13 (Special Issue):49–60.
- Hoang DT, Chernomor O, Von Haeseler A, Minh BQ, Vinh LS (2018) UFBoot2: improving the ultrafast bootstrap approximation. Molecular Biology and Evolution 35 (2):518–522.
- Huang WY, Cai YZ, Hyde KD, Corke H, Sun M (2008) Biodiversity of endophytic fungi associated with 29 traditional Chinese medicinal plants. Fungal diversity 33:61–75.
- Huanaluek N, Jayawardena RS, Maharachchikumbura SS, Harishchandra DL (2021) Additions to pestalotioid fungi in Thailand: *Neopestalotiopsis hydeana* sp. nov. and *Pestalotiopsis hydei* sp. nov. Phytotaxa 479 (1):23–43.
- Hyde KD, Nilsson RH, Alias SA, Ariyawansa HA, Blair JE, Cai L, de Cock AWAM, Dissanayake AJ, Glockling SL, Goonasekara ID, Gorczak M, Hahn M, Jayawardena RS, van Kan JAL, Laurence MH, Lévesque CA, Li XH, Liu JK, Maharachchikumbura SSN, Manamgoda DS, Martin FN, McKenzie EHC, McTaggart AR, Mortimer PE, Nair PVR, Pawłowska J, Rintoul TL, Shivas RG, Spies CFJ, Summerell BA, Taylor PWJ, Terhem RB, Udayanga D, Vaghefi N, Walther G, Wilk M, Wrzosek M, Xu JC, Yan JY, Zhou N (2014) One stop shop: backbones trees for important phytopathogenic genera: I (2014). Fungal Diversity 67 (1):21–125.
- Jayawardena SC, Hyde KD, Ariyawansa HA, Bhat J, Buyck B, Cai L, Dai YC,

- Abd-Elsalam KA, Ertz D, Hidayat I, Jeewon R, Jones EBG, Bahkali AH, Karunarathna SC, Liu JK, Luangsa-ard JJ, Lumbsch HT, Maharachchikumbura SSN, McKenzie EHC, Moncalvo JM, Ghobad-Nejhad M, Nilsson H, Pang KL, Pereira OL, Phillips AJL, Raspé O, Rollins AW, Romero AI, Etayo J, Selçuk F, Stephenson SL, Suetrong S, Taylor JE, Tsui CKM, Vizzini A, Abdel-Wahab MA, Wen TC, Boonmee S, Dai DQ, Daranagama DA, Dissanayake AJ, Ekanayaka AH, Fryar SC, Hongsanan S, Jayawardena RS, Li WJ, Perera RH, Phookamsak R, de Silva NI, Thambugala KM, Tian Q, Wijayawardene NN, Zhao RL, Zhao Q, Kang JC, Promputtha I (2015) The Faces of Fungi database: fungal names linked with morphology, phylogeny and human impacts. *Fungal Diversity* 74:3–18. <https://doi.org/10.1007/s13225-015-0351-8>
- Jayawardena RS, Hyde KD, de Farias ARG, Bhunjun CS, Fernandez HS, Manamgoda DS, Udayanga D, Herath IS, Thambugala KM, Manawasinghe IS, Gajanayake AJ, Samarakoon BC, Bundhun D, Gomdola D, Huanraluek N, Sun Y-r, Tang X, Promputtha I, Thines M (2021) What is a species in fungal plant pathogens? *Fungal Diversity* 109 (1):239–266. <https://doi.org/10.1007/s13225-021-00484-8>
- Jayawardena RS, Hyde KD, McKenzie EHC, Jeewon R, Phillips AJL, Perera RH, de Silva NI, Maharachchikumburua SSN, Samarakoon MC, Ekanayake AH, Tennakoon DS, Dissanayake AJ, Norphanphoun C, Lin C, Manawasinghe IS, Tian Q, Brahmanage R, Chomnunti P, Hongsanan S, Jayasiri SC, Halleen F, Bhunjun CS, Karunarathna A, Wang Y (2019) One stop shop III: taxonomic update with molecular phylogeny for important phytopathogenic genera: 51–75 (2019). *Fungal Diversity* 98 (1):77–160. <https://doi.org/10.1007/s13225-019-00433-6>
- Jayawardena RS, Liu M, Maharachchikumbura SSN, Zhang W, Xing QK, Hyde KD, Nilthong S, Li XH, Yan JY (2016) *Neopestalotiopsis vitis* sp. nov. causing grapevine leaf spot in China. *Phytotaxa* 258 (1):63–74.
- Jia M, Chen L, Xin HL, Zheng CJ, Rahman K, Han T, Qin LP (2016) A friendly relationship between endophytic fungi and medicinal plants: a systematic review. *Frontiers in microbiology* 7:906.
- Jiang N, Bonthond G, Fan XL, Tian CM (2018) *Neopestalotiopsis rosicola* sp. nov. causing stem canker of *Rosa chinensis* in China. *Mycotaxon* 133 (2):271–283.
- Jiang N, Fan XL, Tian CM (2021) Identification and Characterization of Leaf-Inhabiting Fungi from *Castanea Plantations* in China. *Journal of Fungi* 7. <https://doi.org/10.3390/jof7010064>
- Jiang YL, Zhou YK, Ma KZ, Mao YL, Yu XT, Liu Y, Hou CL (2017) Identification of several strains of *Pestalotiopsis*-like fungi from *Camellia oleifera*. *Journal of Anhui Agricultural University* 44 (4):609–616.
- Joshi SD, Sanjay R, Baby UI, Mandal A (2009) Molecular characterization of *Pestalotiopsis* spp. associated with tea (*Camellia sinensis*) in southern India using RAPD and ISSR markers.
- Keshri PK, Rai N, Verma A, Kamble SC, Barik S, Mishra P, Singh SK, Salvi P, Gautam V (2021) Biological potential of bioactive metabolites derived from

- fungal endophytes associated with medicinal plants. *Mycological Progress* 20 (5):577–594. <https://doi.org/10.1007/s11557-021-01695-8>
- Khoo YW, Tan HT, Khaw YS, Li S, Chong KP (2022) First Report of *Neopestalotiopsis cubana* Causing leaf blight on *Ixora chinensis* in Malaysia. *Plant Disease* (Online).
- Kjer J, Debbab A, Aly AH, Proksch P (2010) Methods for isolation of marine-derived endophytic fungi and their bioactive secondary products. *Nature protocols* 5 (3):479–490.
- Kumaran RS, Kim HJ, Hur BK (2010) Taxol promising fungal endophyte, *Pestalotiopsis* species isolated from *Taxus cuspidata*. *Journal of Bioscience and Bioengineering* 110 (5):541–546.
- Kumar V, Cheewangkoon R, Gentekaki E, Maharachchikumbura SSN, Brahmanage RS, Hyde KD (2019) *Neopestalotiopsis alpapicalis* sp. nov. a new endophyte from tropical mangrove trees in Krabi Province (Thailand). *Phytotaxa* 393 (3):251–262.
- Larsson A (2014) AliView: a fast and lightweight alignment viewer and editor for large datasets. *Bioinformatics* 30 (22):3276–3278. <https://doi.org/10.1093/bioinformatics/btu531>
- Li L, Yang Q, Li H (2021) Morphology, Phylogeny, and Pathogenicity of Pestalotioid Species on *Camellia oleifera* in China. *Journal of Fungi* 7 (12). <https://doi.org/10.3390/jof7121080>
- Li Y, Zhang S, Wang T, Zhou S, Wu Y, Huang X, Lin H, Su X (2022) First report of *Taxus media* branch blight caused by *Neopestalotiopsis clavispora* in China. *Plant Disease* (Online).
- Liu C, Luo F, Zhu T, Han S, Li S (2021) Leaf Spot Disease Caused by *Pestalotiopsis kenyana* on *Zanthoxylum schinifolium* in Sichuan Province, China. *Plant Disease* 105 (11):3747.
- Liu F, Hou LW, Raza M, Cai L (2017) *Pestalotiopsis* and allied genera from *Camellia*, with description of 11 new species from China. *Scientific reports* 7:866.
- Liu JK, Hyde KD, Jones EBG, Ariyawansa HA, Bhat DJ, Boonmee S, Maharachchikumbura SSN, McKenzie EHC, Phookamsak R, Phukhamsakda C, Shenoy BD, Abdel-Wahab MA, Buyck B, Chen J, Chethana KWT, Singtripop C, Dai DQ, Dai YC, Daranagama DA, Dissanayake AJ, Doilom M, D'souza MJ, Fan XL, Goonasekara ID, Hirayama K, Hongsan S, Jayasiri SC, Jayawardena RS, Karunarathna SC, Li WJ, Mapook A, Norphanphoun C, Pang KL, Perera RH, Peršoh D, Pinruan U, Senanayake IC, Somrithipol S, Suetrong S, Tanaka K, Thambugala KM, Tian Q, Tibpromma S, Udayanga D, Wijayawardene NN, Wanasinghe D, Wisitrassameewong K, Zeng XY, Abdel-Aziz FA, Adamčík S, Bahkali AH, Boonyuen N, Bulgakov T, Callac P, Chomnunti P, Greiner K, Hashimoto A, Hofstetter V, Kang JC, Lewis D, Li XH, Liu XZ, Liu ZY, Matsumura M, Mortimer PE, Rambold G, Randrianjohany E, Sato G, Sri-Indrasutdhi V, Tian CM, Verbeken A, von Brackel W, Wang Y, Wen TC, Xu JC, Yan JY, Zhao RL, Camporesi E (2015) Fungal diversity notes 1–110: taxonomic and phylogenetic contributions to fungal species. *Fungal Diversity* 72

- (1):1–197. <https://doi.org/10.1007/s13225-015-0324-y>
- Long Y, Hsiang T, Huang J (2009) First report of leaf spot of *Smilax china* caused by *Alternaria longipes* in China. *Plant Pathology* 58.
- Ma XY, Maharachchikumbura SSN, Chen BW, Hyde KD, McKenzie EHC, Chomnunti P, Kang JC (2019) Endophytic pestalotioid taxa in *Dendrobium* orchids. *Phytotaxa* 419 (3):268–286
- Mahapatra S, Banerjee J, Kumar K, Pramanik S, Pramanik K, Islam S, Das S (2018) Leaf spot and fruit rot of strawberry caused by *Neopestalotiopsis clavispora* in Indo-Gangetic plains of India. *Indian Phytopathology* 71 (2):279–283. <https://doi.org/10.1007/s42360-018-0043-x>
- Maharachchikumbura SSN, Guo LD, Cai L, Chukeatirote E, Wu WP, Sun X, Crous PW, Bhat DJ, McKenzie EHC, Bahkali AH, Hyde KD (2012) A multi-locus backbone tree for *Pestalotiopsis*, with a polyphasic characterization of 14 new species. *Fungal Diversity* 56 (1):95–129.
- Maharachchikumbura SSN, Guo LD, Liu ZY, Hyde KD (2016a) *Pseudopestalotiopsis ignota* and *Ps. camelliae* spp. nov. associated with grey blight disease of tea in China. *Mycological Progress* 15 (3):22.
- Maharachchikumbura SSN, Guo LD, Chukeatirote E, McKenzie EHC, Hyde KD (2013) A destructive new disease of *Syzygium samarangense* in Thailand caused by the new species *Pestalotiopsis samarangensis*. *Tropical Plant Pathology* 38 (3):227–235.
- Maharachchikumbura SSN, Hyde KD, Groenewald JZ, Xu J, Crous PW (2014) *Pestalotiopsis* revisited. *Studies in Mycology* 79:121–186.
- Maharachchikumbura SSN, Laignon P, Hyde KD, Al-Sadi AM, Liu ZY (2016b) Characterization of *Neopestalotiopsis*, *Pestalotiopsis* and *Truncatella* species associated with grapevine trunk diseases in France. *Phytopathologia Mediterranea*:380–390.
- Miller MA, Pfeiffer W, Schwartz T Creating the CIPRES Science Gateway for inference of large phylogenetic trees. In: Proceeding of the 2010 gateway computing environments workshop (GCE), New Orleans, Louisiana, 2010. pp 1–8. <https://doi.org/10.1109/GCE.2010.5676129>
- Minh BQ, Nguyen MAT, von Haeseler A (2013) Ultrafast approximation for phylogenetic bootstrap. *Molecular biology and evolution* 30 (5):1188–1195.
- Moreau C (1949). *Micromycetes africains*. I. *Revue de Mycologie, Supplement Colonial* (Paris) 14:15–22.
- Moslemi A, Taylor PWJ (2015) *Pestalotiopsis chamaeropsis* causing leaf spot disease of round leaf mint-bush (*Prostanthera rotundifolia*) in Australia. *Australasian Plant Disease Notes* 10 (1):29.
- Nag Raj TR (1993). *Coelomycetous anamorphs with appendage-bearing conidia*. Mycologue Publications, Waterloo, Ontario, Canada.
- Nguyen LT, Schmidt HA, Von Haeseler A, Minh BQ (2015) IQ-TREE: a fast and effective stochastic algorithm for estimating maximum-likelihood phylogenies. *Molecular biology and evolution* 32 (1):268–274
- Norphanhoun C, Jayawardena RS, Chen Y, Wen TC, Meepol W, Hyde KD (2019)

- Morphological and phylogenetic characterization of novel pestalotioid species associated with mangroves in Thailand. *Mycosphere* 10:531–578
- Nozawa S, Ando K, Phay N, Watanabe K (2018) *Pseudopestalotiopsis dawaina* sp. nov. and *Ps. kawthaungina* sp. nov.: two new species from Myanmar. *Mycological Progress* 17 (7):865–870. <https://doi.org/10.1007/s11557-018-1398-1>
- Nozawa S, Yamaguchi K, Van Hop D, Phay N, Ando K, Watanabe K (2017) Identification of two new species and a sexual morph from the genus *Pseudopestalotiopsis*. *Mycoscience* 58 (5):328–337.
- Nozawa S, Seto Y, Watanabe K (2019) First report of leaf blight caused by *Pestalotiopsis chamaeropsis* and *Neopestalotiopsis* sp. in Japanese andromeda. *Journal of General Plant Pathology* 85 (6):449–452.
- Nylander JAA (2004) MrModeltest v2.2 Program distributed by the author: 2. Evolutionary Biology Centre, Uppsala University.
- Park H, Shim JS, Kim JS, Choi HS, Eom AH (2017) Five Previously Unreported Endophytic Fungi Isolated from the Leaves of Woody Plants in Korea. *The Korean Journal of Mycology* 45 (4):345–354.
- Prasannath K, Shivas RG, Galea VJ, Akinsanmi OA (2021) *Neopestalotiopsis* species associated with flower diseases of *Macadamia integrifolia* in Australia. *Journal of Fungi* 7 (9):771
- Rambaut, A. FigTree. Version 1.4.2; University of Edinburgh: Edinburgh, UK, 2014.
- Rannala B, Yang ZH (1996) Probability distribution of molecular evolutionary trees: A new method of phylogenetic inference. *Journal of Molecular Evolution* 43:304–311. <https://doi.org/10.1007/BF02338839>
- Rasool-Hassan BA (2012) Medicinal plants (importance and uses). *Pharmaceut Analytica Acta* 3 (10):2153–2435
- Rasool A, Bhat KM, Sheikh AA, Jan A, Hassan S (2020) Medicinal plants: Role, distribution and future. *Journal of Pharmacognosy and Phytochemistry* 9 (2):2111–2114
- Reddy MS, Murali T, Suryanarayanan TS, Rajulu MBG, Thirunavukkarasu N (2016) *Pestalotiopsis* species occur as generalist endophytes in trees of Western Ghats forests of southern India. *Fungal Ecology* 24:70–75
- Ronquist F, Teslenko M, Van Der Mark P, Ayres DL, Darling A, Höhna S, Larget B, Liu L, Suchard MA, Huelsenbeck JP (2012) MrBayes 3.2: efficient Bayesian phylogenetic inference and model choice across a large model space. *Systematic Biology* 61 (3):539–542. <https://doi.org/10.1093/sysbio/sys029>
- Qiu L, Liu J, Kuang W, Zhang K, Ma J (2022) First Report of *Pestalotiopsis chamaeropsis* Causing Leaf Spot on *Eurya nitida* in China. *Plant Disease* 106 (1):329.
- Santos J, Hilário S, Pinto G, Alves A (2022) Diversity and pathogenicity of pestalotioid fungi associated with blueberry plants in Portugal, with description of three novel species of *Neopestalotiopsis*. *European Journal of Plant Pathology* 162 (3):539–555.
- Schafhauser T, Jahn L, Kirchner N, Kulik A, Flor L, Lang A, Caradec T, Fewer DP,

- Sivonen K, van Berkel WJ (2019) Antitumor astins originate from the fungal endophyte *Cyanoderrella asteris* living within the medicinal plant *Aster tataricus*. *Proceedings of the National Academy of Sciences* 116 (52):26909–26917
- Senanayake IC, Rathnayake AR, Marasinghe DS, Calabon MS, Gentekaki E, Lee HB, Hurdeal VG, Pem D, Dissanayake LS, Wijesinghe SN, Bundhun D, Nguyen TT, Goonasekara ID, Abeywickrama PD, Bhunjun CS, Jayawardena RS, Wanasinghe DN, Jeewon R, Bhat DJ, Xiang MM (2020) Morphological approaches in studying fungi: collection, examination, isolation, sporulation and preservation. *Mycosphere* 11 (1):2678–2754. <https://doi.org/10.5943/mycosphere/11/1/20>
- Sieber TN, Sieber-Canavesi F, Petrini O, Ekramoddoullah AKM, Dorworth CE (1991) Characterization of Canadian and European Melanconium from some *Alnus* species by morphological, cultural, and biochemical studies. *Canadian Journal of Botany* 69 (10):2170–2176.
- Silvério ML, Calvacanti MAdQ, Silva GAd, Oliveira RJVd, Bezerra JL (2016) A new epifoliar species of *Neopestalotiopsis* from Brazil. *Agrotrópica* 28 (2):151–158
- Song Y, Geng K, Hyde KD, Zhao WS, Wei JG, Kang JC, Wang Y (2013) Two new species of *Pestalotiopsis* from Southern China. *Phytotaxa* 126 (1):22–32
- Song Y, Tangthirasunun N, Maharachchikumbura SSN, Jiang YL, Xu JJ, Hyde KD, Wang Y (2014) Novel *Pestalotiopsis* species from Thailand point to the rich undiscovered diversity of this chemically creative genus. *Cryptogamie, Mycologie* 35 (2):139–149
- Steyaert RL (1949) Contribution à l'étude monographique de *Pestalotia* de Not. et *Monochaetia* Sacc. (*Truncatella* gen. nov. et *Pestalotiopsis* gen. nov.). 19:285–354.
- Sun J, Guo L, Zang W, Ping W, Chi D (2008) Diversity and ecological distribution of endophytic fungi associated with medicinal plants. *Science in China Series C: Life Sciences* 51 (8):751–759.
- Sutton BC (1980). *The Coelomycetes. Fungi imperfecti with pycnidia, acervuli and stromata*. Commonwealth Mycological Institute, Kew, Surrey, UK.
- Tibpromma S, Hyde KD, McKenzie EHC, Bhat DJ, Phillips AJL, Wanasinghe DN, Samarakoon MC, Jayawardena RS, Dissanayake AJ, Tennakoon DS, Doilom M, Phookamsak R, Tang AMC, Xu JC, Mortimer PE, Promputtha I, Maharachchikumbura SSN, Khan S, Karunarathna SC (2018) Fungal diversity notes 840–928: micro-fungi associated with Pandanaceae. *Fungal Diversity* 93:1–160.
- Trifinopoulos J, Nguyen LT, von Haeseler A, Minh BQ (2016) W-IQ-TREE: a fast online phylogenetic tool for maximum likelihood analysis. *Nucleic acids research* 44 (W1):W232–W235.
- Ul-Haq I, Ijaz S, Khan NA (2021) Genealogical concordance of phylogenetic species recognition-based delimitation of *Neopestalotiopsis* species associated with leaf spots and fruit canker disease affected guava plants. *Pakistan Journal of Agricultural Sciences* 58 (4).
- Vaidya G, Lohman DJ, Meier R (2011) *SequenceMatrix: concatenation software for*

- the fast assembly of multi-gene datasets with character set and codon information. *Cladistics* 27:171-180.
<https://doi.org/10.1111/j.1096-0031.2010.00329.x>
- Venkatasubbaiah P, Grand L, Van Dyke CG (1991) A new species of *Pestalotiopsis* on *Oenothera*. *Mycologia* 83 (4):511–513.
- Wang RB, Chen SZ, Zheng BW, Liu PQ, Li Bj, Weng QY, Chen QH (2019) Occurrence of leaf spot disease caused by *Neopestalotiopsis clavisporea* on *Taxus chinensis* in China. *Forest Pathology* 49 (5):e12540.
- Wang Y, Ran SF, Maharachchikumbura SSN, Al-Sadi AM, Hyde KD, Wang HL, Wang T, Wang YX (2017) A novel *Pestalotiopsis* species isolated from *Bulbophyllum thouars* in Guangxi province, China. *Phytotaxa* 306 (1):96–100.
- Wang YC, Xiong F, Lu QH, Hao XY, Zheng MX, Wang L, Li NN, Ding CQ, Wang XC, Yang YJ (2019) Diversity of *Pestalotiopsis*-like species causing gray blight disease of tea plants (*Camellia sinensis*) in China, including two novel *Pestalotiopsis* species, and analysis of their pathogenicity. *Plant Disease* 103 (10):2548–2558.
- Watanabe K, Nozawa S, Hsiang T, Callan B (2018) The cup fungus *Pestalopezia brunneopruinosa* is *Pestalotiopsis gibbosa* and belongs to Sordariomycetes. *PloS one* 13 (6):e0197025.
- Wei JG, Xu T (2004) *Pestalotiopsis kunmingensis* sp. nov., an endophyte from *Podocarpus macrophyllus*. *Fungal Diversity* 15:247–254.
- Weber D, Sterner O, Anke T, Gorzalczyk S, Martino V, Acevedo C (2004) Phomol, a new antiinflammatory metabolite from an endophyte of the medicinal plant *Erythrina crista-galli*. *The Journal of Antibiotics* 57 (9):559–563.
- Wei JG, Xu T, Huang WH, Guo LD, Pan XH (2008) Molecular phylogenetic investigation on relationship of endophytic and pathogenic *pestalotiopsis* species. *Journal of Zhejiang University* 34 (4).
- Xiao J, Lin LB, Hu JY, Duan DZ, Shi W, Zhang Q, Han WB, Wang L, Wang XL (2018) Pestalustaines A and B, unprecedented sesquiterpene and coumarin derivatives from endophytic fungus *Pestalotiopsis adusta*. *Tetrahedron Letters* 59 (18):1772–1775.
- Xu MF, Jia OY, Wang SJ, Zhu Q (2016) A new bioactive diterpenoid from *Pestalotiopsis adusta*, an endophytic fungus from *Clerodendrum canescens*. *Natural Product Research* 30 (23):2642–2647.
- Xu J, Ebada SS, Proksch P (2010) *Pestalotiopsis* a highly creative genus: chemistry and bioactivity of secondary metabolites. *Fungal Diversity* 44 (1):15–31.
<https://doi.org/10.1007/s13225-010-0055-z>
- Xu J, Yang XB, Lin Q (2014) Chemistry and biology of *Pestalotiopsis*-derived natural products. *Fungal Diversity* 66 (1):37–68.
<https://doi.org/10.1007/s13225-014-0288-3>
- Yang LN, Miao XY, Bai QR, Wang MQ, Gu ML, Zhao TC (2017) *Neopestalotiopsis clavisporea* causing leaf spot on *Phedimus aizoon var. latifolius*, a new disease in China. *Plant Disease* 101 (11):1952
- Yang Q, Zeng XY, Yuan J, Zhang Q, He YK, Wang Y (2021) Two new species of

Neopestalotiopsis from southern China. Biodiversity data journal 9

- Yan X, Meng T, Qi Y, Fei N, Liu C, Dai H (2019) First Report of *Pestalotiopsis adusta* Causing Leaf Spot on Raspberry in China. Plant Disease 103 (10):2688–2688.
- Yun YH, Ahn GR, Kim SH (2015) First report and characterization of *Pestalotiopsis ellipospora* causing canker on *Acanthopanax divaricatus*. Mycobiology 43 (3):366–370.
- Zhang GJ, Cheng XR, Ding HX, Liang S, Li Z (2021) First Report of *Pestalotiopsis lushanensis* Causing Brown Leaf Spot on *Sarcandra glabra* in China. Plant Disease 105 (4):1219.
- Zhang YM, Maharachchikumbura SSN, McKenzie EHC, Hyde KD (2012) A novel species of *Pestalotiopsis* causing leaf spots of *Trachycarpus fortunei*. Cryptogamie, Mycologie 33 (3):311–318.
- Zhang Z, Liu R, Liu S, Mu T, Zhang X, Xia J (2022) Morphological and phylogenetic analyses reveal two new species of Sporocadaceae from Hainan, China. MycoKeys 88:171–192. <https://doi.org/10.3897/mycokeys.88.82229>

Dear Editor,

We appreciate you and the reviewers for your precious time in reviewing our paper and providing valuable comments. We have carefully considered the comments and tried our best to address every one of them. We hope the manuscript after careful revisions meet your high standards.

Here is a point-by-point response to the reviewers' comments and concerns. All modifications in the "Marked-Up Manuscript" file have been highlighted.

Reviewer 1

1. "these 27 isolates represent 17 species distributed in three genera; including six new *Neopestalotiopsis* species, four new *Pestalotiopsis* species, and six new records." $6+4+6=16$, but it mentioned that there are 17 species, what about the other one

R: Thank you so much. We corrected this sentence to "these 26 isolates represent 17 species distributed in three genera: including seven new species and eight new records." By looking up google, we found that "including" means containing as part of the whole being considered. So contents after "including" are not all species.

2. "Genealogical Concordance Phylogenetic Species Recognition (GCPSR)". Should be "Genealogical concordance phylogenetic species recognition (GCPSR)"

R: Thank you so much. We corrected these.

3. "occurring commonly as important plant pathogens, endophytes, and saprophytes." delete "important"

R: Thank you so much. We deleted this word on page 1, line 22.

4. "(Guba 1961, Barr 1975, Nag Raj 1993, Maharachchikumbura et al. 2014, Hyde et al. 2014, Jayawardena et al. 2019, 2021, Norphanphoun et al. 2019, Ul Haq et al. 2021, Yang et al. 2021)." cite the major references is enough

R: Thanks for your suggestion. I deleted some literature.

5. "analyses" should be "analysis"

R: Thank you so much. We corrected it on page 2, line 13.

6. "*viz. Neopestalotiopsis, Pestalotiopsis, and Pseudopestalotiopsis*" i would suggest the authors provide more information for these three genera.

R: Thank you so much. We added some information as follows: The use of molecular data in resolving pestalotioid species was revisited by Maharachchikumbura et al. (2014) and they separated this group into three genera, *viz. Neopestalotiopsis, Pestalotiopsis, and Pseudopestalotiopsis*. Phylogenetically, these three genera formed distinct clades. Morphologically, *Neopestalotiopsis* differs from *Pseudopestalotiopsis* and *Pestalotiopsis* by its versicolourous (two upper median cells darker than the lowest median cell) median cells and indistinct conidiophores, while *Pseudopestalotiopsis* can be easily distinguished from *Pestalotiopsis* by darker colored concolourous (for those possessing equally pigmented median cells) median cells (Maharachchikumbura et al. 2014). Also, quantitatively, *Neopestalotiopsis, Pestalotiopsis* seem to infect plants more readily than *Pseudopestalotiopsis*.

7. "(Huang et al. 2008, Jia et al. 2016, Xu et al. 2010, 2014, Reddy et al. 2016, Ma et

al. 2019).” i guess the references are list year by year, if so, "Jia et al. 2016" should list after Xu et al. 2010, 2014. check all of MS carefully please.

R: We are sorry for our carelessness. We agreed with you and corrected it on page 3, line 3.

8. “The fresh samples of different medicinal plants were collected in southwest China and Thailand from 2019 to 2022.” leaves? twigs? or any other parts of the plant, please provide detailed information

R: Thank you so much. We provided more information as follows: The fresh healthy leaves, diseased leaves and twigs of different medicinal plants were collected from terrestrial habitats in southwest China and Thailand from 2019 to 2022.

9. “followed by soaking in 4% NaOCl for 1 min” 1 min is enough? normally it should be 3 mins.

R: Thank you so much. We isolated endophytes from young fresh leaves. At first, we soaked the leaves in 4% NaOCl for 3 minutes, but no mycelium grew, and later we shortened the time to 2 minutes, which also failed. Then we adjusted the time to 1 minute, there were mycelia growth, and there was no contamination in the control experiment. So, we think, for fresh young leaves of plants, one minute is enough.

10. “the Culture Collection of the Department of Plant Pathology, Agriculture College, Guizhou University (GUCC)” GZCC may not a international even national culture collection, please deposit the pure cultures to an national culture collection. for your cultures which collected from China, deposit them to CGMCC.

R: Thank you so much. We deposited our cultures to CGMCC and KUNCC and updated the KUNCC codes in the MS. But we haven't got the CGMCC codes due to COVID-19.

11. “the herbarium of the Department of Plant Pathology, Agricultural College, Guizhou University (HGUP), Guiyang, China” same as above, store the specimens to HKAS, otherwise the new taxa may invalid

R: Thank you so much. We already deposited our specimens into HKAS and updated the specimen codes in the MS.

12. “The maximum likelihood (ML) analyses were carried out using IQ-TREE” “analyses were” should be “analysis was”

R: Thank you so much. But in this study, we carried out the ML and BYPP tree for three genera and we prefer using “analyses were”.

13. do not forget to provide the accession number

R: Thank you so much. We provided the accession numbers in the MS now.

14. this is Herbarium number, not strain number, is it mean that you directly extract DNA from fruiting body of the fungi? if so, please update the method parts. if not, please provide strain number for all of newly obtained.

R: Thanks for your suggestions. Yes, we directly extract DNA from the fruiting body of some fungi, and we updated the method on page 4, line 10. For others, we provided strain numbers in the MS.

15. “SAUCC212272” add space, check all

R: Thank you so much. We checked the original literature (doi: 10.3897/mycokeys.88.82229), no spaces needed.

16. "GUCC 21504" i guess 21-504??

R: Thank you so much. We checked the original literature (doi: 10.3897/BDJ.9.e70446) and determined that it was "GUCC 21504".

17. "HKAS1124560" add space, check all

R: Thank you so much. We checked my MS carefully and corrected them.

18. "IFXX;" "update all of IF and FOF numbers please"

R: Thank you so much. We registered the names of the new species at Fungorum names and also applied for the FOF numbers. We provided codes in the MS now.

19. "Conidia $18.5\text{--}30.0 \times 4.5\text{--}7.5\mu\text{m}$, $x \pm \text{SD} = 25 \pm 2.5 \times 6.0 \pm 0.68 \mu\text{m}$ (n = 40)," "Conidia" should italic. in case of we use the μm as unit, so please use integer for the measurement value: at least the nearest half, need to do decimals to round up and round down numbers

R: Thank you for pointing this out. We carefully checked the manuscript and corrected all measurement values, and words that needed to be italicized.

20. "Qiannan Buyei and Miao Autonomous Prefecture" "Buyei" should be "Bouyei"

R: Thank you for pointing this out. We corrected these names throughout the MS.

21. "ex-type-living" delete "living"

R: Thank you so much. We deleted them following your suggestion.

22. "GUCC 21001" 21-001?

R: Thank you so much. We checked the original literature (doi: 10.3897/BDJ.10.e90709) and determined that it was "GUCC 21001".

23. "dark brown to dark" ?? i guess "black"

R: Thank you so much. We agree with you and correct it to black.

24. "ex-type culture" make all of this in same format, please check it is singular or plural carefully for the whole paper as well.

R: Thank you so much. We carefully checked the manuscript and made them in the same format.

25. "Part 2 Pestalotiopsis" For all of new Pestalotiopsis species introduced in this paper, the authors need to check the newly paper on Frontiers in Microbiology: Gu R, Bao D-F, Shen H-W, Su X-J, Li Y-X and Luo Z-L (2022) Endophytic Pestalotiopsis species associated with Rhododendron in Cangshan Mountain, Yunnan Province, China. Front. Microbiol. 13:1016782.

In case of this paper have been accepted by the Journal and it is process to paper proof state now, so please make sure the species in your paper are not repeat with the published species.

R: Thanks for your suggestion. We added these species to our phylogenetic tree and we confirmed that our species were not conspecific with these published species.

26. "Index fungorum (2022)" the authors need to cite the original references, not the IF dataset, it is incorrect check and update all, please.

R: Thanks for your suggestion. We agreed with you and updated all missing references in the MS.

Reviewer 2

1. Authors did not follow the recent taxonomic conflicts on *Neopestalotiopsis* and introduced several new species, I would like the get a clear answer why some

studies prefer to introduce new species as *Neopestalotiopsis* sp. while you choose to introduce them as new species.

R: Thanks for your comments. As far as we know, since the establishment of *Neopestalotiopsis*, a total of 49 new species have been introduced by mycologists based on morphological and phylogenetic analysis (Maharachchikumbura et al. 2014, Jayawardena et al. 2016, Jiang et al. 2018, Freitas et al. 2019, Kumar et al. 2019, Norphanphoun et al. 2019, Diogo et al. 2021, Prasannath et al. 2021, Fiorenza et al. 2022). Although some scholars believe that these two methods are not enough to introduce new species, we can see from their analysis that the taxonomic status of their strains is not stable enough or lacks sister branches (Belisário et al. 2020, Gerardo-Lugo et al. 2020, Jiang et al. 2021). Therefore, we suggest that new species can be introduced if the following conditions are met: a) there are obvious differences in morphological characteristics, b) there are obvious branch lengths in the phylogenetic analysis, c) strains can form stable clades with known species. In our study, there are 15 isolates representing 9 different clades, and only 4 new species are introduced. For the strains that do not meet the above conditions, we still keep *Neopestalotiopsis* sp.

2. As I can see in the material and methods, the authors have not chosen 2022 pulsations to construct a phylogenetic tree, thus I am not sure still your species are new. Therefore, both trees required an update following the 2022 papers.

R: Thanks for your reminder. We added sequences from the most recently published species and reanalyzed the phylogenetic tree. The updated tree in the MS.

3. Add PHI analysis for all new species, if do not have species in the same clade, you can choose species from nearby clades.

R: Thanks for your suggestion. We updated PHI analysis for every new species in the MS.

4. Did you isolate pathogens? Or do you only observe fruiting bodies from leaf spots? If isolated, please mention it clearly in the methods.

R: Thank you so much. We only isolated fungi from fruiting bodies on leaf spots.

5. In descriptions, I recommend rounding off small values, because 0.5 μm is beyond our eye capacity yet your software for measuring is automated. Also, descriptions are consists of different type font types while considering the avg symbol.

R: Thanks for your suggestion. We carefully checked the manuscript and corrected all measurement values and font types.

6. Since there are a lot of new species, I do not think we required descriptions of new hosts, you can only include material examined and notes.

R: Thanks for your suggestion. We agreed with you and deleted descriptions and photo-plates for new records.

7. I have several problems related to PHI analysis; in this case, I would prefer to see the alignment. However, the final alignment of the paper is not available for revision.

R: Thank you so much. We provided the final alignment in supplementary file1, namely, *N*-Alignment and *P*-Alignment. However, since the journal only accepts Excel and PDF formats as supplementary materials, we cannot provide FASTA

format.

8. are you sure PHI analysis is a GCPSR? PHI is an independent approach to understand the recombination, yet the GCPSR has its own meaning

R: Thanks for your reminder. PHI analysis is not a GCPSR. GCPSR used to identify independent evolutionary lineages. PHI is to check for recombination. We corrected it.

9. i do not agree with the morphology is unreliable. authors need to read and choose the facts carefully. this happened because everyone is totally relying on phylogenetic species

R: Thank you so much. We agree with you. We are not saying that morphology is unreliable, but that there are doubts about only relying on the host and conidial color to identify species. we have changed “With the development of DNA-based phylogenetic analyses, the traditional classification system has been proved unreliable.” into “With the development of DNA-based phylogenetic analysis, the traditional classification system based on host and conidial color is controversial.”

10. “For pathogens and saprophytes, single spore isolations were used to obtain pure cultures, following the methods described by Senanayake et al. (2020).” this sentence need to re write, 1st pathogens will be isolated and then do single spore to get pure cultures? did all pathogens isolates sporulated? to do the single spore isolation as saprobe samples

R: Thank you so much. Usually, the diseased specimens we collect have fruiting bodies. We isolated these fungi using single spore isolation rather than tissue isolation. Some of them sporulated.

11. “TEF” this should have 1 alpha. “TUB” it would be nice to abbreviate in to lower case italic

R: Thank you very much. We changed every “TEF” to “*tef1-α*” and “TUB” to “*tub2*” in the MS.

12. “(Maharachchikumbura et al. 2016b, Liu et al. 2017, Nozawa et al. 2017, Norphanphoun et al. 2019, Prasannath et al. 2021)” seems you are missing the most recently published papers. De Silva et al 2021; Yang et al 2021 please follow the other most recently published papers

R: Thank you so much. We added recently published papers on page 6, line 8 as follows: (De Silva et al 2021, Yang et al 2021, Chaiwan et al. 2022, Crous et al. 2022, Fiorenza et al.2022, Gu et al. 2022).

13. “ \bar{x} ” check these throughout the paper and correct accordingly, do not use as a picture

R: Thank you so much. We checked these throughout the MS and corrected them.

14. “there are 4 base pairs differences between GUCC 814 and GUCC 21001 in the ITS gene and 10 base pairs differences in the TEF gene.” what about tub?

R: Thank you so much. There are two base pairs differences in the *tub2* gene.

15. “Due to the lack of enough closely related species, the GCSPR was not used to evaluate its placement” this can not be accepted, you can use several different taxa which are not in the same clade yet nearby to make sure. if they are different, indeed will give a different values

R: Thanks for your suggestion. We redid this analysis and put the final result in the MS (Fig 2).

10. “However, they did not introduce it as a new species due to the lack of neighboring species to compare the morphology.” are you sure this is the reason?

follow this paper to understand what is exactly going on with this genus

Gerardo-Lugo, S.S.; Tovar-Pedraza, J.M.; Maharachchikumbura, S.S.N.; Apodaca-Sánchez, M.A.; Correia, K.C.; Saucedo-Acosta, C.P.; Camacho-Tapia, M.; Hyde, K.D.; Marraiki, N.; Elgorban, A.M.; et al. Characterization of *Neopestalotiopsis* species associated with mango grey leaf spot disease in Sinaloa, Mexico. *Pathogens* 2020

R: Thank you so much. We checked Jiang et al. (2021) and found that they introduced their taxa as *Neopestalotiopsis* sp.1. rather than new species due to a lack of distinguished characters from close clades. We read Gerardo-Lugo et al. (2020) following your suggestion and agreed with you and keep these taxa as *Neopestalotiopsis* sp.1.

16. “*Neopestalotiopsis chinensis* appears to be a common phytopathogen as it has been found in leaf spots on different plants.” can you confirm or you suggest it could be

R: Thank you so much. We corrected this sentence to “we speculate *Neopestalotiopsis* sp.1 could be a common phytopathogen as it has been found in leaf spots on different plants.”

17. “*Conidia* 17.0–22.0(–24.2) × 5.0–8.2” throughout the MS, round off your values.

R: Thank you so much. We checked throughout the paper and corrected these values.

18. “Part 2: *Pestalotiopsis*” add genetic authority

R: Thank you so much. We added genetic authority for these three genera on page 12, line 4, page 26, line 28 and page 36, line 13 following your suggestions.

Thank you so much!

Best wishes,

Ya-Ru Sun

Ruvishika S. Jayawardena

Jing-E Sun

Yong Wang

November 23, 2022

Prof. Yong Wang
Guizhou University
Department of Plant Pathology, Agriculture College, Guizhou University
Guiyang 550025
China

Re: Spectrum03987-22R1 (Pestalotioid species associated with medicinal plants in southwest China and Thailand)

Dear Prof. Yong Wang:

Your manuscript has been accepted, and I am forwarding it to the ASM Journals Department for publication. You will be notified when your proofs are ready to be viewed.

Sincerely,

Florian Freimoser
Editor, Microbiology Spectrum
